# Single-base tiled screen unveils design principles of PspCas13b for potent and off-target-free RNA silencing

Wenxin Hu [1,2], Amit Kumar[1,2,8], Syed Faraz Ahmed[3,4], Shijiao Qi[1,2], David K. G. Ma[1,2,5], Honglin Chen[1,2], Gurjeet J. Singh[1,2], Joshua M. L. Casan [1,2], Michelle Haber[6,7], Ilia Voskoboinik[1,2], Matthew R. McKay[3,4], Joseph A. Trapani[1,2], Paul G. Ekert [1,2,5,6,7] & Mohamed Fareh [1,2] ✉

The development of precise RNA-editing tools is essential for the advancement of RNA therapeutics. CRISPR (clustered regularly interspaced short palindromic repeats) PspCas13b is a programmable RNA nuclease predicted to offer superior specificity because of its 30-nucleotide spacer sequence. However, its design principles and its on-target, off-target and collateral activities remain poorly characterized. Here, we present single-base tiled screening and computational analyses that identify key design principles for potent and highly selective RNA recognition and cleavage in human cells. We show that the de novo design of spacers containing guanosine bases at precise positions can greatly enhance the catalytic activity of inefficient CRISPR RNAs (crRNAs). These validated design principles (integrated into an online tool, https://cas13target.azurewebsites.net/) can predict highly effective crRNAs with ~90% accuracy. Furthermore, the comprehensive spacer–target mutagenesis revealed that PspCas13b can tolerate only up to four mismatches and requires ~26-nucleotide base pairing with the target to activate its nuclease domains, highlighting its superior specificity compared to other RNA or DNA interference tools. On the basis of this targeting resolution, we predict an extremely low probability of PspCas13b having off-target effects on other cellular transcripts. Proteomic analysis validated this prediction and showed that, unlike other Cas13 orthologs, PspCas13b exhibits potent on-target activity and lacks collateral effects.

Precise editing of cellular transcriptomes with sequence-specific RNA-targeting tools is crucial for understanding biological and pathological processes and has the potential to transform the treatment of various genetic disorders[1]. Although conventional eukaryotic RNA interference (RNAi) can achieve targeted RNA silencing[2], its utility is limited by inherent permissive target recognition, which causes widespread off-target effects[3–6]. A novel class of programmable prokaryotic RNA nucleases with a more stringent target recognition has been recently discovered, which confers sequence-specific antiviral immunity to bacteria and archaea[7–9]. In this class of nucleases, target RNA recognition and silencing is mediated by type VI CRISPR (clustered regularly interspaced short palindromic repeats) effectors called Cas13 (refs. 6–9). In this system, a programmable segment of the CRISPR RNA (crRNA) called the 'spacer' guides the Cas13 protein to selectively recognize a complementary target RNA through RNA–RNA base pairing, triggering HEPN (higher eukaryotes and prokaryotes nucleotide-binding)

nuclease activation and target RNA degradation[10,11]. The decoupling of target binding and nuclease activity in Cas13 presents an opportunity to engineer chimeric proteins by fusing catalytically inactive (dead) Cas13 (dCas13) with other RNA editors, enabling various targeted RNA manipulation applications[8,12–19].

The RNA-targeting capabilities of Cas13 are highly efficient and reversible, making it a promising modality for specifically suppressing target transcripts without the risk of permanently altering the genome in both somatic and germline cells[20–22]. The long spacer sequence (22–30 nucleotides) of Cas13 enzymes contributes to their attractiveness, as it could convey rigorous target recognition and limited off-target effects compared to classical eukaryotic RNAi and CRISPR–Cas9 tools[22–24]. Despite their many advantages, some Cas13 orthologs have been reported to exhibit collateral nuclease activity. This occurs when spacer–target base pairing activates indiscriminate degradation of RNA molecules in the vicinity of the enzyme's HEPN nuclease domain[8,9]. This process was first described in vitro and in bacteria[9,15,16,25,26] but the extent of collateral activity in mammalian cells remains controversial[6,27–30]. Among the known Cas13 orthologs, RfxCas13d has emerged as a potent enzyme for RNA editing in part because of its compact size and amenability to viral vector-mediated delivery[30–34]. However, RfxCas13d appears to possess nonspecific collateral activity that degrades proximal RNA molecules, thereby attenuating its specificity[28,35–39].

In our effort to identify a Cas13 variant that is highly specific, we turned our focus to a poorly characterized Cas13 ortholog called PspCas13b, which belongs to a distinct subgroup of the Cas13 family[11,40]. This subgroup possesses an inverted orientation of the spacer and direct repeat region, as well as limited protein sequence homology with RfxCas13d and other Cas13 orthologs[11,40,41]. PspCas13b also possesses one of the longest spacer sequences (30 nucleotides), which is predictive of superior specificity[40,41]. However, the development of PspCas13b for precise RNA editing in mammalian cells is limited by the poor understanding of the molecular principles governing crRNA design and target recognition, as well as the status of its on-target, off-target and collateral activity.

To gain insight into the molecular mechanisms of PspCas13b, we developed a single-base tiled crRNA screen that probes spatial target accessibility with single-nucleotide resolution. Surprisingly, we found that the potency of neighboring crRNAs varied significantly and that shifting the target binding site by just a single base could completely abolish silencing activity. We identified consensus motifs within the spacer sequence that are critical for potency and found that incorporating these motifs into otherwise poorly effective crRNAs can enhance their abundance and catalytic activity in cells and in vitro. We validated these design principles by successfully silencing various transcripts in human HEK 293T cells and developed an online tool (https://cas13target.azurewebsites.net/) that can predict highly potent crRNAs for efficient and specific gene silencing with over 90% accuracy. Our comprehensive mutagenesis studies also revealed the molecular basis of spacer–target base pairing, where PspCas13b can tolerate 3–4

nucleotide mismatches within its 30-nucleotide-long spacer–target duplex. Beyond this threshold, its activity becomes largely impaired, providing a rational probabilistic model that explains its extremely high specificity. Importantly, our proteomic analysis reaffirmed this specificity by revealing efficient on-target silencing and a lack of any off-target or collateral activity against the human proteome of HEK 293T cells. Overall, these findings provide important insights into the molecular mechanisms underlying PspCas13b activity and pave the way for the development of more effective and specific RNA-editing strategies.

## Results

### The silencing efficiency of various crRNAs is highly variable
To elucidate PspCas13b crRNA design principles, we developed a quantitative fluorescence-based silencing assay, in which we targeted the transcript of the mCherry reporter gene. To achieve this, we cotransfected HEK 293T cells with three plasmids encoding mCherry, PspCas13b-BFP and either nontargeting (NT) or mCherry-targeting crRNAs (Fig. 1a). Fluorescence microscopy analysis of cells transfected with mCherry-targeting crRNAs showed pronounced silencing activity in contrast to no appreciable silencing in cells expressing NT crRNAs. However, when we used a second crRNA with a spacer sequence fully matching the coding sequence of mCherry RNA, we did not observe any silencing activity (Fig. 1b). This contrast in the silencing efficiency of the two crRNAs led us to question whether parameters such as the efficiency of crRNA transcription, crRNA loading, spacer nucleotide composition, target accessibility and the presence of a potential protospacer-flanking sequence (PFS) may influence the efficiency of PspCas13b. To test these hypotheses, we designed 16 crRNAs that fully base pair with the coding sequence of the mCherry mRNA at various positions (Fig. 1c). To accurately determine the silencing efficacy of each crRNA, we performed crRNA dose-dependent silencing assays in which cells were transfected with 0, 1, 5 and 20 ng of each of the 16 mCherry-targeting crRNAs. We noticed marked differences in the silencing efficacy of the various crRNAs, even when they were designed to target neighboring RNA locations (Fig. 1d and Extended Data Fig. 1a,b). We confirmed that the mCherry silencing is mediated by the nuclease activity of PspCas13b crRNA, as catalytically inactive dPspCas13b or crRNA alone failed to silence mCherry (Fig. 1e–g).

Overall, this finding suggested there are key determinants of PspCas13b silencing activity beyond target accessibility. Identifying such determinants is crucial for efficient reprogramming.

### Single-base resolution view on PspCas13b silencing activity
To further understand the spectrum of crRNA potency, we investigated the silencing activity of PspCas13b across a defined targeted region, reasoning that silencing efficiency is likely intrinsic to the spatial characteristics of the crRNA sequence and binding sites. We focused our study on crRNA 12 (binding position 455) and crRNA 16 (binding position 655), which exhibited high and moderate silencing, respectively (Fig. 1d). We designed three-nucleotide-resolution tiled

**Fig. 1 | The efficiency of PspCas13b silencing is highly variable between crRNAs. a**, Schematic of PspCas13b silencing assay used to monitor mCherry RNA target recognition and degradation. **b**, Representative fluorescence microscopy images show the silencing of mCherry transcripts with a potent or ineffective targeting crRNA versus an NT control crRNA in HEK 293T cells ($n = 3$). **c**, Schematic of the 16 PspCas13b crRNAs targeting various regions of mCherry RNA. The detailed spacer and direct repeat regions of a representative crRNA are indicated on the right. **d**, crRNA dose-dependent silencing of mCherry transcripts with either NT (orange) or targeting (gray) crRNAs. Data points in the graph are averages of the normalized mean fluorescence from four representative fields of view imaged in $n = 3$ or 4 replicates (crRNAs 1, 4, 8 and 9 and 14, $n = 4$; crRNAs NT, 2, 3, 5, 6, 7, 10, 11, 12, 13, 15 and 16, $n = 3$). The data are represented in arbitrary units (AU). Errors are the s.e.m. and $P$ values of a

one-way ANOVA are indicated (95% confidence interval). NS, not significant. **e**, Silencing of mCherry transcripts with active PspCas13b, dPspCas13b or crRNA only (no PspCas13b) and either NT crRNA or crRNA 12. Data points in the graph are averages of the normalized mean fluorescence from four representative fields of view imaged in $n = 3$ replicates. The data are represented in AU. Errors are the s.e.m. and $P$ values of an unpaired two-tailed Student's $t$-test are indicated (95% confidence interval). **f**, RT–qPCR assays measuring the silencing efficiency of either NT crRNA or crRNA 12 by active PspCas13b, dPspCas13b or crRNA only ($n = 3$). Data are the normalized means and errors are the s.e.m. Results were analyzed by an unpaired two-tailed Student's $t$-test and $P$ values are indicated (95% confidence interval). **g**, Western blot analysis examining the expression level of active PspCas13b or dPspCas13b and mCherry proteins in HEK 293T cells expressing either crRNA 12 or NT crRNA ($n = 3$).

crRNAs spanning a 30-nucleotide target region surrounding the crRNA 12 and crRNA 16 binding positions (Fig. 2a; and Extended Data Fig. 2a). In this tiled design, the binding sites of each adjacent crRNA are spaced by just three nucleotides; thus, their silencing profiles should reveal the relationship between efficacy and target accessibility. We again observed considerable heterogeneity in the potency of these tiled

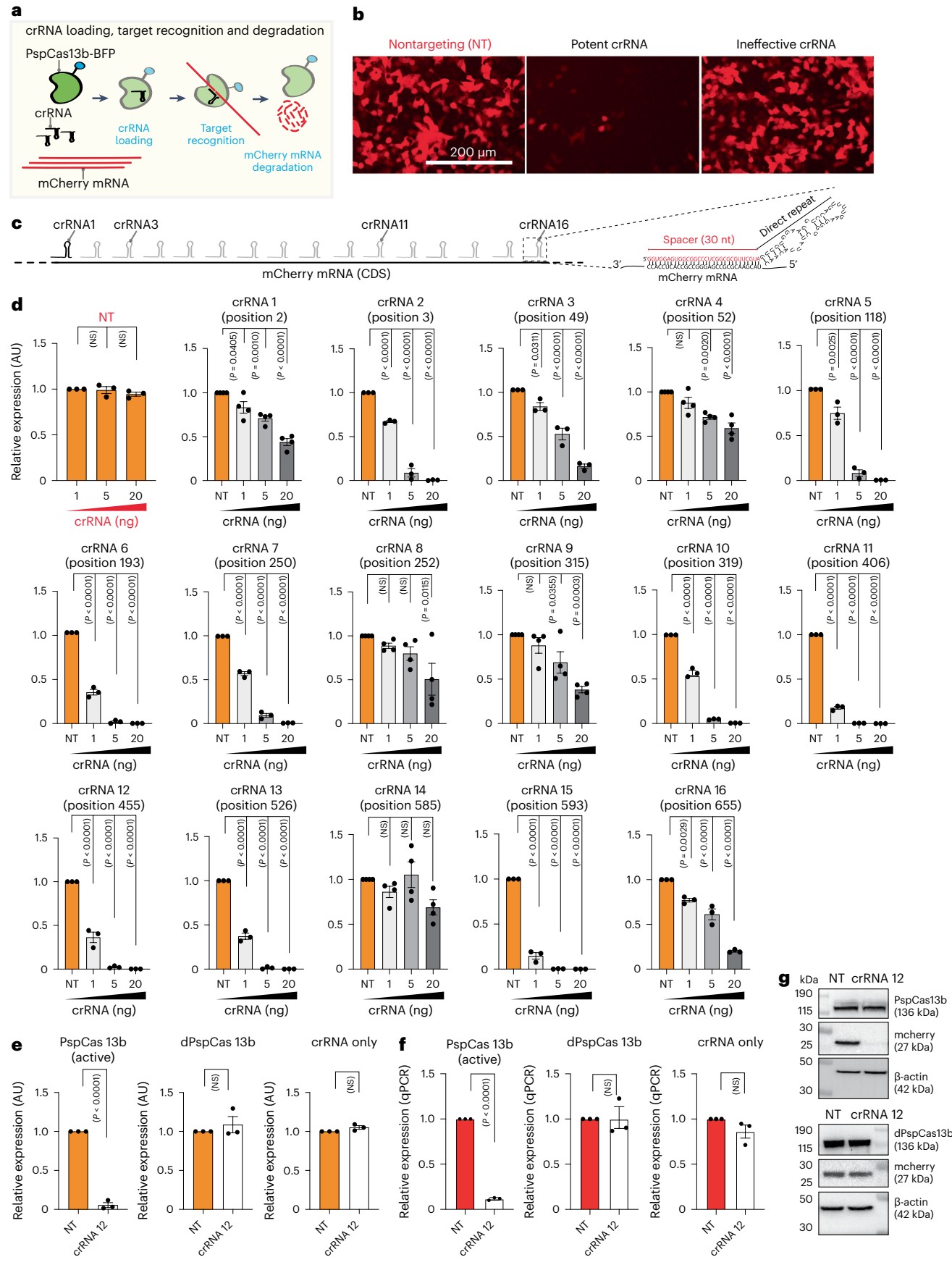

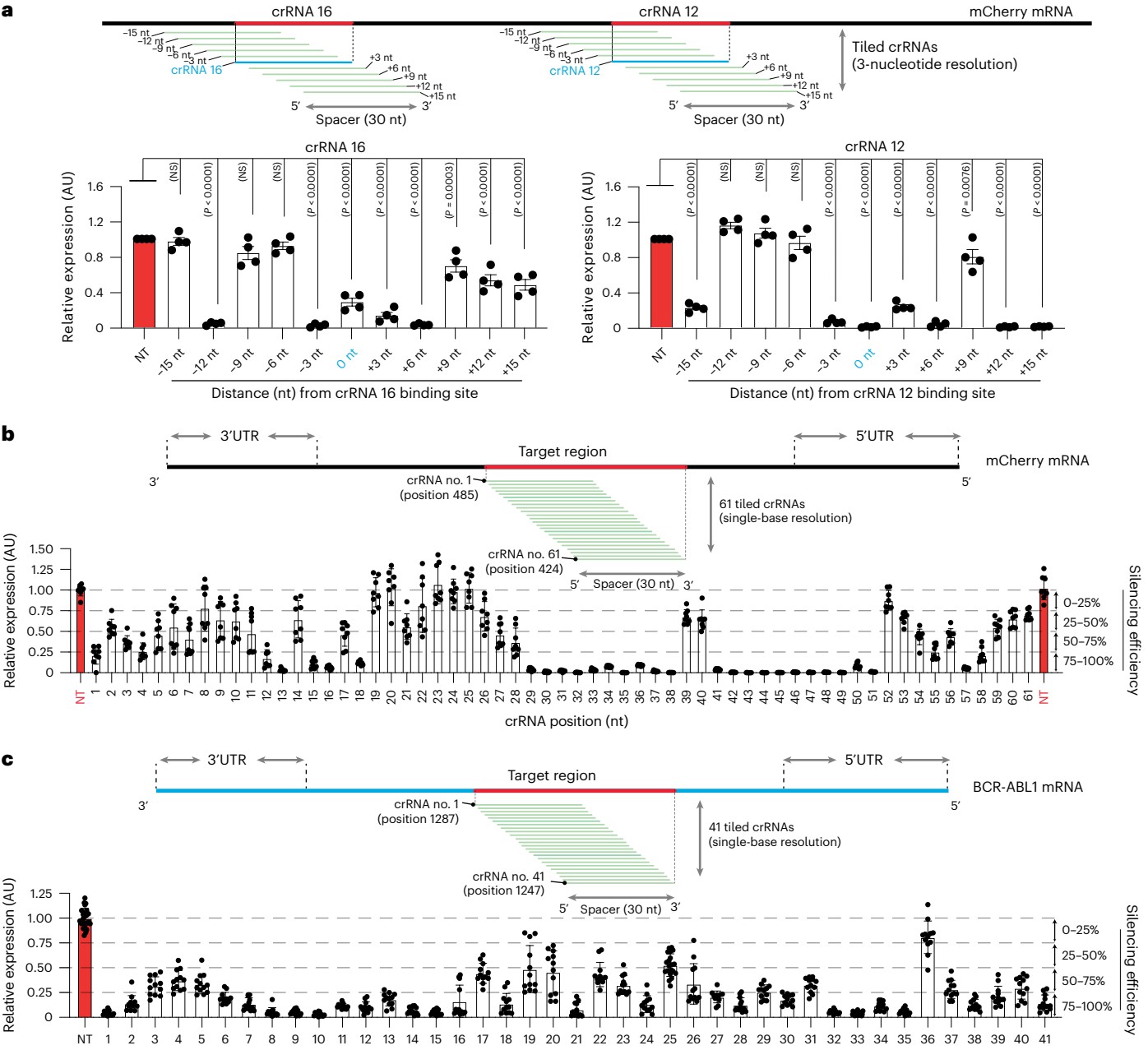

**Fig. 2 | Variations in the efficiency of single-base tiled crRNAs reveal key determinants of potency beyond target accessibility. a**, Schematic illustrating mCherry RNA regions covered by three-nucleotide-resolution tiled crRNAs. Quantification of silencing efficiency with tiled crRNAs targeting the mCherry regions surrounding crRNA 12 (left) and crRNA 16 (right). Data points in the graph are averages of the normalized mean fluorescence from four representative fields of view per experiment imaged in $n = 4$ replicates. The data are represented in AU. Errors are the s.e.m. and $P$ values of a one-way ANOVA are indicated (95% confidence interval). **b**, Schematic showing the sequence of mCherry RNA covered by 61 single-nucleotide tiled crRNAs around the targeted region of crRNA 12. The graph quantifies the silencing efficiency obtained with 61 tiled crRNAs in HEK 293T cells. Data points in the graph are the normalized mean

fluorescence from four representative fields of view imaged in $n = 2$ replicates. UTR, untranslated region. The data are represented in AU. Errors are the s.d. with the 95% confidence interval. **c**, eGFP fluorescence reporter assay using 41 single-nucleotide tiled crRNAs targeting BCR-ABL1(P190)-eGFP mRNA. The BCR-ABL1(P190)-eGFP mRNA was ectopically expressed in HEK 293T cells using plasmid transfection. This screen with single-base tiled crRNAs aimed to uncover the relationship linking RNA sequence, accessibility and PspCas13b silencing efficiency. The schematic shows the sequence of BCR-ABL1 mRNA covered by 41 tiled crRNAs and the RNA–RNA duplex formed by spacer–target interaction. Data points in the graph are the normalized mean fluorescence from four representative fields of view imaged in $n = 3$ replicates. The data are represented in AU. Errors are the s.d. with the 95% confidence interval.

crRNAs despite their physical proximity, with some adjacent crRNAs demonstrating contrasted silencing efficacy. These data indicated that physical barriers such as RNA-binding proteins or structured RNA motifs are unlikely to explain the fluctuation in silencing between spatially adjacent crRNAs that are separated by just three nucleotides (Fig. 2a and Extended Data Fig. 2a–c).

To further enhance our understanding, we maximized the spatial resolution of this approach by designing 61 single-base tiled crRNAs targeting the mCherry coding sequence (Fig. 2b). Consistent with previous data, we again observed markedly diverse silencing profiles of neighboring crRNAs. For instance, crRNA 13 achieved silencing exceeding 95% efficiency but shifting the targeted region by only

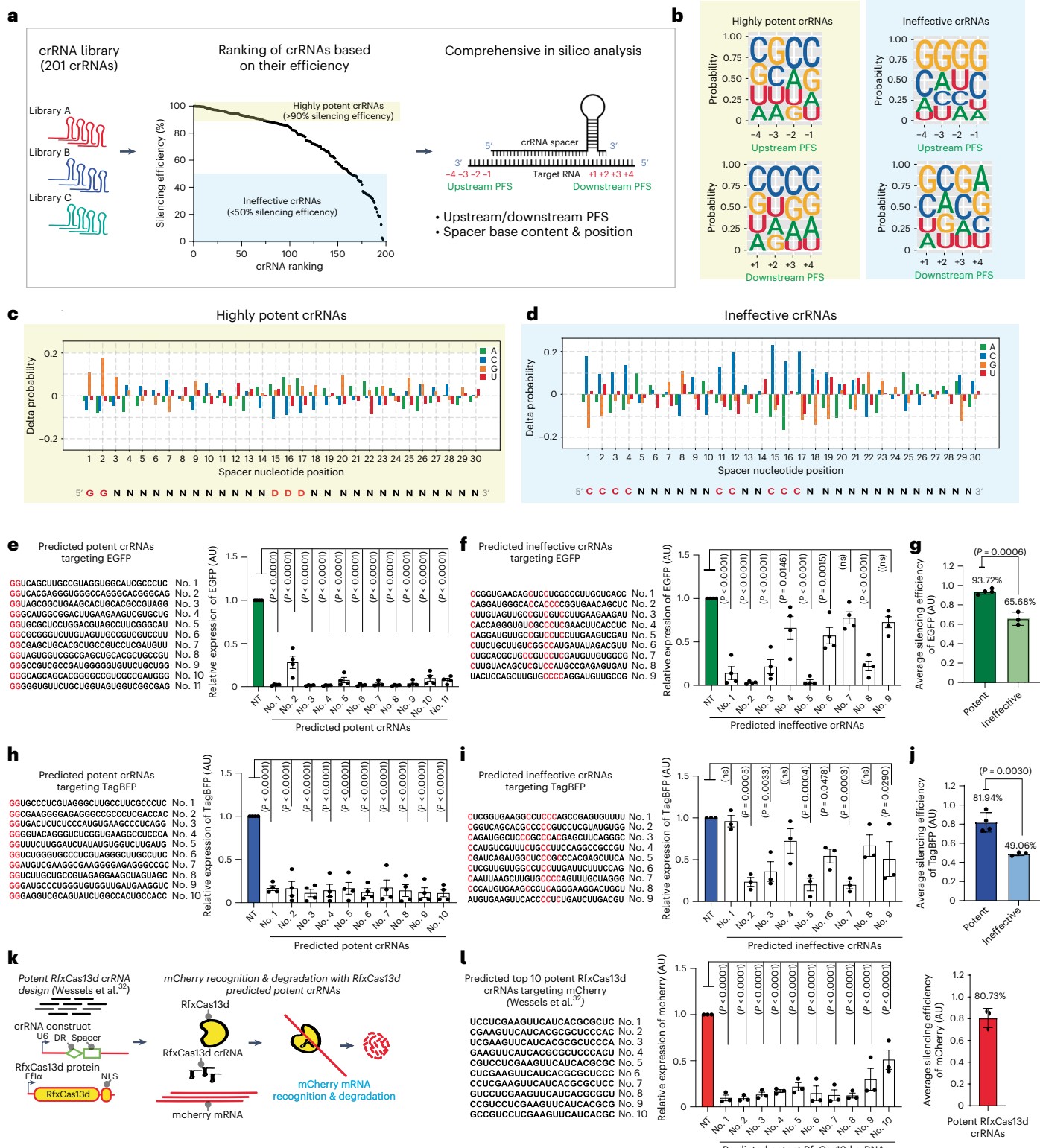

one nucleotide (crRNA 14) dramatically reduced the efficiency to ~30%. Similarly, crRNA 51 yielded ~99% silencing efficiency while its adjacent crRNA 52 did not show any appreciable silencing activity (Fig. 2b). We observed similarly high fluctuations in the silencing profiles of single-base crRNAs targeting the oncogenic fusion transcript BCR-ABL1. For instance, the binding sites of crRNAs 14 and 15 have 29 overlapping consecutive nucleotides and are separated by just a single nucleotide yet their silencing efficiency is drastically different (Fig. 2c).

These data strengthened our contention that silencing efficacy is unlikely to be solely dependent on the target accessibility and that other factors including specific nucleotide positions within the spacer or target, as well as a possible PFS, may influence key steps of target silencing such as crRNA transcription, loading and target recognition.

### Key design principles for potent RNA silencing
In an effort to uncover fundamental principles that dictate PspCas13b silencing efficiency, we expanded our dataset by analyzing the silencing

**Fig. 3 | Computational analysis and functional validation reveal nucleotide positions within the spacer as key determinants of silencing efficiency.** **a**, Bioinformatics pipeline exploring the parameters affecting PspCas13b silencing. PFS positions are indicated. A total of 201 crRNAs are ranked by silencing efficiency. Highly potent (>90% efficiency, yellow box) and ineffective (<50% efficiency, blue box) crRNAs were analyzed for PFS and spacer nucleotide positions. **b**, Position weight matrices (PWMs) depicting the positional nucleotide probabilities of upstream or downstream PFSs in the potent (left) or ineffective (right) crRNAs. **c,d**, Delta nucleotide probabilities of the potent (**c**) and ineffective (**d**) spacer sequences that compare filtered spacer nucleotide positions to the baseline nucleotide distribution. The red color in the spacer sequence highlights the position of key residues. **e,h**, Prospective design and validation of predicted potent crRNAs harboring a G-G motif at the 5′ end of spacers targeting eGFP (**e**) and TagBFP (**h**) (n = 4). **f,i**, Predicted ineffective crRNAs lacking a 5′ G-G motif and harboring C bases in the central region of spacers targeting eGFP (**f**; n = 4) and TagBFP (**i**; n = 3). The red color in the spacer sequence

indicates nucleotide insertion or substitution. Data points are the averaged mean fluorescence from four representative fields of view per condition. The data are represented in AU. Errors are the s.e.m. and P values of a one-way ANOVA are indicated (95% confidence interval). **g,j**, The average silencing efficiency of predicted potent and ineffective crRNAs targeting eGFP (**g**) and TagBFP (**j**) transcripts (potent crRNAs, n = 3; inefficient crRNAs, n = 4). Data are the normalized means and errors are the s.e.m. Results were analyzed by an unpaired two-tailed Student's t-test (95% confidence interval). **k**, RfxCas13d silencing assay to target mCherry transcript in HEK 293T cells using the top ten potent crRNAs predicted by the RfxCas13d guide prediction platform (Wessels et al.[32]). NLS, nuclear localization sequence. **l**, Left, the silencing efficiency of the top ten RfxCas13d crRNAs. Data points represent the averaged mean fluorescence from four representative fields of view per condition (n = 3) in AU. Errors are the s.e.m. and one-way ANOVA P values are indicated (95% confidence interval). Right, the average silencing efficiency of predicted crRNAs. Data points represent the normalized means and errors are the s.e.m. (95% confidence interval) (n = 3).

profiles of 201 individual crRNAs targeting various transcripts[17]. First, we questioned whether crRNA or target folding, spacer–target stability and spacer nucleotide content correlate with PspCas13b potency. The data suggest that the folding of the crRNA and the targeted sequence into complex secondary structures can only moderately limit Psp-Cas13b silencing efficiency, possibly perturbing crRNA loading or target accessibility (Extended Data Fig. 3). Interestingly, we observed a strong negative correlation between C nucleotide enrichment in the spacer sequence and crRNA potency (r = −0.30, P < 0.0001; Extended Data Fig. 3h). In contrast, spacers that are rich in G nucleotides showed a positive correlation with crRNA potency (Extended Data Fig. 3i).

Next, we pooled these 201 crRNAs and ranked them by silencing efficiency. The crRNAs that achieved >90% silencing efficiency were designated as potent crRNAs and those with less than 50% efficiency were considered ineffective crRNAs. The crRNAs with ambiguous silencing profiles (efficiencies ranging from 50% to 90%) were excluded from the analysis. We sought to identify molecular features capable of differentiating potent and ineffective crRNA cohorts (Fig. 3a). Many CRISPR–Cas variants possess an upstream or downstream PFS that restricts targeting activity and prevents degradation of their own nucleic acids[42]. To investigate the existence of a PFS that could constrain PspCas13b silencing, we generated weight matrix plots that analyzed nucleotide composition at each position four bases upstream and downstream of the targeted sequence in the highly potent and ineffective cohorts of crRNAs. We found no detectable bias in nucleotide composition at various target flanking sites, suggesting that PspCas13b activity is not subject to PFS motifs (Fig. 3b).

Lastly, we questioned whether the nucleotide composition of the spacer could influence PspCas13b silencing efficiency. Nucleotide content analysis revealed an enrichment of G bases in the potent crRNA group and an enrichment of C bases in the ineffective crRNA cohort (Extended Data Fig. 4c–e), indicating that G-enriched spacers are associated with higher potency, whereas C-enriched spacers are associated with low potency.

To elucidate the significance of G and C bases at specific positions within the spacer sequence, we performed unbiased analyses of nucleotide composition at all 30 positions of the spacer in both highly potent and ineffective crRNA cohorts (Fig. 3c,d and Extended Data Fig. 4f,g). The analysis revealed marked differences in nucleotide positions between the two crRNA cohorts. We show that G bases at the 5′ end, particularly a G-G sequence at the first and second positions, was strongly associated with highly potent crRNAs. Conversely, G bases were depleted and C bases were enriched at the 5′ end of spacers in the ineffective crRNA cohort. In addition to this C-rich motif at the 5′ end of ineffective crRNAs, we also identified a significant enrichment of C bases at positions 11, 12, 15, 16 and 17 (Fig. 3c,d and Extended Data Fig. 4f,g). These data revealed key nucleotide positions that may determine the potency of crRNAs, which could serve as predictive parameters of crRNA potency.

### Functional validation of crRNA prediction and design

The above in silico analysis enabled us to identify consensus motifs within the spacer to predict potent and ineffective crRNAs. We postulated that potent crRNAs should include a G-G sequence at the first and second positions of the spacer and should lack C bases in positions 11, 12, 15, 16 and 17 (GGNNNNNNNNDDNNDDDNNNNNNNNNNNNNN, where D is a G, U or A nucleotide and N is any nucleotide). Conversely, crRNAs containing a C base in spacer positions 1, 2, 3, 4, 11, 12, 15, 16 and 17 are predicted to yield poor silencing efficiency (CCCCNNNNNNCCNNC-CCNNNNNNNNNNNNNN).

We tested the predictive accuracy of this spacer-based design through a prospective unbiased design of crRNAs targeting eGFP (enhanced green fluorescent protein) and TagBFP (blue fluorescent protein), two mRNA targets that we did not investigate previously. Notably, of the 21 predicted potent crRNAs, 20 achieved very high silencing efficiency of either eGFP or TagBFP mRNA. Conversely, the majority of predicted ineffective crRNAs failed to efficiently silence eGFP and TagBFP transcripts (Fig. 3e–j). By generating predictions on the basis of an existing dataset and verifying their accuracy in

**Fig. 4 | Incorporation of a 5′ G-rich motif enhances the potency of ineffective crRNAs. a–g**, Incorporation of a G-rich motif at the 5′ end of ineffective spacer sequences (crRNA39 (**a**), crRNA40 (**b**), crRNA24 (**c**), crRNA25 (**d**), crRNA26 (**e**), crRNA27 (**f**) and crRNA28 (**g**)) targeting mCherry through the insertion or substitution of a G base greatly enhanced their silencing efficiency. The red color in the spacer sequence indicates nucleotide insertion or substitution. Data points in the graph represent the averaged mean fluorescence from four representative fields of view per condition imaged (**a,b**, n = 3; **c–g**, n = 4). WT, wild type. The data are represented in AU. Errors are the s.e.m. and P values of a one-way ANOVA are indicated (95% confidence interval). **h**, Schematic of PspCas13b silencing assay performed in HEK 293T cells using IVT or chemically synthesized crRNAs with or without a 5′ G-G motif. **i,j**, Assessment of the silencing efficiency of IVT (**i**) and chemically synthesized (**j**) crRNA pairs that either fully base pair with the target

(WT) or harbor a mismatched 5′ G-G motif in their spacer sequence. The silencing efficiency was monitored at the 24-h and 48-h time points after transfection of crRNAs. Data points in the graph are the averaged mean fluorescence from four representative fields of view per imaged condition (n = 3). The data are represented in AU. Errors are the s.e.m. and P values of an unpaired two-tailed Student's t-test are indicated (95% confidence interval). **k**, Gel electrophoresis showing the expression profile of purified recombinant PspCas13b (n = 3). **l**, In vitro cleavage of a 100-nucleotide ssRNA (labeled with a 5′ 6-FAM probe) using recombinant PspCas13b loaded with crRNA 39 with or without a 5′ G-G motif at the 0-h, 2-h and 4-h time points (n = 1). **m**, In vitro RNA cleavage obtained with PspCas13b incubated with crRNA with or without a 5′ G-G motif and a target ssRNA labeled with 5′ 6-FAM. The arrows indicate cleaved RNA products. The cleavage was assessed at the 0-h and 2-h time points (n = 3).

previously unexplored transcripts, these findings illustrated that our design, which relies on spacer nucleotides, is both precise and applicable across different transcripts.

Previous studies used library screening and machine learning approaches to uncover the hidden design principles of RfxCas13d (refs. 30,32–34), the most commonly used Cas13 ortholog. We sought

to compare the efficiency of our design of PspCas13b to the benchmark crRNA design tool that is available for RfxCas13d (Fig. 3k)[32]. We selected the top ten predicted potent crRNAs for RfxCas13d targeting the coding sequence of the mCherry reporter and probed their silencing efficiency, which achieved an average silencing of 80.7% (Fig. 3l). Our PspCas13b design of potent crRNAs showed ~87.8% average

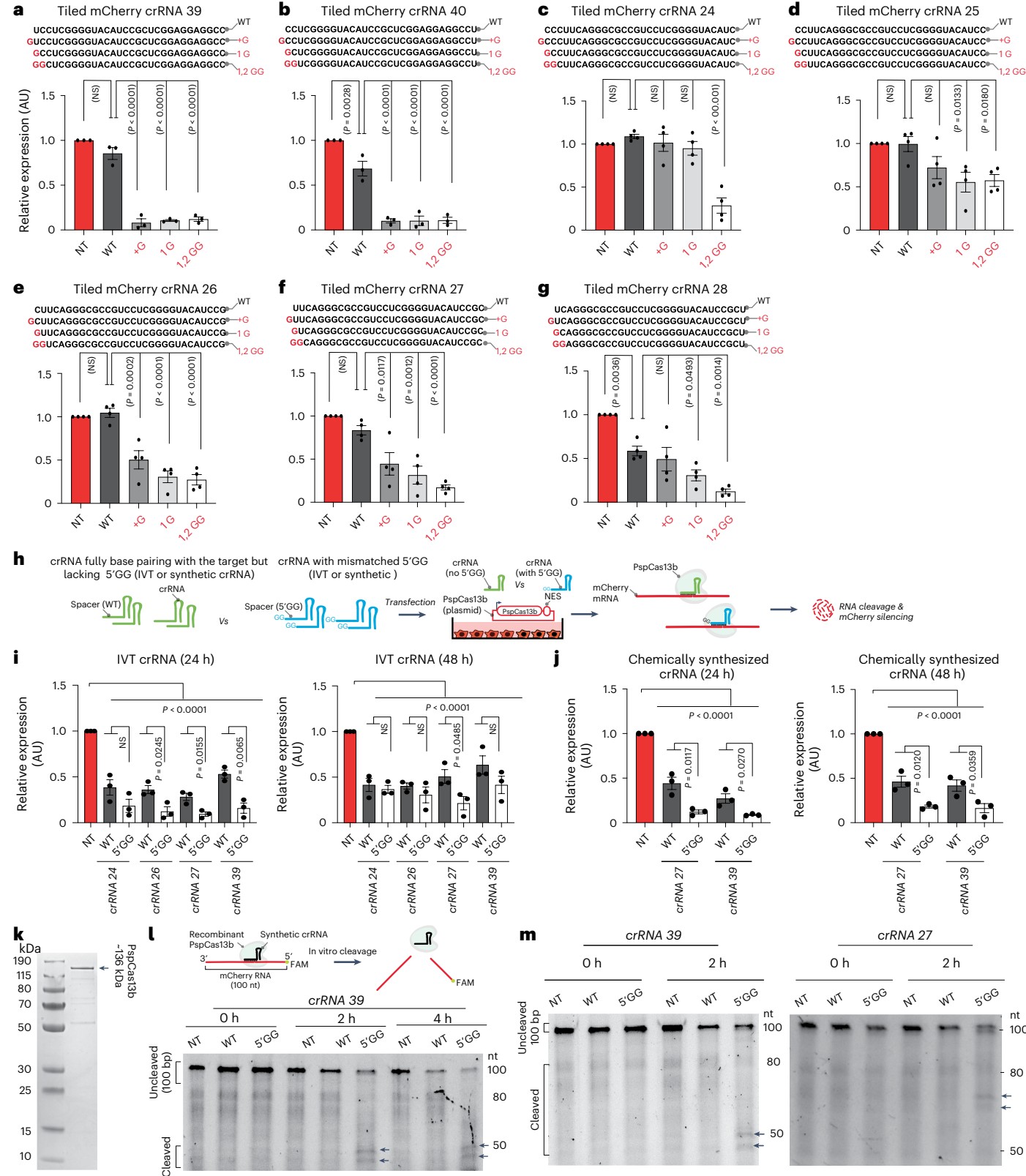

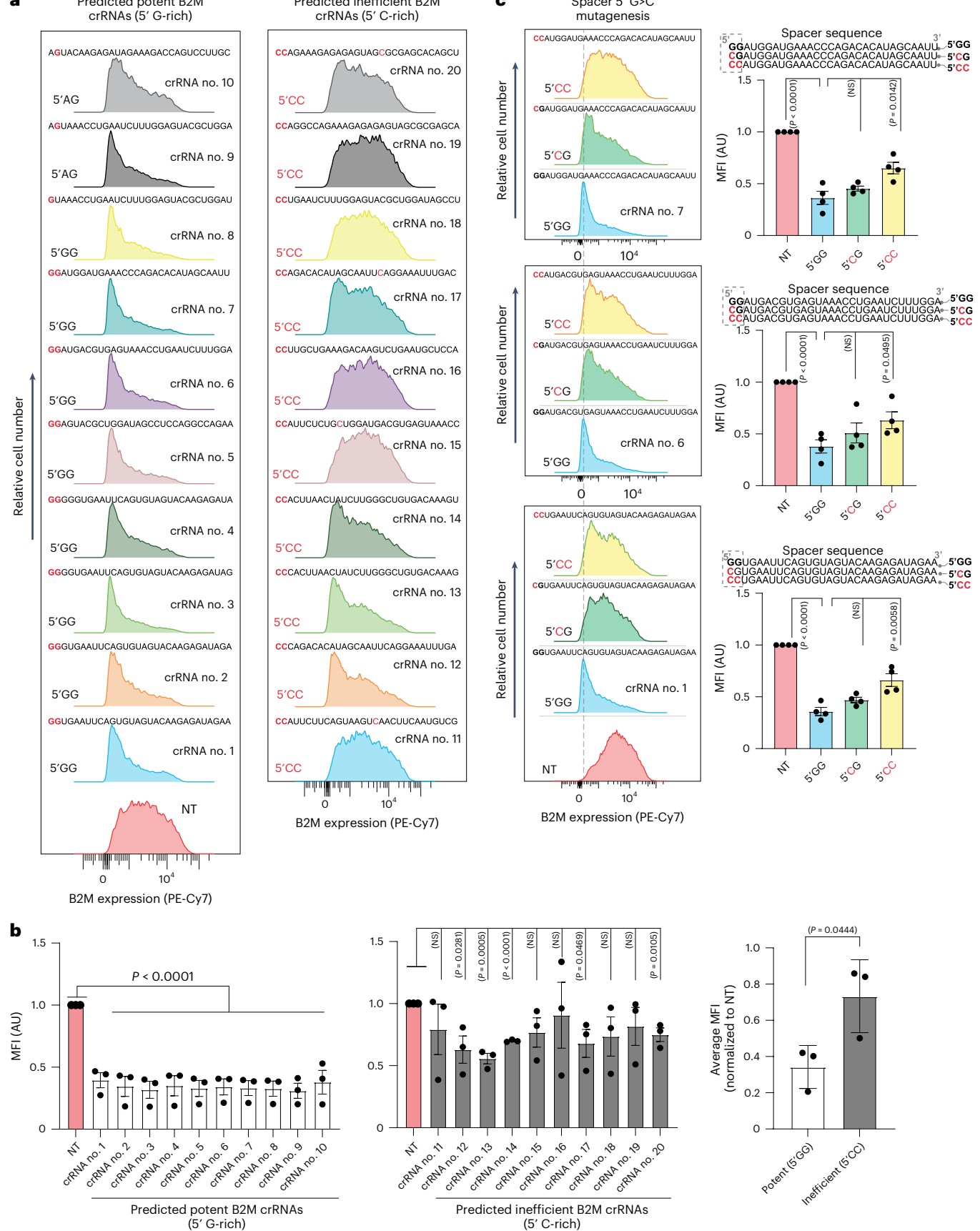

silencing efficiency (eGFP and TagBFP together; Fig. 3g,j), slightly outperforming the RfxCas13d design algorithm (Fig. 3l), which further validated the accuracy of our prediction tool.

To further investigate the enrichment of a G-rich motif at the 5′ end of potent crRNAs and C bases at the 5′ end of ineffective crRNAs, we hypothesized that altering these sequences in a bona fide spacer

**Fig. 5 | Validation of prediction accuracy against a human endogenous mRNA. a**, Representative flow cytometry histograms depicting the silencing activity of ten predicted potent crRNAs and ten predicted ineffective crRNAs targeting B2M mRNA compared to an NT crRNA. **b**, Quantification of the mean fluorescence intensity (MFI) of B2M is shown in the histograms below obtained from $n = 3$ replicates. The histogram on the right shows the average silencing efficiency of ten potent (white bars) and ten inefficient (gray bars) B2M crRNAs.

Data are the normalized means and errors are the s.e.m. Results were analyzed by an unpaired two-tailed Student's $t$-test and $P$ values are indicated (95% confidence interval). **c**, Flow cytometry plots showing the impact of G>C substitutions (red) at the first or first and second nucleotide positions of the spacer (dashed rectangles) of three potent crRNAs targeting B2M ($n = 4$). Data are the normalized means and errors are the s.e.m. Results were analyzed by an unpaired two-tailed Student's $t$-test and $P$ values are indicated (95% confidence interval).

sequence may either worsen or improve their silencing efficiency. First, we selected 11 crRNAs that possess a G-G sequence at the first and second positions of the spacer, which we altered to C-C by spacer mutagenesis. The data showed substantial compromise in the silencing efficiency of these crRNAs (Extended Data Fig. 5a). We also mutated one, two or three G bases at the 5′ end of the spacer to C and found that the substitution of three or two C bases at the 5′ end of the spacer reduced silencing by >99% and ~70%, respectively, while a single G>C base substitution at spacer positions 1, 2 or 3 had no significant effect on the potency of the crRNA (Extended Data Fig. 5b,c).

Next, we selected seven ineffective crRNAs lacking a G-G sequence at their 5′ end, and then modified them by inserting an additional G at the first position, substituting the first nucleotide to G or substituting the first and second nucleotides to G-G (Fig. 4a–g). Importantly, the data demonstrated that G sequences at the 5′ end of the spacer greatly increased the potency of crRNA despite the introduction of spacer–target mismatch (Fig. 4a–g).

A previous study indicated that promoters dependent on RNA polymerase III, such as U6, can achieve an increased transcription rate when the resulting small RNA possesses A or G bases at the 5′ end[43]. We questioned whether the improvement in silencing efficiency of crRNAs harboring a G-rich motif at their 5′ end could be secondary to changes in crRNA abundance. We quantified the expression levels of the original crRNA or mutated crRNAs harboring G motifs at their 5′ end using reverse transcription (RT)– qPCR. Although not statistically significant, we observed a trend of increased crRNA abundance when a G-rich motif was present at the 5′ end (Extended Data Fig. 6a–e).

To further understand whether a G-G sequence at the 5′ end can enhance crRNA potency beyond transcription upregulation, we in vitro transcribed (IVT) four pairs of crRNAs that possessed or did not possess a 5′ G-G motif (Extended Data Fig. 6f). A 15% PAGE and TapeStation analysis confirmed that the IVT crRNAs were 66 nucleotides in length, consistent with the size of synthetic crRNAs (purchased from Integrated DNA Technologies (IDT)) (Extended Data Fig. 6f).

We cotransfected equal amounts of these IVT crRNAs into HEK 293T cells together with the PspCas13b plasmid to compare their silencing efficiency (Fig. 4h). At 24 h after transfection, all four crRNAs with a 5′ G-G sequence achieved significantly higher silencing efficiency than their unmodified counterparts. At 48 h, one IVT crRNA containing a 5′ G-G sequence maintained significantly superior silencing activity compared with its unmodified counterparts (Fig. 4i). Similar results were obtained using synthetic commercial crRNAs with or without a 5′ G-G motif (Fig. 4h,j). Together, these experiments showed that the 5′ G-G motif further enhances the silencing activity of PspCas13b beyond augmented crRNA transcription.

### The 5′ G-G motif confers higher catalytic activity

We questioned whether the superior RNA silencing activity obtained with crRNAs harboring a 5′ G-G motif could be because of enhanced crRNA loading and/or increased cleavage activity. To test this hypothesis, we purified recombinant PspCas13b (Fig. 4k) and tested its crRNA loading and RNA cleavage activity in vitro. We first preincubated synthetic crRNAs with increasing concentrations of recombinant PspCas13b and performed electrophoretic mobility shift assays (EMSAs). The data indicated that PspCas13b could bind both crRNAs with a similar affinity regardless of the 5′ G-G motif (Extended Data Fig. 7). Next, we developed an in vitro cleavage assay by reconstituting a tertiary nucleoprotein complex containing recombinant PspCas13b, synthetic crRNA and a 100-nucleotide-long mCherry RNA as a target, which was labeled with 6-carboxyfluorescein (6-FAM) at its 5′ end. When loaded with a targeting crRNA (crRNA 39) containing a 5′ G-G motif, the recombinant PspCas13b exhibited potent cleavage of the target, which resulted in two RNA fragments with sizes ranging from 40 to 50 nucleotides. This cleavage pattern was absent when we used an NT crRNA or a targeting crRNA without PspCas13b protein (Extended Data Fig. 6g). When we compared the cleavage activity obtained with wild-type or mismatched 5′ G-G crRNAs, the latter exhibited a higher cleavage potency in vitro as evidenced by the appearance of two cleaved RNA fragments (Fig. 4l,m). Together, these data suggested that the incorporation of a 5′ G-G sequence enhances both crRNA transcription and PspCas13b cleavage activity. The results also indicated that when crRNA design choices are restricted, the de novo design of crRNAs incorporating mismatched G bases at these key positions can substantially increase their potency despite introducing nucleotide mismatches with the target.

### The 5′ G-G motif enables potent silencing of endogenous RNAs

Next, we aimed to determine whether the design principles we discovered using reporter fluorescent target transcripts could also apply to endogenous mRNA targets. To achieve this, we prospectively predicted ten potent crRNAs (5′ G-rich) and ten ineffective crRNAs (5′ C-rich) targeting the endogenous β-2-microglobulin (B2M) transcript that encodes a surface protein component of the major histocompatibility complex class I complex (Fig. 5a,b).

The findings supported our predictions, as all predicted potent crRNAs effectively downregulated the B2M protein, while the majority of predicted ineffective crRNAs failed to do so (Fig. 5a,b). We further showed that substituting the G bases of three potent B2M crRNAs with C bases at spacer position 1 or positions 1 and 2 compromised their silencing activity (Fig. 5c). Together, these data further supported the generalizability of our targeting rules and their relevance for silencing exogenous and endogenous transcripts.

**Fig. 6 | Comprehensive mutagenesis of the PspCas13b spacer–target interaction reveals the interface between mismatch tolerance and loss of activity. a–g**, Comprehensive analysis of spacer–target interaction examining the specificity and mismatch tolerance of PspCas13b at various positions of the crRNA spacer sequence. Perturbation of spacer–target interaction through spacer mutagenesis to introduce consecutive mismatches of 3, 6, 9, 12, 15, 18, 21, 24, 27 and 30 nucleotides at the 3′ end (**a**) and 5′ end (**b**) of the spacer. Perturbation of spacer–target interaction through spacer mutagenesis to introduce consecutive mismatches of six (**c**), five (**d**), four (**e**) and three (**f**)

nucleotides at various positions of the spacer. Perturbation of spacer–target interaction through spacer mutagenesis to introduce various numbers of nonconsecutive nucleotide mismatches at various positions (**g**). MSM, mismatch. The nucleotides in red highlight mismatch positions in the spacer sequence. Data points in the graph are averages of the mean fluorescence from four representative fields of view per condition imaged (**a,b**, $n = 5$; **c–f**, $n = 4$; **g**, $n = 3$). The data are represented in AU. Errors are the s.e.m. and $P$ values from a one-way ANOVA are indicated (95% confidence interval).

To facilitate the use of our optimized and validated spacer nucleotide-based design of potent crRNA, we created a user-friendly web page (https://cas13target.azurewebsites.net/) to assist the community with their silencing assays. This in silico tool requires only the target sequence as input to create single-base tiled spacer sequences and rank them on the basis of their predicted potency. The web application can also assess potential off-target transcripts within the human transcriptome for the top ten most potent spacers (see Methods for more details).

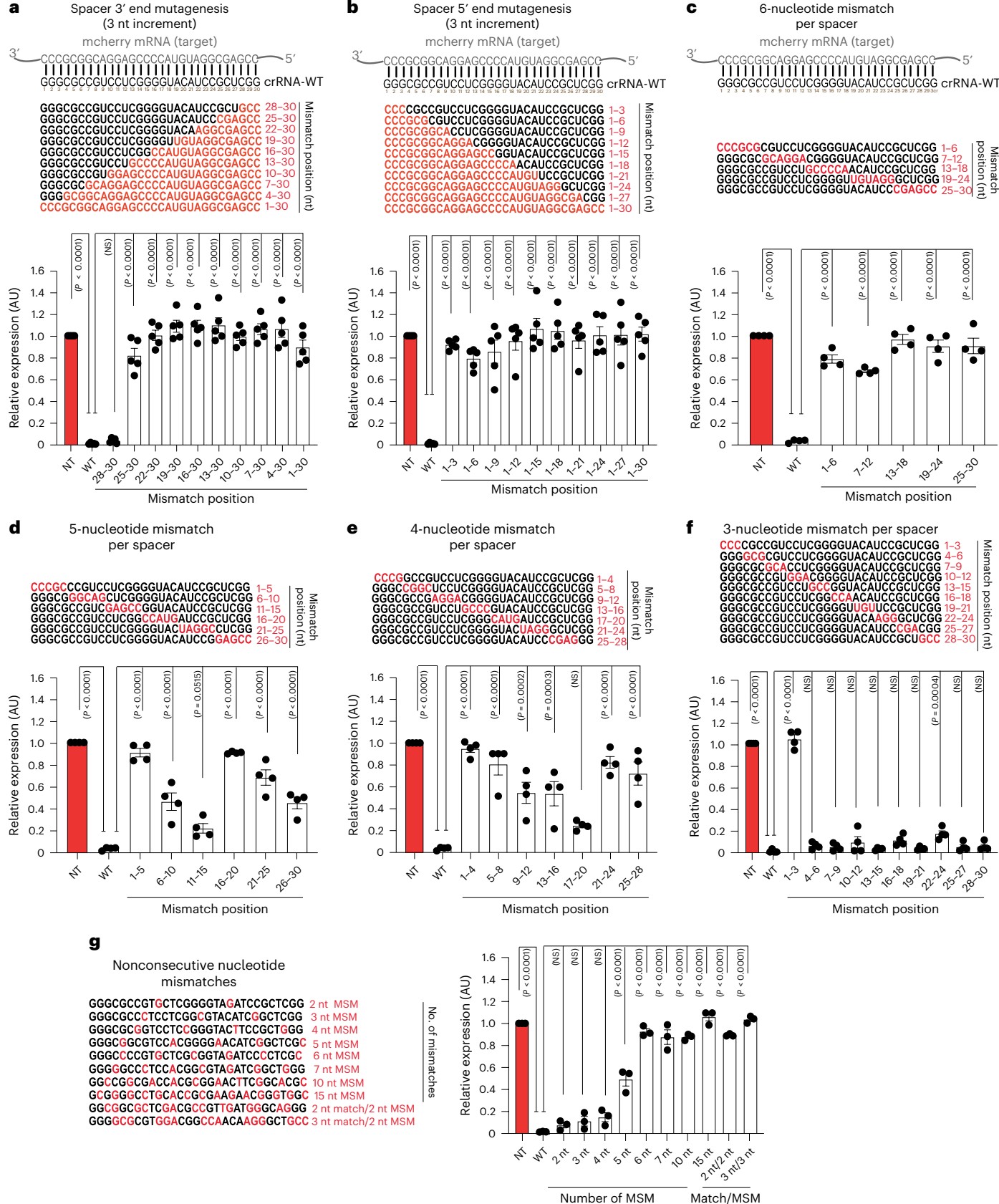

**Spacer mutagenesis reveals PspCas13b mismatch tolerance**

Understanding the specificity, off-targeting potential and capability of PspCas13b to discriminate between two transcripts that share extensive sequence homology is extremely important for evaluating the potential and limitations of PspCas13-based RNA silencing. To study crRNA spacer promiscuity and the consequent PspCas13b targeting resolution, we conducted a comprehensive spacer mutagenesis to introduce mismatches with the target at various spacer positions. First, we introduced successive mismatches of 3, 6, 9, 12, 15, 18, 21, 24, 27 and 30 nucleotides starting from the 3′ and 5′ ends of the spacer (Fig. 6a,b). The three-nucleotide mismatches at the 3′ end of the spacers (positions 28–30) did not affect the silencing efficiency of this crRNA, whereas mismatches greater than three nucleotides completely abrogated its silencing (Fig. 6a). In contrast to the 3′ end, all 5′ end mismatches resulted in a complete loss of silencing, including three-nucleotide mismatches at the 5′ end (Fig. 6b). According to our earlier findings (Extended Data Fig. 5), silencing loss consequent to the introduction of a three-nucleotide mutation at the 5′ end is likely attributable to the substitution of a G-G-G motif by a C-C-C sequence rather than spacer–target mismatch itself (Figs. 3–5).

We also created crRNA constructs harboring mismatches of six, five, four and three nucleotides at different spacer positions and probed their silencing efficiency in live cells (Fig. 6c–f). Overall, the six-nucleotide mismatches largely compromised the efficiency of PspCas13b regardless of the mismatch position (Fig. 6c). The five-nucleotide mismatches at positions 6–10, 11–15 and 26–30 exhibited a partial loss of silencing, while mismatches at positions 1–5, 16–20 and 21–25 led to a near-complete or complete loss of silencing (Fig. 6d). The four-nucleotide mismatches at positions 9–12, 13–16 and 17–20 retained partial silencing activity, whereas mismatches at positions 1–4, 5–8, 21–24 and 25–28 yielded a complete loss of silencing (Fig. 6e). Notably, crRNA constructs harboring three-nucleotide mismatches at various spacer positions were well tolerated and yielded no or minor loss of silencing, except for mutations at positions 1–3 that, as anticipated, led to a total loss of silencing, likely because of G-G-G removal at the 5′ end (Fig. 6f).

Whilst the preceding experiments established the tolerance for consecutive spacer–target mismatches, we questioned whether the silencing profile of nonconsecutive mismatches may differ. We destabilized the spacer–target interaction by introducing nonconsecutive matches of 2, 3, 4, 5, 6, 7, 10 and 15 nucleotides spread throughout the spacer (Fig. 6g). We noticed that nonconsecutive mismatches of 2–4 nucleotides were tolerated and led to a negligible loss of silencing. However, nonconsecutive mismatches of more than four nucleotides led to a substantial or complete loss of silencing. Likewise, multiple successive mismatches of two or three nucleotides spread throughout the spacer sequence also completely abolished its silencing activity (Fig. 6g).

Next, we wanted to benchmark the mismatch tolerance of Psp-Cas13b with the commonly used RfxCas13d ortholog. A bona fide RfxCas13d crRNA1 (used in Fig. 3l) efficiently silenced the mCherry transcript. We generated 14 additional RfxCas13d crRNA mutants that harbored either consecutive or nonconsecutive mismatches with the target at various spacer positions. Overall, by comparing the mismatch tolerance profiles of PspCas13b and RfxCas13d, we showed that these two enzymes exhibit distinct patterns of position-dependent mismatch sensitivity (Extended Data Fig. 8a,b). This is unsurprising given the poor sequence homology between these two Cas13 orthologs.

We further questioned whether introducing mismatches versus truncations in PspCas13b spacers at the 5′ or 3′ ends would lead to similar or distinct silencing profiles. The truncation of a motif three nucleotides or longer from the 5′ end led to a substantial loss of silencing. Conversely, the truncation of a three-nucleotide motif from the 3′ end did not reduce the silencing efficiency, whereas the excision of 6, 9, 12 and 15 nucleotides led to a gradual loss of silencing activity (Extended Data Fig. 8c,d). Interestingly, the comparison of mismatched and truncated spacers suggested that unpaired nucleotides within the spacer–target duplex may create further steric hindrances that exacerbate the loss of silencing activity.

Next, we questioned whether mismatch tolerance can be influenced by the length of the spacer. To test this hypothesis, we used a 27-nucleotide spacer sequence with a truncated 3′ end (Δ28–30) that previously exhibited full silencing activity compared to the full-length spacer. Then, we generated 16 additional mutant spacers by introducing consecutive or nonconsecutive mismatches at various positions. As expected, the substitution of the 5′ G-G-G nucleotides with a C-C-C motif led to a complete loss of silencing. The introduction of three consecutive mismatches at spacer positions 22–24 also led to near-complete loss of silencing, whereas unpaired bases at spacer positions 4–6, 7–9 and 10–12 led to substantial loss of silencing. Conversely, three consecutive mismatches at positions 13–15, 16–18 and 19–21 were fully tolerated (Extended Data Fig. 8e). Additionally, single-nucleotide mismatches at spacer positions 7 and 14 were fully tolerated, whereas a single mismatch at position 21 led to a substantial loss of silencing activity. Likewise, two, three, four and five nonconsecutive mismatches led to a substantial or complete loss of activity regardless of their positions (Extended Data Fig. 8f). Together, these data suggested that shortening the PspCas13b spacer exacerbates its mismatch intolerance.

These data revealed the targeting resolution of PspCas13b and suggested that nonconsecutive mismatches of more than four nucleotides compromises PspCas13b activity. In addition, the data suggested that endogenous targets with partial sequence homology are unlikely to be impacted by off-target silencing because of the required minimum base pairing of ~25 consecutive or nonconsecutive nucleotides. These mutagenesis data provided further evidence that highly effective crRNAs can be readily designed with minimal or no off-target effects.

**Quantitative proteomics reveals the high specificity of PspCas13b**

Permissive target recognition because of spacer–target mismatch tolerance, together with collateral degradation of neighboring RNAs,

---

**Fig. 7 | Proteomic analysis reveals high specificity and a lack of collateral activity against endogenous proteins. a**, Diagramming (created with BioRender.com) MS assays for exploring the off-target effects and collateral activities of PspCas13b in HEK 293T cells with ectopic BCR-ABL1(p190)-IRES-eGFP mRNA as a target. The cells were transfected with PspCas13b, BCR-ABL1 and targeting (T) or NT crRNA plasmids. dPspCas13b and crRNA only (no PspCas13b) were used as controls. Proteins were extracted 48 h after transfection, followed by trypsin digestion and MS analysis. **b,f,j**, eGFP reporter assays to assess BCR-ABL1 silencing obtained with PspCas13b (**b**), dPspCas13b (**f**) or crRNA only (**j**). Data points are the averaged mean fluorescence from four representative fields of view per imaged condition (*n* = 3). Errors are the s.e.m. and *P* values from an unpaired two-tailed Student's *t*-test are indicated (95% confidence interval). **c,g,k**, RT–qPCR assays measuring BCR-ABL1 knockdown obtained with PspCas13b (**c**), dPspCas13b (**g**) or crRNA only (**k**) (*n* = 3). Data are the normalized means and

errors are the s.e.m. Results were analyzed by an unpaired two-tailed Student's *t*-test and *P* values are indicated (95% confidence interval). **d,h,l**, Representative western blots examining the expression of PspCas13b and BCR-ABL1 proteins obtained with PspCas13b (**d**), dPspCas13b (**h**) or crRNA only (**l**). Right, the graphs quantify the BCR-ABL1 and PspCas13b protein levels, with each data point representing the ratio of BCR-ABL1 to β-actin or PspCas13b to β-actin normalized to the NT crRNA condition (**h,l**, *n* = 3; **d**, n = 5). Error bars represent the s.e.m. and *P* values were calculated by an unpaired two-tailed Student's *t*-test (95% confidence interval). **e,i,m**, Left, volcano plots showing the proteome of cells expressing PspCas13b (**e**), dPspCas13b (**i**) or crRNA only (**m**). Data points represent proteins, with log2FC > 1 (upregulation, blue) or log2FC < −1 (downregulation, red) and *P* < 0.05 indicating significantly differential expression. BCR-ABL1 and eGFP are labeled. Right, linear regression plots from the same experiments. Results were analyzed by an unpaired two-tailed Student's *t*-test (*n* = 5).

can cause off-target effects and undermine the specificity of various RNA-targeting CRISPR enzymes[28,29,36,44]. The comprehensive mutagenesis described above (Fig. 6) suggested that PspCas13b is highly specific and is unlikely to silence other cellular transcripts because of its stringent requisite base pairing and low mismatch tolerance. Yet, a direct assessment of the specificity of PspCas13b through a proteome-wide analysis in mammalian cells could simultaneously probe off-target effects related to both mismatch tolerance and collateral activity.

We aimed to quantitate global protein abundance when the PspCas13b nuclease domain was in an active state (through spacer–target base pairing) or inactive state (no spacer–target base pairing). We chose the BCR-ABL1-eGFP transcript as an RNA target to assess the specificity of PspCas13b in HEK 293T cells. If PspCas13b possesses high fidelity, its on-target activity would suppress BCR-ABL1-eGFP protein expression alone, without impacting the expression profile of other endogenous cellular proteins. However, if target-activated PspCas13 exhibits lower

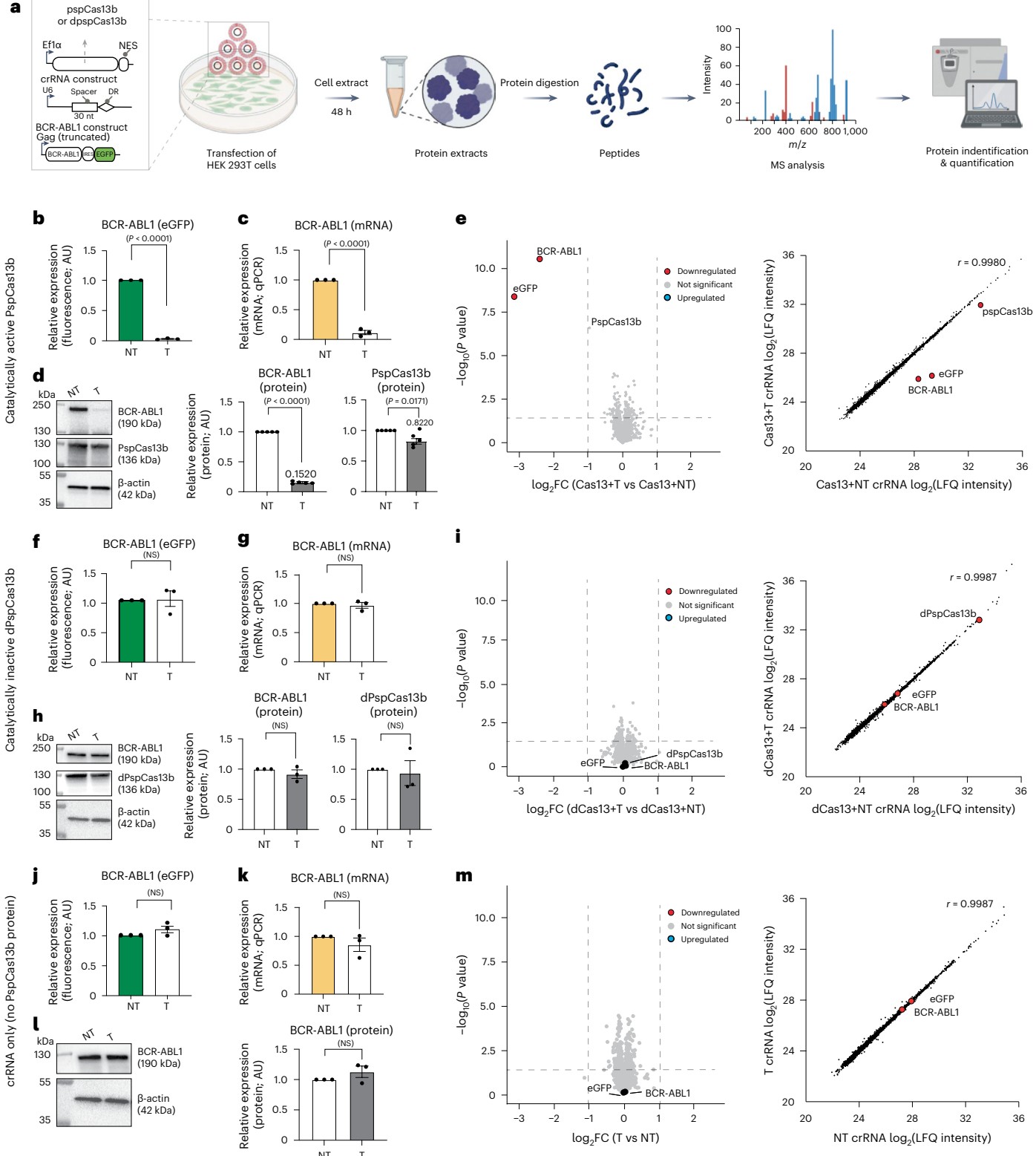

specificity through mismatch tolerance and/or collateral activities, nonspecific endogenous protein degradation would occur. To test these hypotheses, we harvested protein lysates from HEK 293T cells after 48 h of expression of BCR-ABL1-eGFP, PspCas13b and an NT or BCR-ABL1-targeting crRNA (Fig. 7a). Fluorescence, RT–qPCR and western blot analyses all showed potent cleavage of the BCR-ABL1 transcript and efficient silencing of its protein (Fig. 7b–d). In addition to BCR-ABL1, we observed a minor reduction (~18%) in the expression level of PspCas13b protein in the targeting condition, whereas the housekeeping protein β-actin remained unchanged (Fig. 7d). We then performed a comparative mass spectrometry (MS) analysis of global protein abundance in NT and targeting conditions. The data revealed that, among the 3,837 human proteins we detected, only the target BCR-ABL1 protein and its eGFP reporter (encoded in the same BCR-ABL1-IRES-eGFP transcript) were strongly silenced when PspCas13b was activated, without any noticeable perturbation of human endogenous proteins (Fig. 7e,f). Consistent with the western blot analysis, the PspCas13b expression level was only moderately reduced. Importantly, this proteomic analysis further indicated that PspCas13b is highly specific and the activation of its HEPN nuclease domains does not lead to global off-target or collateral silencing of endogenous transcripts. To confirm that the silencing of BCR-ABL1 is mediated by the activation of the nuclease domain of PspCas13b, we performed additional control experiments using dPspCas13b or a targeting crRNA alone (no PspCas13b). Fluorescence, RT–qPCR, western blot and proteomic analyses showed that dPspCas13b or a BCR-ABL1-specific crRNA alone is unable to silence the target (Fig. 7f–m).

To further investigate the specificity of PspCas13b, we benchmarked its nuclease activity against the RfxCas13d ortholog that is known to possess pronounced collateral activity[27,28,36–38]. We reprogrammed PspCas13b to silence mCherry in HEK 293T cells while coexpressing eGFP as a nontarget transcript. Fluorescence analysis demonstrated strong suppression of mCherry with no observable collateral effect on eGFP (Extended Data Fig. 9a–c). However, RfxCas13d exhibited a robust silencing of both mCherry (intended target) and eGFP (unintended nontarget) (Extended Data Fig. 9d–f). Together, these results indicated that PspCas13b exhibits superior specificity because of the absence of noticeable collateral activity.

## Discussion

Our study elucidated the key molecular principles of PspCas13b RNA silencing, which enabled the de novo design of potent crRNAs to silence various exogenous and endogenous transcripts. The results also demonstrated the high on-target specificity of PspCas13b and the lack of off-target or collateral activity in human HEK 293T cells.

### Design principles of PspCas13b for potent RNA silencing

Cas13 nucleoproteins are a promising tool for the effective and specific silencing of targeted transcripts, without the risk of permanent alterations to genomic DNA[20–22]. Several studies have investigated the design principles of widely used Cas13 orthologs such as LwaCas13a (ref. 45) and RfxCas13d (refs. 32,33,45). However, the PspCas13b ortholog belongs to a distinct subgroup of Cas13 (ref. 11). The unique crRNA structure and protein domain architecture of PspCas13b imply that it is likely to exhibit distinct targeting mechanisms governed by particular design principles that have remained largely uninvestigated. Indeed, our single-nucleotide resolution tiled crRNA screens and the comprehensive in silico analysis demonstrated that, unlike other CRISPR systems[46], PspCas13b silencing is not restricted by a defined PFS, underlining its greater design flexibility (Fig. 3).

We identified several factors that strongly predict PspCas13b activity, including a G-rich motif at the 5′ end of the spacer, which is a key determinant of crRNA potency (Figs. 3–5). Although G·C and C·G base pairing contribute the same number of hydrogen bonds for the stability of the spacer–target duplex, the superior silencing obtained with spacers containing a 5′ G-rich motif cannot be explained solely by thermodynamic stability. In fact, when we substituted ribonucleotides at the 5′ end of an ineffective spacer with mismatched G bases, we systematically improved silencing, despite introducing mismatches that are inherently unfavorable for duplex RNA stability (Fig. 4). These findings suggest that the G-rich motif at the 5′ end of the spacer may not greatly contribute to spacer–target base pairing. We speculate that the 5′ G-G sequence may be buried inside a pocket within the PspCas13b protein, potentially not directly engaging in base pairing with the target. Such recognition of RNA termini by specific protein pockets is not uncommon, as analogous interactions have been reported for other enzymes involved in eukaryotic RNAi[3,47]. This potential interaction between the 5′ G-G motif of a crRNA and protein pocket could effectively lock the enzyme in an active state, thereby enhancing its catalytic activity.

Integrating these validated discoveries into our design process enabled us to consistently generate highly potent crRNAs (Figs. 4 and 5). Therefore, we created an open-access and user-friendly online algorithm (https://cas13target.azurewebsites.net/) to assist the wider scientific community with the design of predicted potent crRNAs for PspCas13b. Overall, our findings demonstrate the potential of PspCas13b to achieve precise and efficient gene silencing, with greater design flexibility and expanded target range.

### The high specificity and lack of collateral activity of PspCas13b

The specificity of the spacer–target interaction is crucial to the ability of PspCas13b to selectively silence target RNAs in the RNA-crowded cellular environment[48], especially in the context of partial sequence homology between various RNAs. The silencing activity is sensitive to the number of consecutive or nonconsecutive mismatches at various locations of the spacer. Mismatch intolerance occurs when ~4–5 unpaired nucleotides are introduced between the spacer and the target, which creates a sharp decline in silencing activity, possibly because of duplex destabilization and/or steric hindrances that constrain conformational changes required for nuclease domain activation.

This comprehensive mutagenesis study revealed the interface between mismatch tolerance and intolerance, which provides a blueprint for crRNA design to silence various transcripts with near-single-base resolution, further expanding the targeting spectrum to single-nucleotide variants that drive numerous human diseases including cancers[49,50].

The mutagenesis study also indicated that the stringent mismatch intolerance of this nuclease is unlikely to promote off-target effects across the transcriptome. Accordingly, it is exceedingly unlikely that a targeting crRNA would encounter an endogenous transcript, other than the target and its isoforms, that would activate PspCas13b nuclease activity (Supplementary Table 1). While the likelihood of off-target effects is low, it is essential to note that the human transcriptome contains numerous paralog transcripts and splicing isoforms that share significant sequence homology. Consequently, our crRNA design algorithm (https://cas13target.azurewebsites.net/) incorporates an off-target prediction function, ensuring the identification of potent crRNAs that minimize the risk of off-target effects (Extended Data Fig. 10).

As this off-target prediction requires experimental validation, we performed proteome-wide analysis to examine the on-target, off-target and collateral activities of PspCas13 in human cells. Our comparative proteomic analyses revealed no noticeable off-target effects or widespread collateral activity against other endogenous cellular proteins in HEK 293T cells. This interesting finding suggested that (1) the PspCas13b ortholog has no significant collateral activity and/or (2) perhaps unlike the established collateral activity in vitro and in bacteria, the higher cellular organization of human cells may limit or prevent the collateral degradation of endogenous cellular RNAs and subsequent protein repression by PspCas13b. Previous studies, however, have used different orthologs of the Cas13 family, such as LwaCas13a and RfxCas13d, to investigate collateral activity in mammalian cells. These studies reported conflicting findings, which may be because of

various factors such as cell lines, target abundance and the analytical techniques used to measure collateral activity[29,30,36,51,52]. Overall, the majority of previously published studies suggested that RfxCas13b and LwaCas13a demonstrate collateral activity to some extent, whereas our study suggested that PspCas13b has no or very limited collateral activity in HEK 293T human cells (Fig. 7 and Extended Data Fig. 9).

Overall, this study provides a detailed view of the action mechanisms of PspCas13b and offers valuable guidelines for its reprogramming for efficient and specific RNA silencing.

## Online content

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

[1]Peter MacCallum Cancer Centre, Melbourne, Victoria, Australia. [2]Sir Peter MacCallum Department of Oncology, The University of Melbourne, Parkville, Victoria, Australia. [3]Department of Electrical and Electronic Engineering, The University of Melbourne, Parkville, Victoria, Australia. [4]Department of Microbiology and Immunology, The Peter Doherty Institute for Infection and Immunity, The University of Melbourne, Melbourne, Victoria, Australia. [5]Murdoch Children's Research Institute, Royal Children's Hospital, Parkville, Victoria, Australia. [6]Children's Cancer Institute, Lowy Cancer Research Centre, UNSW Sydney, Sydney, New South Wales, Australia. [7]School of Women's and Children's Health, UNSW Sydney, Sydney, New South Wales, Australia. [8]Present address: Diagnostic Genomics, Monash Health Pathology, Monash Medical Centre, Clayton, Victoria, Australia.
✉e-mail: mohamed.fareh@petermac.org

# Methods

## Design and cloning of crRNAs

The cloning of PspCas13b crRNAs was performed according to a previous publication[12]. Briefly, individual crRNAs were cloned into the pC0043-PspCas13b crRNA backbone (Addgene 103854, a gift from F. Zhang's lab), which contains a PspCas13b crRNA direct repeat sequence and two BbsI restriction sites for cloning of the spacer sequence. The crRNA backbone was digested by BbsI restriction enzymes (New England Biolabs (NEB), R3539) for 2 h at 37 °C following the manufacturer's instructions. The digested backbone was purified with NucleoSpin Gel and a PCR Cleanup Kit (Macherey-Nagel, 740609.50), aliquoted and stored at −20 °C. PspCas13b crRNA spacers were designed as single-stranded forward and reverse DNA oligos containing CACC and CAAC overhangs, respectively, ordered from Sigma or IDT. The forward and reverse DNA oligos were annealed in 1× NEBuffer 3.1 through 5-min incubation at 95 °C followed by a slow cooldown in the heating block overnight. The annealed oligos were ligated into the digested PspCas13b crRNA backbone using T4 ligase (Promega, M1801) for 3 h at room temperature.

Similarly, RfxCas13d crRNA spacers were designed as single-stranded forward and reverse DNA oligos containing AAAC and AAAA overhangs, respectively, allowing for ligation into a BsmBI-digested (NEB, R0739S) plasmid encoding an RfxCas13d direct repeat (Addgene 138150, a gift from N. Sanjana's lab).

All crRNA spacer sequences and target sequences used in this study are listed in Supplementary Tables 2 and 3. All crRNAs generated in this study were verified by Sanger sequencing (Australian Genome Research Facility). The primers used for Sanger sequencing are listed in Supplementary Table 4.

## Plasmid amplification and purification

The procedures for plasmid amplification and purification were described in a previous publication[17]. Briefly, TOP10 or Stbl3 bacteria were used for transformation. Ligated plasmids were transformed into chemically competent bacteria by heat shock at 42 °C for 45 s, followed by 2 min on ice. The transformed bacteria were incubated in 500 µl of Luria–Bertani (LB) broth containing 75 µg ml⁻¹ ampicillin (Sigma-Aldrich, A9393) for 1 h at 37 °C in a shaking incubator (200 r.p.m.). The bacteria were pelleted by centrifugation at 6,000g for 1 min at room temperature, resuspended in 100 µl of LB broth and plated onto a prewarmed 10-cm LB agar plate containing 75 µg ml⁻¹ ampicillin, which was then incubated at 37 °C overnight. On the next day, single colonies were picked and transferred into bacterial starter cultures and incubated for ~6 h for miniprep (Macherey-Nagel, NucleoSpin Plasmid Mini kit for plasmid DNA, 740588.50) or maxiprep (Macherey-Nagel, NucleoBond Xtra Maxi Plus, 740416.50) DNA purification according to the standard manufacturer's protocol.

## Cell culture

HEK 293T cell line (American Type Culture Collection, CRL-3216) was cultured in high-glucose DMEM (Thermo Fisher, 11965092) containing 10% heat-inactivated FBS (Thermo Fisher, 10100147), 100 mg ml⁻¹ penicillin–streptomycin (Thermo Fisher, 151401220) and 2 mM GlutaMAX (Thermo Fisher, A1286001). HEK 293T cells were maintained at a confluency between 20% and 80% in incubators at 37 °C with 10% $CO_2$.

## In vitro transcription of crRNA

The DNA template for in vitro transcription of crRNA was ordered from IDT as complementary single-stranded DNA oligonucleotides containing a TAATACGACTCACTATA T7 promoter, followed by the spacer and direct repeat sequences. The forward and reverse DNA oligos were added to 1× NEBuffer 3.1. The two strands were first denatured through 5-min incubation at 95 °C followed by a gradual cooldown in the heating block overnight for annealing. The annealed DNA was used as a template for in vitro transcription using the mMESSAGE

mMACHINE T7 Transcription Kit (Thermo Fisher, AM1344) according to the manufacturer's instructions. We performed a 6-h incubation to maximize the yield arising from small DNA templates, followed by DNase treatment. IVT crRNAs were purified with Monarch RNA Cleanup kits (NEB, T2040L) to remove enzymes and unincorporated nucleotides according to the manufacturer's instructions. The purified crRNAs were quantified with NanoDrop 2000/2000c Spectrophotometers (Thermo Fisher, ND-2000) and the crRNA length and integrity were evaluated using 15% urea-PAGE and an Agilent 2200 Tapestation (Agilent, G2964AA) with RNA ScreenTape (Agilent, 5067-5576) according to the manufacturer's instructions. Of note, we observed a 3–5-fold greater yield of IVT crRNAs incorporating a 5′ G-G sequence compared to their counterparts without a 5′ G-G sequence. After RNA purification, we measured crRNA concentrations and corrected for any difference in the yield to deliver the same amount of crRNA to the cells. All PspCas13b crRNA spacer sequences are listed in Supplementary Table 2.

## RNA silencing assays using plasmid transfections

All transfection experiments were performed using an optimized Lipofectamine 3000 transfection protocol (Thermo Fisher, L3000015). For RNA silencing in HEK 293T, cells were plated at approximately 30,000 cells per 100 µl per well in tissue culture-treated flat-bottom 96-well plates (Corning) 18 h before transfection. For each well (unless stated otherwise), 100 ng of DNA plasmids (22 ng of PspCas13b-NES-3xFLAG-T2A-BFP (Addgene 173029), pC0046-EF1a-PspCas13b-NES-HIV (Addgene 103862), pC0049-EF1a-dPspCas13b-NES-HIV (Addgene 103865) or pLentiRNACRISPR-007-RfxCas13d (Addgene 138149)), 22 ng of crRNA plasmid and 56 ng of the target gene plasmid were mixed with 0.2 µl of P3000 reagent in serum-free Opti-MEM (Thermo Fisher, 31985070) to a total of 5 µl (mix 1). Separately, 4.7 µl of Opti-MEM was mixed with 0.3 µl of Lipofectamine 3000 (mix 2). Mixes 1 and 2 were added together and incubated for 20 min at room temperature; then, 10 µl of transfection mixture was added to each well. Supplementary Table 5 summarizes the transfection conditions used in the 96-well, 24-well and 12-well plates. After transfection, cells were incubated at 37 C° in 10% $CO_2$ and the transfection efficacy was monitored 24–72 h after transfection by fluorescence microscopy. For the RfxCas13d silencing assays, the culture medium was supplemented with a final concentration of 1 µg ml⁻¹ doxycycline at the moment of transfection.

## RNA silencing assays using IVT and chemically synthesized crRNAs

Similar transfection conditions in HEK 293T were used except we substituted crRNA plasmid with 200 ng of IVT crRNAs or 400 ng of chemically synthesized crRNAs (IDT). The choice of 200 ng of IVT crRNA and 400 ng of chemically synthesized crRNAs was based on the optimization assays where 200 ng of IVT crRNA and 400 ng of chemically synthesized crRNAs achieved good silencing. All PspCas13b crRNA sequences are listed in Supplementary Table 2.

## Fluorescence microscopy analysis

For RNA silencing experiments, the fluorescence intensity was monitored using the EVOS M5000 FL Cell Imaging System (Thermo Fisher). Images were taken at 48 h after transfection and the fluorescence intensity of each image was quantified using a lab-written macro in ImageJ software. Briefly, all images obtained from a single experiment were simultaneously processed using a batch-mode macro. First, images were converted to 8-bit, threshold-adjusted and converted to black and white using the 'Convert to Mask' function and the fluorescence intensity per pixel was measured using the 'Analyze Particles' function. Each single mean fluorescence intensity was obtained from four different fields of view for each crRNA and subsequently normalized to the NT control crRNA. A twofold or higher reduction in fluorescence intensity was considered biologically relevant.

## PspCas13b protein purification

pC0068 PspCas13 (B12) His6-TwinStrep-SUMO-BsaI (Addgene 115219) was transformed into BL21(DE3) competent *Escherichia coli* cells (Thermo Fisher, C600003). Colonies were picked and grown overnight in Terrific Broth supplemented with ampicillin. Starter cultures were used to seed 1.6 l of medium and the culture was grown with shaking at 37 °C until the optical density at 600 nm reached 0.5. Then, cultures were cooled in the cold room to reach 18 °C and induced with IPTG (Sigma, I6758) to a final concentration of 0.5 mM. Induced cultures were grown for 16 h with shaking at 18 °C. After growth, cells were collected by centrifugation at 5,000*g* for 15 min at 4 °C and the cell pellet was frozen on dry ice and stored at −80 °C for later purification.

For protein purification, the frozen cell pellet was thawed and resuspended in lysis buffer (20 mM Tris-HCl, 400 mM NaCl, 1% Triton X-100 and 10 mM β-mercaptoethanol; pH 8.0) supplemented with cOmplete, Mini, EDTA-free protease inhibitor cocktail tablets (Roche, 11836153001). After resuspension, the solution was sonicated on ice with a Sonics Vibra-Cell VCX130 Ultrasonic Processor (Sonics and Materials) for 5 min with a 50% amplitude, pulse on for 10 s with a cooling period of 10 s between each pulse. The clarified supernatant was supplemented with 2 ml of His60 Ni Superflow Resin (Takara Bio, 635660) and incubated with gentle shaking for 1 h at 4 °C. The resin was washed with 20 and 30 mM imidazole to prevent nonspecific proteins from binding. The lysate and Ni-NTA resin mixture was loaded onto a TALON 2-ml disposable gravity column (Takara Bio, 635606) and eluted with elution buffer (20 mM Tris-HCl pH 8.0, 400 NaCl, 10 mM β-mercaptoethanol and 200 mM imidazole). Protein was concentrated using an Amicon 100-kDa centrifugal filter (Sigma, UFC810024) and buffer-exchanged into storage buffer (600 mM NaCl, 50 mM Tris-HCl pH 7.5, 5% glycerol and 2 mM DTT), quantified by Bolt Bis-Tris Plus 4–12% gels (Thermo Fisher, NW04120BOX) in 1× MES SDS running buffer (Thermo Fisher, B0002) with a PageRuler Plus Prestained Protein Ladder (Thermo Fisher, 26619), aliquoted and frozen at −80 °C for storage.

## In vitro nuclease assays

Unless otherwise indicated, in vitro nuclease assays were performed with 200 nM purified PspCas13b, 25 nM 6-FAM-labeled single-stranded RNA (ssRNA) targets (IDT) and 200 nM chemically synthesized crRNAs (IDT) in nuclease assay buffer (40 mM Tris-HCl, 60 mM NaCl and 6 mM MgCl$_2$; pH 7.5) supplemented with 20 U of RNase inhibitor (Promega, N2111). Reactions were incubated for 2–4 h at 37 °C and then quenched with the addition of 0.8 U of proteinase K (NEB, P8107S) for 15 min at 25 °C. The reactions were mixed with equal parts of RNA loading dye (NEB, B0363S) and denatured at 95 °C for 5 min and then cooled on ice for 2 min. Samples were analyzed by denaturing gel electrophoresis on 15% PAGE TBE-Urea (Thermo Fisher, EC6885BOX) run at 45 °C. Gels were imaged using a ChemiDoc Imaging System (Bio-Rad). All PspCas13b crRNA sequences and target ssRNA sequences used in this study are listed in Supplementary Tables 2 and 3.

## EMSA

For the EMSA, binding experiments were performed with 10 nM crRNAs incubated with various concentrations of PspCas13b ranging from 6.25 to 800 nM. Binding assays were performed using a binding buffer similar to the nuclease assay buffer lacking MgCl$_2$ and supplemented with 10 mM EDTA, 5% glycerol and 5 µg ml$^{-1}$ heparin to avoid nonspecific interactions. Reactions were then incubated at 37 °C for 30 min and then resolved on 6% PAGE TBE gels (Thermo Fisher, EC6265BOX) at 4 °C using 0.5× TBE buffer. Gels were imaged using a ChemiDoc Imaging System (Bio-Rad).

## Cell flow cytometry

For B2M surface marker staining, up to $1 \times 10^6$ cells were incubated in 50 µl of PBS with 2% FBS (v/v) containing B2M antibody for 30 min on ice in the dark. The cells were then washed twice with 200 µl of PBS with

2% FBS (v/v) before being resuspended in 200 µl of PBS with 2% FBS (v/v) for flow cytometry analysis. Flow cytometry analysis was performed using the FACSymphony Cell Analyzer A5 or A3 (BD Biosciences). All flow cytometry profiles were analyzed using FlowJo version 10 software (Tree Star). Supplementary Table 6 lists the antibodies that were used for cell flow cytometry.

## Western blot

Cells were washed three times with ice-cold PBS and lysed on ice in radioimmunoprecipitation assay (RIPA) lysis buffer (50 mM Tris pH 8.0 (Sigma-Aldrich, T1530), 150 mM NaCl, 1% NP-40 (Sigma-Aldrich, I18896), 0.1% SDS and 0.5% sodium deoxycholate (Sigma-Aldrich, D6750)) containing protease inhibitor cocktail (Roche, 11836153001) and phosphatase inhibitor cocktail (Roche, 4906845001). Samples were incubated for 30 min at 4 °C with rotation (25 r.p.m.) and centrifuged at 16,000*g* for 10 min at 4 °C. The supernatant was transferred to a new tube. Protein concentrations were quantified using the Pierce BCA Protein Assay Kit (Thermo Fisher, 23225) according to the manufacturer's instructions. A total of 10 µg of protein diluted in 1× Bolt LDS sample buffer (Thermo Fisher, B007) and 1× Bolt sample reducing agent (Thermo Fisher, B009) was denatured at 95 °C for 5 min. Samples were resolved by Bolt Bis-Tris Plus 4–12% gels (Thermo Fisher, NW04120BOX) in 1× MES SDS running buffer (Thermo Fisher, B0002) and transferred to 0.45 µM PVDF membranes (Thermo Fisher, 88518) by a Trans-Blot Semi-Dry electrophoretic transfer cell (Bio-Rad) at 20 V for 30 min. Alternatively, samples were resolved by 4–15% Criterion TGX Precast Midi Protein gels (Bio-Rad, 5671084) in 1× Tris-glycine SDS running buffer (Bio-Rad, 1610732) and transferred to 0.20 µM nitrocellulose membranes (Bio-Rad, 1704159) by a Trans-Blot Turbo Transfer System (Bio-Rad) with a high-molecular-weight protocol. Membranes were incubated in blocking buffer 5% (w/v) BSA (Sigma-Aldrich, A3059) in TBS with 0.15% Tween 20 (Sigma-Aldrich, P1379) for 1 h at room temperature and probed overnight with primary antibodies at 4 °C. Blots were washed three times in TBS with 0.15% Tween 20, followed by incubation with fluorophore-conjugated or HRP-conjugated secondary antibodies for 1 h at room temperature. Membranes were washed in TBS with 0.15% Tween 20 three times and fluorescence or chemiluminescence was detected using the Odyssey CLx Imager 9140 (Li-cor), iBright CL1500 Imaging System (Thermo Fisher) or ChemiDoc Imaging System (Bio-Rad). The antibodies used for western blots are listed in Supplementary Table 6.

## RNA extraction, complementary DNA (cDNA) synthesis and RT–qPCR

For RT–qPCR, total RNA was isolated by standard Trizol–chloroform extraction according to the manufacturer's instructions (Thermo Fisher, 15596026), followed by Dnase treatment with RQ1 RNase-Free DNase according to the manufacturer's instructions (Promega, M6101). Then, 1 µg of total RNA was converted to cDNA using the high-capacity cDNA reverse transcription kit (Thermo Fisher, 4368814) following the manufacturer's instructions. RT–qPCR was performed in a technical duplicate using the StepOne Real-Time PCR system (Thermo Fisher) and PowerUp SYBR Green Master Mix (Thermo Fisher, A25742). The reaction contained 0.2 µl of cDNA, 0.6 µM forward primer and 0.6 µM reverse primer.

*GAPDH* and *HSP90* were used as housekeeping genes for the mCherry and BCR-ABL1 RT–qPCR assays, while 5S ribosomal RNA (rRNA) and *HSP90* were chosen as housekeeping genes for crRNA RT–qPCR assays. The $2^{-\Delta\Delta Ct}$ method was used to normalize the expression of a transcript of interest. Different housekeeping genes gave consistent results. Therefore, results normalized to *GAPDH* and 5S rRNA are shown in the figures. Primers for RT–qPCR are detailed in Supplementary Table 4. It is crucial to highlight that the most reliable assessment of PspCas13b silencing activity is achievable by using RT–qPCR forward and reverse primers positioned on both sides of the crRNA-binding site on the mRNA target.

## Sample processing for MS analysis

First, 48 h after transfection, cells were washed three times with ice-cold PBS and lysed on ice in RIPA lysis buffer (50 mM Tris pH 8.0 (Sigma-Aldrich, T1530), 150 mM NaCl, 1% NP-40 (Sigma-Aldrich, I18896), 0.1% SDS and 0.5% sodium deoxycholate (Sigma-Aldrich, D6750)) containing protease inhibitor cocktail (Roche, 04693159001) and phosphatase inhibitor cocktail (Roche, 4906845001). Then, 10 mM TCEP (Sigma-Aldrich, C4706) was added to 50 μg of protein lysates and incubated at 65 °C for 20 min for disulfide bond reduction. Iodoacetamide (Sigma-Aldrich, A3221) was added to a final concentration of 55 mM to alkylate proteins and samples were incubated at 37 °C for 45 min in the dark. Samples were acidified by the addition of a 2.5% final concentration of phosphoric acid. Then, S-Trap binding buffer (90% methanol and 100 mM TEAB (Sigma-Aldrich, T7408); pH 7.1) was added to the acidified lysate and samples were passed through an S-trap column (PROTIFI, K02MICRO10). The bound samples were washed three times with S-Trap binding buffer. The bound proteins were then digested overnight at 37 °C by trypsin (Thermo Fisher, 90058) in 50 mM TEAB at a 1:10 ratio. The peptides were eluted with 40 μl of 50 mM TEAB, followed by 40 μl of 0.2% aqueous formic acid and finally 35 μl of 50% acetonitrile containing 0.2% formic acid. Samples were dried in a SpeedVac concentrator (Eppendorf Concentrator Plus), resuspended in 20 μl of 2% acetonitrile and 0.1% trifluoroacetic acid, sonicated for 10 min at room temperature, centrifuged at 16,000$g$ for 10 min and transferred into high-performance liquid chromatography (HPLC) vials for analysis.

## MS proteomics analysis

Samples were analyzed by LC–MS/MS using an Orbitrap Exploris 480 (Thermo Scientific) fitted with a nanoflow reversed-phase HPLC (Ultimate 3000 RSLC, Dionex). The nano-LC system was equipped with an Acclaim Pepmap nanotrap column (Dionex C18, 100 Å, 75 μm × 2 cm) and an Acclaim Pepmap RSLC analytical column (Dionex C18, 100 Å, 75 μm × 50 cm). Typically, for each LC–MS/MS run, 1 μl of the peptide mix was loaded onto the enrichment (trap) column at an isocratic flow of 5 μl min$^{-1}$ of 2% acetonitrile containing 0.05% trifluoroacetic acid for 6 min before the analytical column was switched in line. The buffers used for LC were 0.1% (v/v) formic acid in water (solvent A) and 100% acetonitrile with 0.1% (v/v) formic acid (solvent B). The gradient used was 3–23% B for 89 min, 23–40% B for 10 min, 40–80% B for 5 min and 80% B for the final 5 min before equilibration for 10 min at 2% B before subsequent analysis. All spectra were acquired in positive mode with full-scan MS spectra scanning from $m/z$ 300–1,600 at 120,000 resolution with an automatic gain control target of $3 \times 10^6$ with a maximum accumulation time of 25 ms. The peptide ions with a charge state ≥ 2–6 were isolated with an isolation window of 1.2 $m/z$ and fragmented with a normalized collision energy of 30 at 15,000 resolution.

All generated files were analyzed with MaxQuant (version 2.2.0.0) and its implemented Andromeda search engine to obtain protein identifications and their label-free quantitation (LFQ) intensities. Database searching was performed with the following parameters: cysteine carbamidomethyl as a fixed modification, oxidation and acetyl as variable modifications with up to two missed cleavages permitted, main mass tolerance of 4.5 ppm, 1% protein false discovery rate (FDR) for protein and peptide identification and a minimum of two peptides for pairwise comparison in each protein for LFQ. The raw data files were searched against the modified human reference proteomes (UP000005640) supplemented with the amino acid sequence of the exogenous proteins eGFP, BCR-ABL1 p190 and PspCas13b.

The MaxQuant result output was further processed with Perseus (version 2.0.7.0)[53], a module from the MaxQuant suite. After removing reversed and known contaminant proteins, the LFQ values were log$_2$-transformed and the reproducibility across the biological replicates was evaluated by a Pearson's correlation analysis. The replicates were grouped accordingly and all proteins were removed that

had fewer than four 'valid values' in each group. The missing values were replaced by imputation and differential expression analysis between targeting and NT samples was performed using an unpaired two-tailed Student's $t$-test. Proteins with a log$_2$FC (fold change) > 1 were considered as upregulated while proteins with a log$_2$FC < −1 were considered downregulated. An adjusted $P$ value (with 1% FDR) < 0.05 was considered statistically significant. Proteins of interest or proteins that were significantly upregulated or downregulated between targeting and NT crRNAs were highlighted in the volcano and linear regression plots.

## Prediction of RNA secondary structure, RNA minimum free energy (MFE) and RNA–RNA hybridization energy

RNAfold was used to predict the MFE of the crRNA spacer, crRNA (direct repeat and spacer) and the 70-nucleotide target region in the target RNA (20 nucleotides upstream or downstream of the 30-nucleotide spacer-binding region). RNAfold was also used to explore the secondary structure of crRNAs and the target regions in the target RNAs. RNAplex[54] and intaRNA[55] were used to predict the hybridization energy and interaction energy between the crRNA spacer and target RNA, respectively.

## PspCas13b crRNA design tool

The design of the in silico prediction tool was based on design principles learned from the experimental data presented in this article. The R (version 4.2.2)[56] programming language was used for coding and the R Shiny framework (version 1.7.2) was used to develop the software application. The software program was developed by leveraging the Bioconductor (version 2.54.0)[57] and tidyverse[58] libraries and deployed as a web application using the Microsoft Azure platform. Briefly, the software takes an RNA or DNA sequence and generates all single-nucleotide tiled spacers using the input sequence. The program then removes all spacer sequences that possess more than three consecutive T bases that are predicted to act as a transcription termination signal and could yield premature crRNAs. The algorithm scores the remaining spacer sequences on the basis of their nucleotide composition and position. In this study, spacers with a G nucleotide at the first or second position received a maximum score of +60 each. In contrast, a C nucleotide at spacer position 1, 2, 3 or 4 received a penalty score of −60, −60, −50 or −40, respectively. Additionally, C bases at positions 11, 12, 15, 16 and 17 received a −5 score each. All other nucleotides or spacer positions that did not show any enrichment in the potent and ineffective crRNA cohorts received a score of 0. The algorithm then calculated the cumulative score for each spacer and ranked them accordingly. As a result, the top spacers with high scores were enriched with G bases at the first and second positions and depleted from C bases at positions 1, 2, 3, 4, 11, 12, 15, 16 and 17 and were predicted to yield potent silencing. Conversely, the lowest-scoring spacers at the bottom of the list were enriched with C bases at positions 1, 2, 3, 4, 11, 12, 15, 16 and 17 and were predicted to yield ineffective silencing. The prediction accuracy of the algorithm was supported by in silico analysis and functional validation data (Figs. 3–5 and Extended Data Figs. 3–6). In addition to identifying potent crRNAs for PspCas13b, the web application also assessed the potential off-target effects of the top ten predicted potent spacer sequences as a function of their sequence complementarity with various RNA transcripts in the human transcriptome. The web application integrates the National Center for Biotechnology Information (NCBI) basic local alignment search tool for this purpose and reports the on-target and off-target interactions within the human transcriptome (GRCh38.p14; GCF_000001405.40-RS_2023_10 (October 2, 2023)). The webpage displays the percentage of matches and number of nucleotide mismatches with other RNA molecules that possess sequence complementarity with the selected spacer sequence. The software categorizes off-target effects as nonexistent when the number of

mismatches is greater than 15. Building on mutagenesis studies, we predict that crRNAs with partial sequence complementarity, involving five-nucleotide mismatches or longer, are likely to lose their silencing activity. Consequently, any off-target transcript displayed on the prediction webpage with five or more mismatches is considered unlikely to be silenced. In cases where potential off-target effects are identified, the output provides a link to the NCBI records of these human transcripts. To expedite the processing of this web application, we recommend using DNA-coding or RNA-coding sequences with a length shorter than 1,000 nucleotides as input.

This PspCas13b crRNA design tool is open-source and available to the wider scientific community at https://cas13target.azurewebsites.net (Extended Data Fig. 10).

## Data analysis

Data analyses and visualizations (graphs) were performed in GraphPad Prism software version 9, unless stated otherwise. In some figures, identical fluorescence data points from an NT control are displayed in multiple panels to facilitate comparison with the targeting condition. The figures displaying identical NT data points are listed in the Source Data for reference. Specific statistical tests and numbers of independent biological replicates are mentioned in the respective figure legends. The silencing efficiency of various crRNAs was analyzed using an unpaired two-tailed Student's $t$-test or one-way analysis of variance (ANOVA) followed by Dunnett's multiple comparison test where we compared every mean to a control mean as indicated in the figures (95% confidence interval). The $P$ values are indicated in the figures. $P < 0.05$ was considered statistically significant. Pearson's correlation coefficient was used to analyze the correlation between the crRNA silencing efficiency and potential parameters including crRNA MFE, target MFE, crRNA spacer MFE, crRNA–target RNA hybridization or interaction energy, crRNA spacer G+C content and A, U, G or C content. The R package 'ggseqlogo' was used to assess nucleotide preference in the crRNA spacer and PFSs[59]. Delta probability graphs of spacer nucleotides were generated with Matplotlib[60].

## Reporting summary

Further information on research design is available in the Nature Portfolio Reporting Summary linked to this article.

## Data availability

All data are available in the main text and supplementary materials. Source data are available on figshare (https://doi.org/10.6084/m9.figshare.25058588)[61]. All key plasmids constructed in this study, their sequences and maps were deposited to Addgene upon publication. Reference human transcriptome GRCh38.p14 (Annotation Name: GCF_000001405.40-RS_2023_10 (October 2, 2023)) and proteome UP000005640 were used in this study. The MS proteomics data were deposited to the ProteomeXchange Consortium in the PRIDE partner repository with the dataset identifier PXD051089. Source data are provided with this paper.

## Code availability

The bioinformatic codes for the design of predicted potent crRNA are available at https://github.com/faraz107/cas13target. The crRNA design webserver described here is available at https://cas13target.azurewebsites.net/.

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

## Acknowledgements

We thank all lab members from the Trapani, Ekert, Voskoboinik and Fareh labs for facilitating experiments and discussions. We thank the Bio21 MS and proteomics facility for their help with MS experiments. This work was supported by a Cancer Council Victoria Ventures grant (ID 829606) to M.F., P.G.E. and J.A.T., by a Peter MacCallum Cancer Center strategic plan funded the Peter MacCallum Foundation in partnership with the Children's Cancer Institute Australia to M.F., P.G.E. and J.A.T., by a Peter MacCallum Foundation grant to M.F. (ID 2119), by an mAP mRNA Victoria grant to M.F. (RCH0153742) and by the National Health and Medical Research Council of Australia through a program grant to J.A.T. S.F.A. and M.R.M. were supported by the Australian Research Council (ARC) through a Discovery Project (DP 230102850). M.R.M. was the recipient of an ARC Future Fellowship (project number FT200100928). The funders had no role in the study design, data collection and analysis, decision to publish or preparation of the manuscript.

## Author contributions

M.F. conceptualized the study. M.F., J.A.T and P.G.E. supervised the study. W.H. and M.F. designed the experiments. W.H. and M.F. performed the experiments and analyzed the data. G.J.S. optimized the IVT reactions. H.C. helped with the B2M and MS experiments. A.K., S.Q. and W.H. performed the computational analysis with input from M.F. A.K. and D.K.G.M. created the initial crRNA design code with input from W.H. and M.F. S.F.A. and M.R.M. implemented the crRNA design code with off-target prediction and created the webpage that hosts this algorithm. All authors discussed the project and the data. W.H. and M.F. generated graphs and figures and wrote the manuscript. W.H., J.A.T., P.G.E., J.M.L.C., M.H. and M.F. revised and edited the manuscript. All authors read, commented on, edited and approved the manuscript.

## Competing interests

Some findings in this study are subject to a provisional patent deposited by the Peter MacCallum Cancer Center. The authors declare no other competing interests.

## Additional information

**Extended data** is available for this paper at https://doi.org/10.1038/s41594-024-01336-0.

**Correspondence and requests for materials** should be addressed to Mohamed Fareh.

Peer reviewer reports are available. Primary Handling Editor: Dimitris Typas, in collaboration with the *Nature Structural and Molecular Biology* team.

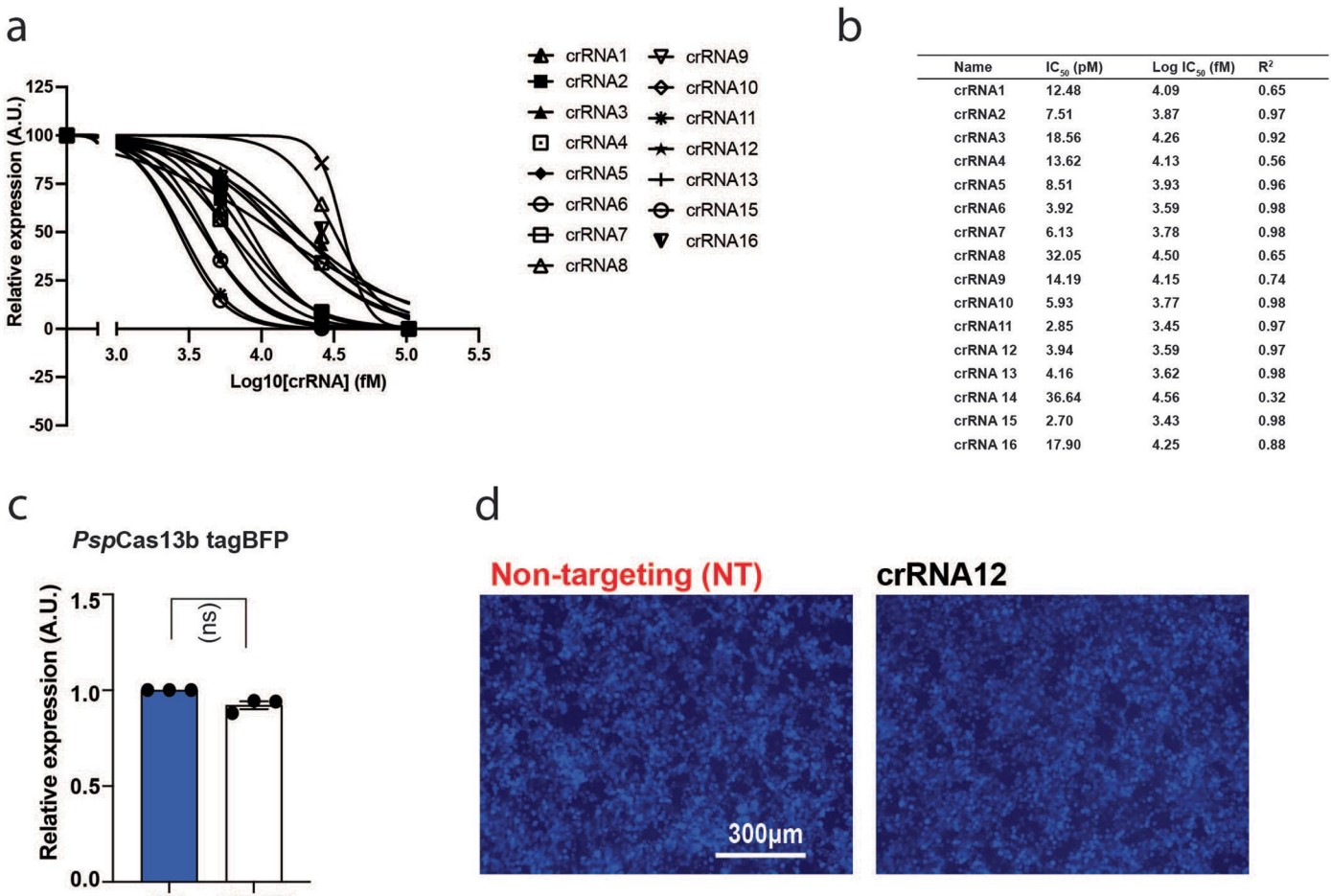

**Extended Data Fig. 1 | The silencing efficiency of various crRNAs is highly variable. (a)** Dose-response fitting of mCherry silencing data with non-targeting crRNA (NT) and 16 targeting crRNAs in Fig. 1d. **(b)** The relative IC$_{50}$ values for 16 crRNAs targeting mCherry transcripts obtained from fitting in Fig. 1d. **(c)** PspCas13b expression in HEK 293 T cells expressing a targeting crRNA12 versus a non-targeting (NT) control crRNA. Data points in the graph are averages of normalized mean fluorescence from 4 representative fields of view imaged in $N = 3$. The data are represented in arbitrary units (A.U.). Errors are SEM and p-values of one-way Anova test are indicated (95% confidence interval).

**(d)** Representative fluorescence microscopy images show the expression of PspCas13b in HEK 293 T cells expressing a targeting crRNA12 versus a non-targeting (NT) control crRNA; $N = 3$. $N$ is the number of independent biological experiments. Source data are provided as a Source data file. Note: Calculating IC50 values for plasmids is not a common practice, given the challenges in quantifying the number of crRNA molecules being produced from a single plasmid in the cell. Thus, we call this value that estimates crRNA potency "IC50" a "relative IC50". The purpose of this 'relative IC50' is to compare the silencing activity of each crRNA in a quantitative manner.

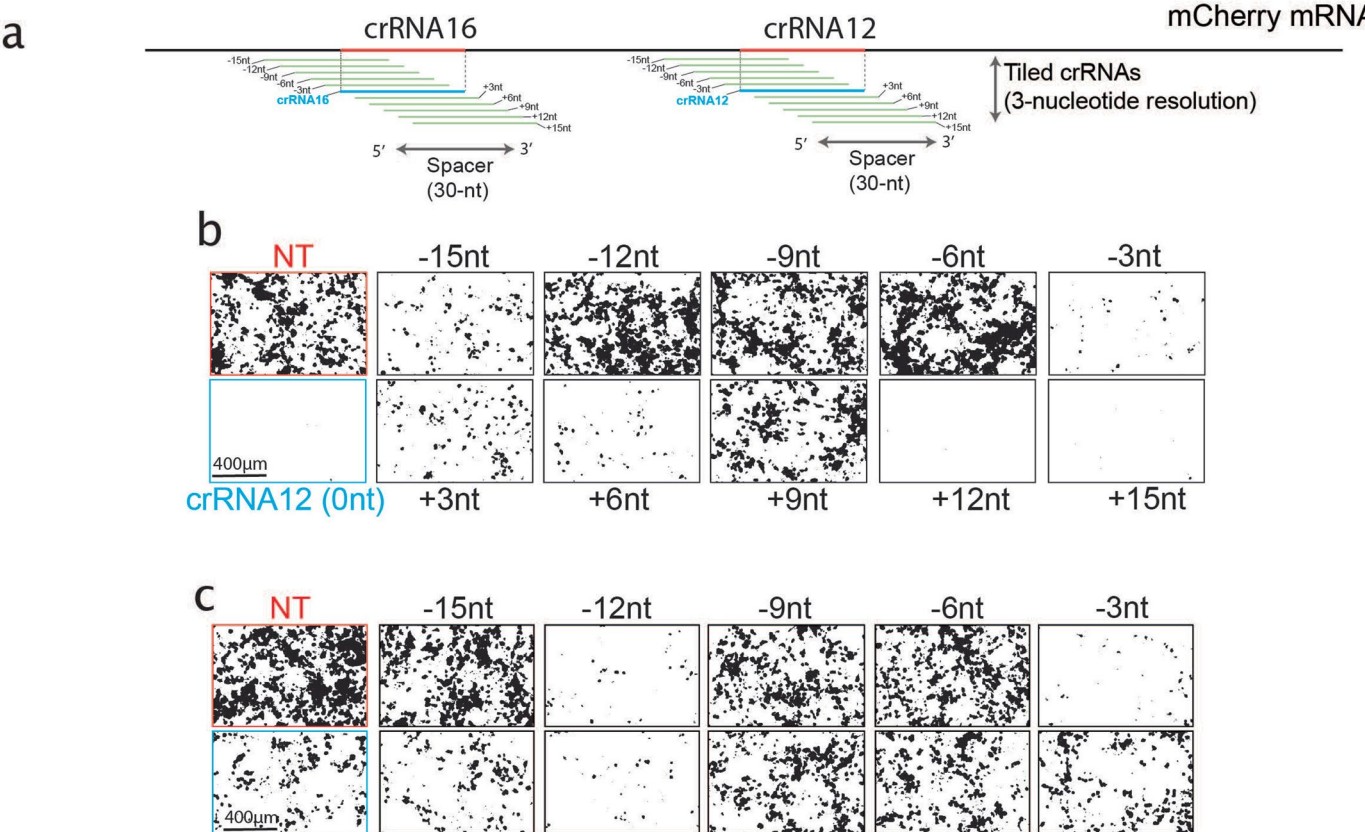

**Extended Data Fig. 2 | High variations in the silencing efficiency of 3-nucleotide tiled crRNAs targeting mCherry transcripts. (a)** The schematic illustrates mCherry RNA regions covered by 3-nucleotide increments tiled crRNAs. **(b-c)** Representative fluorescence microscopy images show the silencing of mCherry transcripts with tiled crRNAs targeting the region surrounding crRNA12 **(b)** and crRNA 16 **(c)**. Images are processed for quantification using ImageJ. Similar results were obtained in $N = 4$. Unprocessed representative images are provided in the Source Data file. $N$ is the number of independent biological experiments.

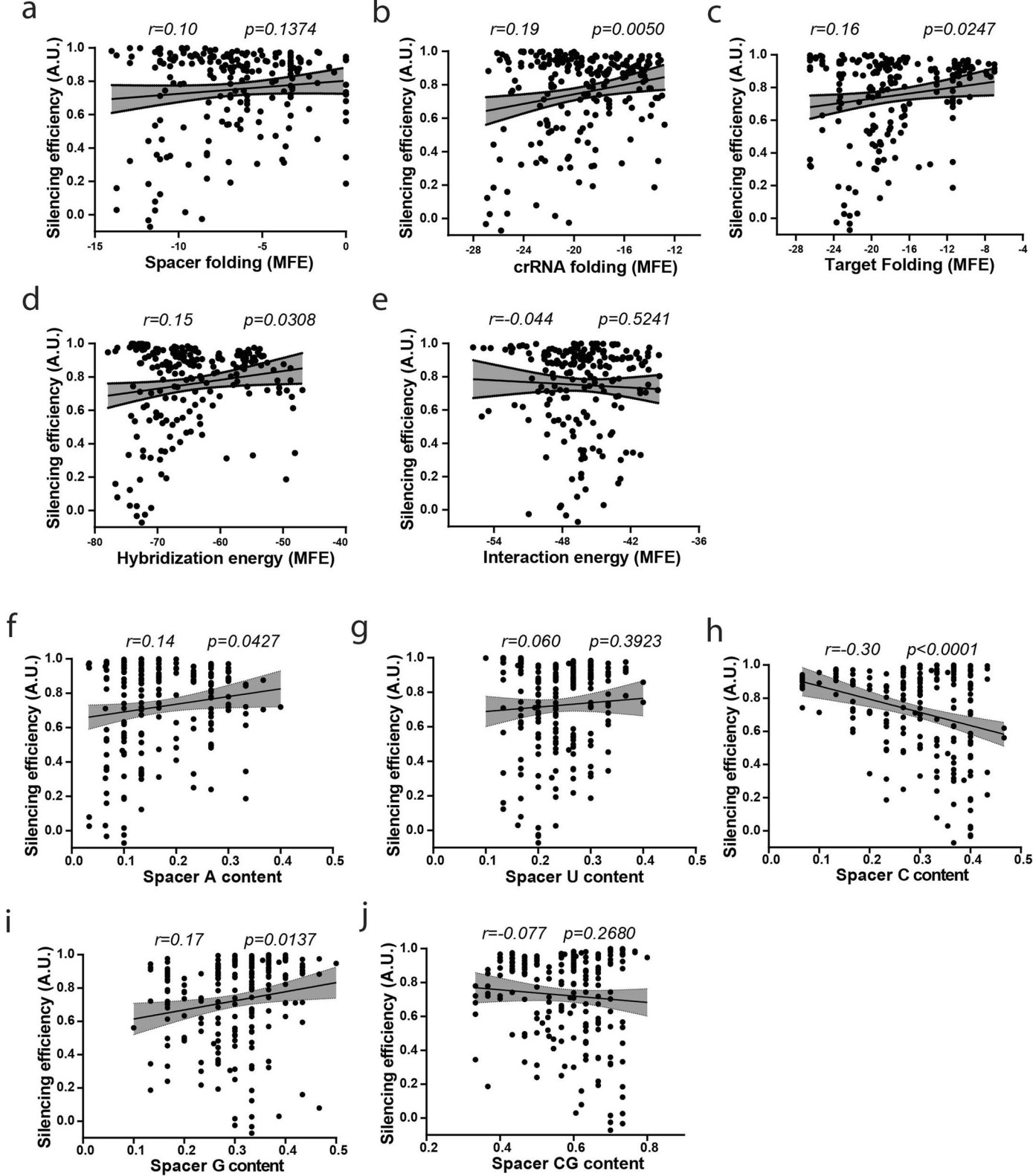

**Extended Data Fig. 3, | Pearson correlation analysis of potential determinants of *Psp*Cas13b silencing efficacy.** Pearson correlation analysis between **(a)** crRNAs silencing efficiency and spacer folding, **(b)** entire crRNA folding (spacer and direct repeat), **(c)** target sequence folding, **(d)** spacer-target hybridization, **(e)** and spacer-target interaction. Data points in the graph are values of the silencing efficiency (normalized means of independent biological replicates) of individual crRNAs and their predicted folding (MFE) or hybridization/interaction energy. Pearson correlation analysis between spacer silencing efficiency and the content of spacer in **(f)** A bases, **(g)** U bases, **(h)** C bases, **(i)** G bases, and **(j)** CG bases. Data points in the graph show the silencing efficiency (normalized means of independent biological replicates) and base content of individual spacer sequences. *r* (correlation coefficient) and *p*-value (95% confidence interval) are indicated in each graph. Results are analysed by unpaired two-tailed Student's t-test. Error bars are SEM. Source data are provided as a Source data file.

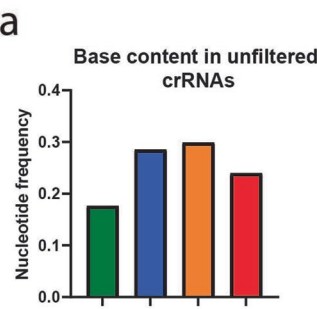

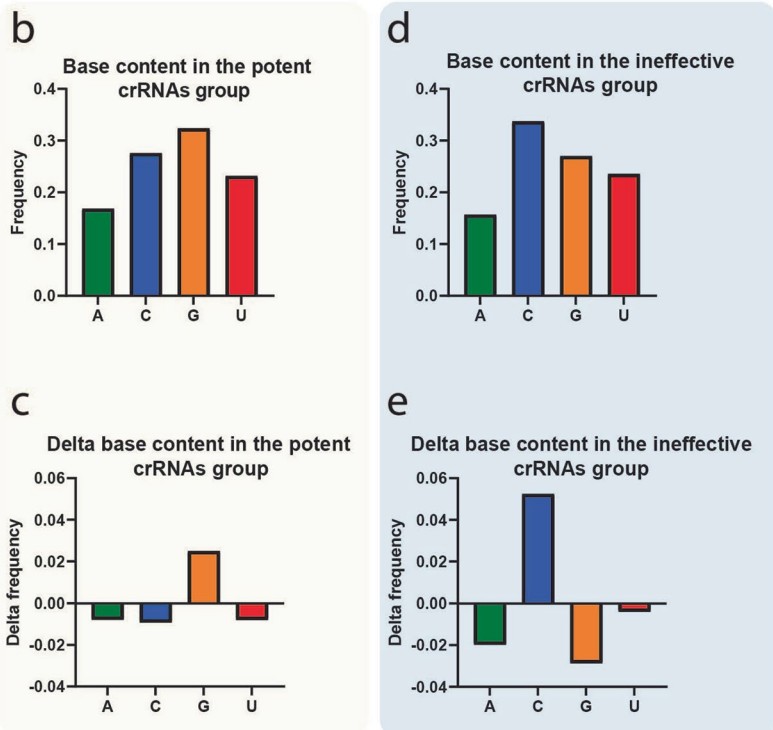

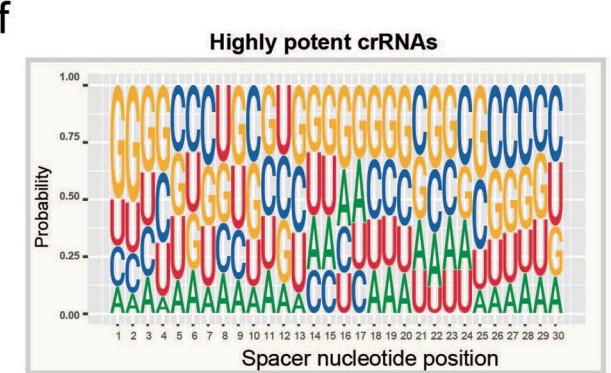

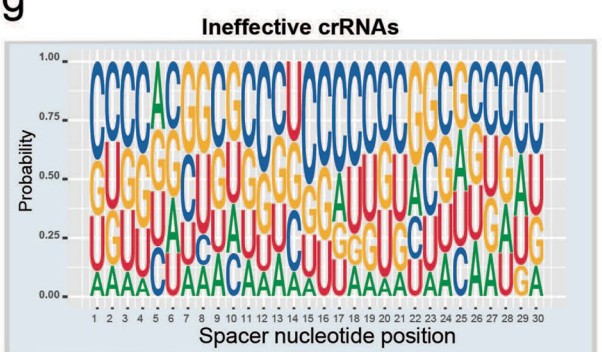

**Extended Data Fig. 4 | Correlation analysis of PspCas13b potency and spacer nucleotide composition. (a)** The frequency of A, C, G, and U bases in unfiltered crRNA spacer sequences. The frequency **(b)** and delta frequency **(c)** of each base in the spacer sequences of filtered potent crRNAs (achieved >90% silencing efficiency). The frequency **(d)** and delta frequency **(e)** of each base in the spacer sequences of filtered ineffective crRNAs (achieved <50% silencing efficiency). **(f-g)** Position Weight Matrices (PWMs) depicting the positional nucleotide probabilities of either the highly potent **(f)** or the ineffective **(g)** crRNA spacer sequences. Source data are provided as a Source data file.

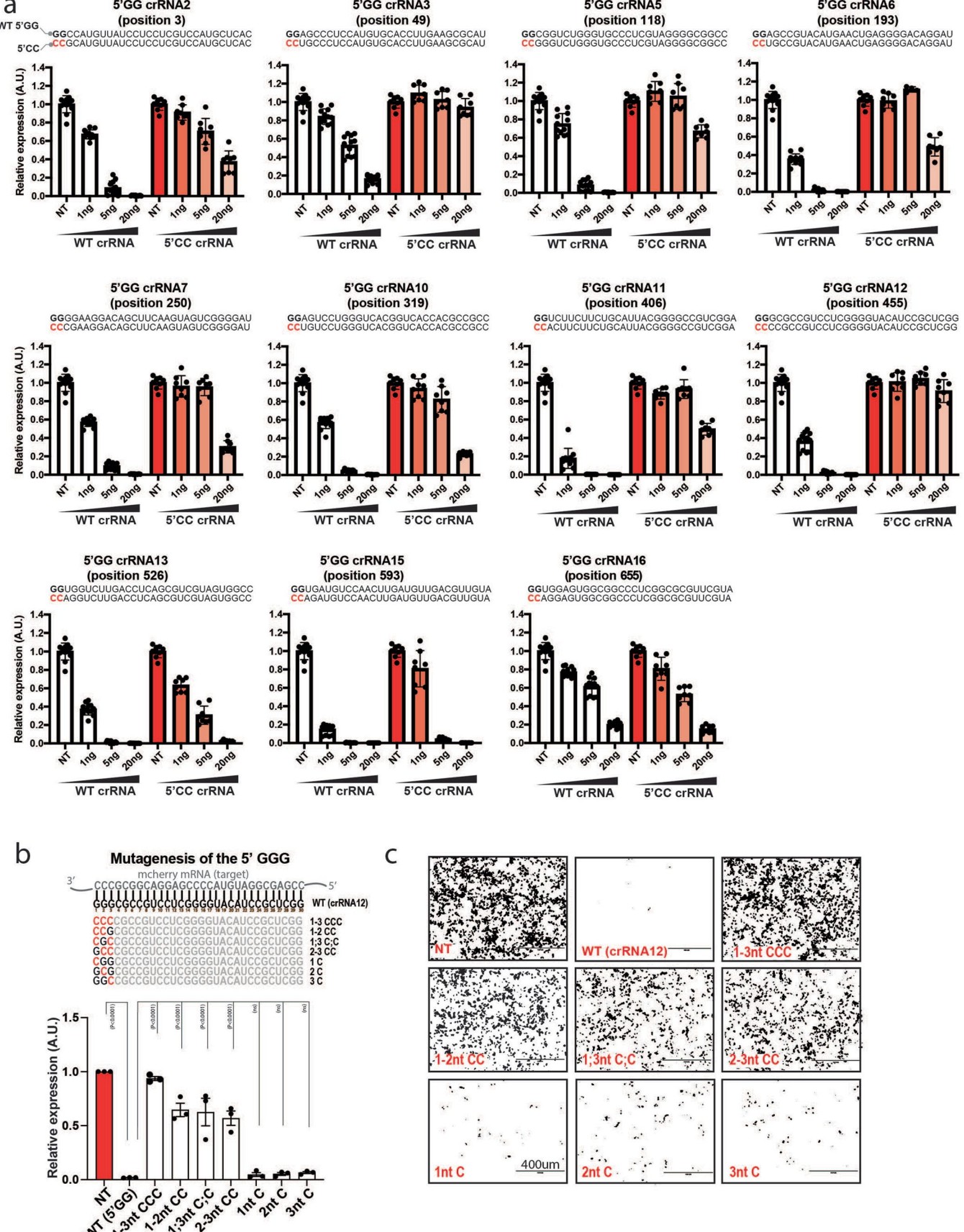

**Extended Data Fig. 5 | See next page for caption.**

**Extended Data Fig. 5 | Mutagenesis study reveals the importance of a 5′ G-rich motif for potent silencing activity of crRNA. (a)** Dose-dependent silencing of mCherry transcript with non-targeting crRNA (NT), 11 unmodified crRNAs (WT) that possess a GG sequence at their 5′end (white bars), or the same 11 crRNAs were the 5′ GG sequence of the spacer is mutated to 5′ CC through 1–3 nucleotides mutagenesis. Data points in the graph are normalized mean fluorescence from 4 different fields of view imaged in $N$ = 2. The data are represented in arbitrary units (A.U.). Errors are SD with a 95% confidence interval. **(b)** Mutagenesis analysis of spacer 1–3 nucleotides (5′ end) examining the impact of C to G substitutions on crRNA silencing efficiency. The nucleotides in red highlight mismatch positions in the spacer sequence. Data points in the graph are averages of mean fluorescence from 4 representative fields of view per condition imaged; $N$ = 3. Errors are SEM and $p$-values of one-way Anova test are indicated (95% confidence interval). The data are represented in arbitrary units (A.U.). **(c)** Representative fluorescence microscopy images show the silencing efficiency of the mCherry transcripts with NT, WT and mutant crRNAs in HEK 293 T cells. NT is a non-targeting control crRNA. Scale bar = 400µm. Similar results were obtained in 3 independent experiments in HEK 293 T cells. $N$ is the number of independent biological replicates. Unprocessed representative images are provided in the Source Data file.

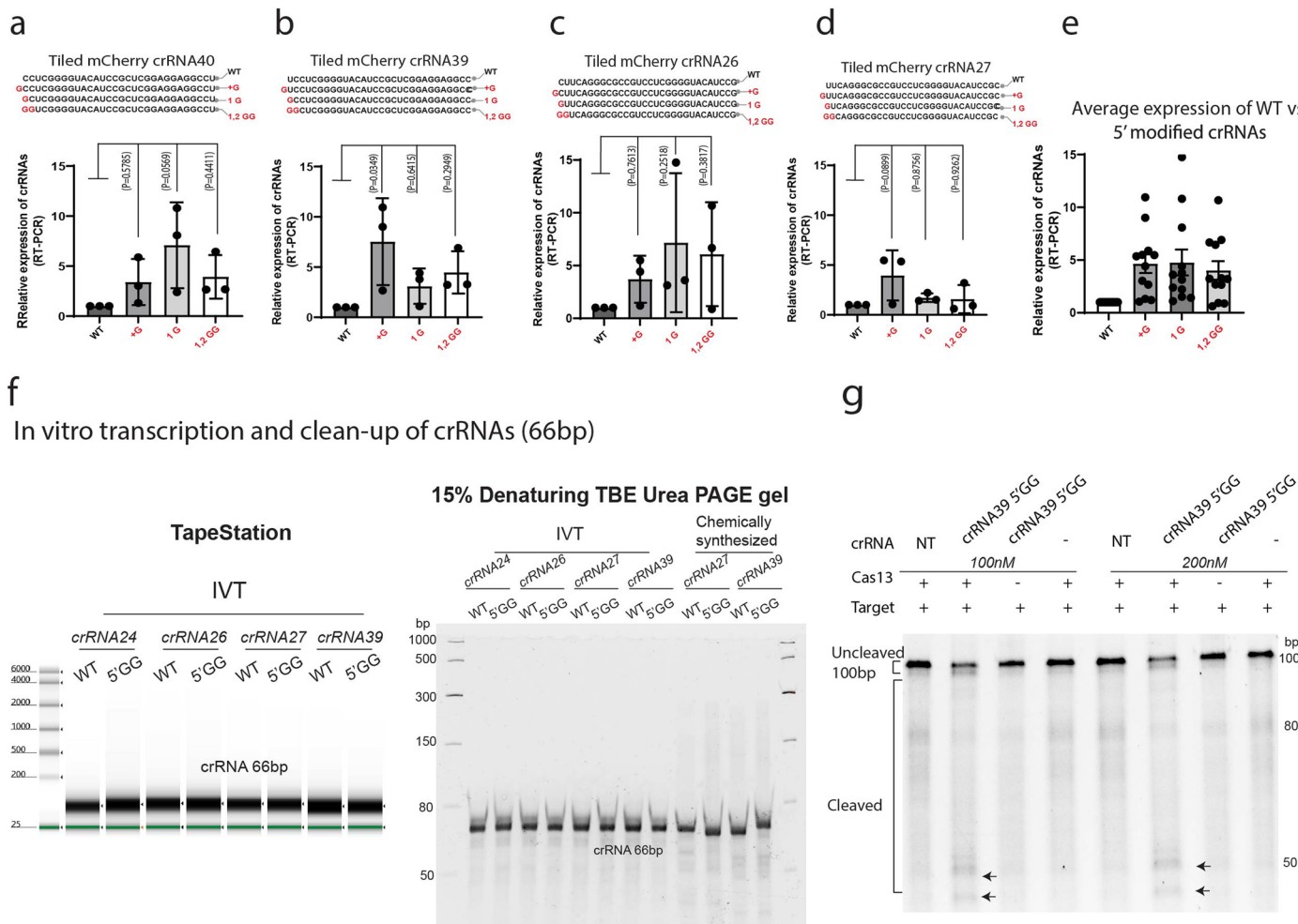

**Extended Data Fig. 6 | 5′ G-rich motif enhances crRNA abundance and PspCas13b cleavage activity. (a-d)** RT-qPCR analysis to assess the expression of WT crRNAs, crRNAs with an extra G at their 5′ end (31-mer spacer), crRNAs with the first spacer nucleotide substituted to a G (30-mer spacer), and crRNAs with the first and second spacer nucleotides substituted to GG (30-mer spacer) for **(a)** crRNA40, **(b)** crRNA39, **(c)** crRNA26 and **(d)** crRNA27. crRNAs expression was measured 48 h post-transfection in HEK 293 T cells, $N = 3$. Data are normalized means and errors are SEM; Results are analysed with one-way Anova test with

$p$-value indicated (95% confidence interval). **(e)** Averaged expression (from a-d) of unmodified or modified crRNAs harbouring G-rich 5′end. Data are normalized means and errors are SEM; **(f)** Quality control analysis of *in-vitro* transcribed crRNAs using TapeStation (left) and 15% Urea PAGE electrophoresis (right); $N = 2$. **(g)** *In vitro* cleavage of *Psp*Cas13b incubated with crRNA39 with a 5′ GG motif against a 5′ 6-FAM labelled 100-nt target ssRNA at 2 h timepoint (see uncropped gels in the Source file); $N = 1$. $N$ is the number of independent biological replicates. Source data are provided as a Source data file.

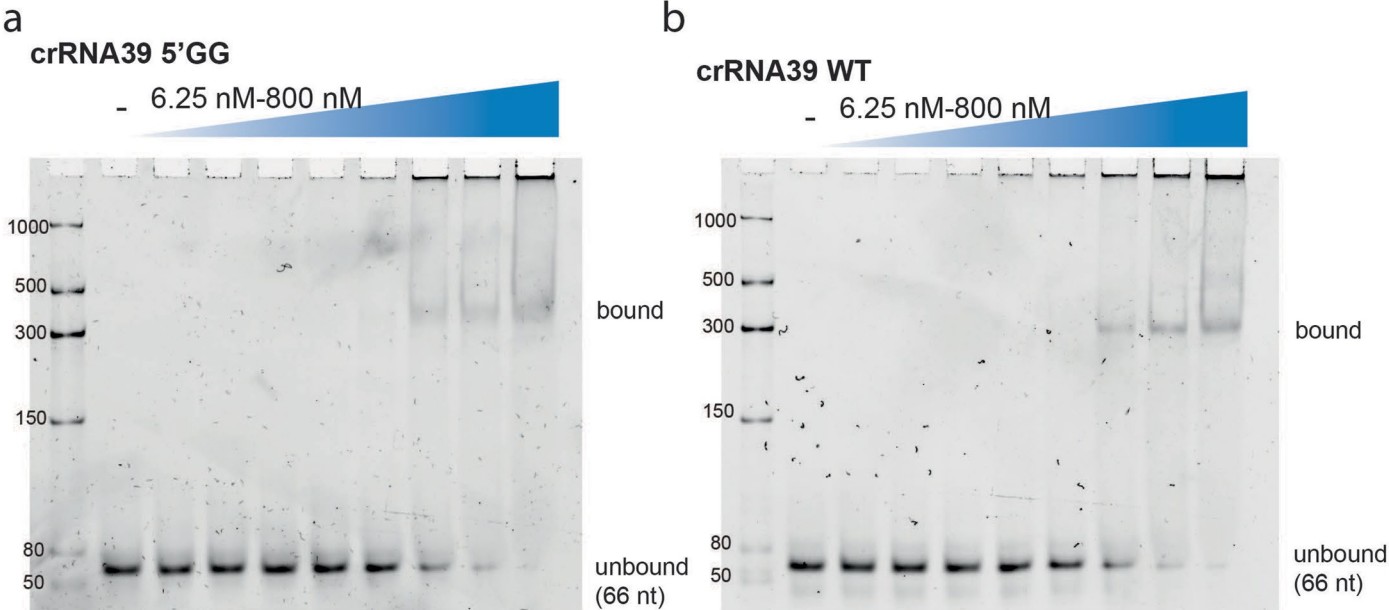

**Extended Data Fig. 7 | 5′G-G motif unlikely to enhance crRNA loading into PspCas13b enzyme.** *Psp*Cas13b binding to **(a)** crRNA39 with a 5′GG motif and **(b)** crRNA39 WT is determined by electrophoretic mobility shift assay (EMSA), *N = 2*. Source data are provided as a Source data file.

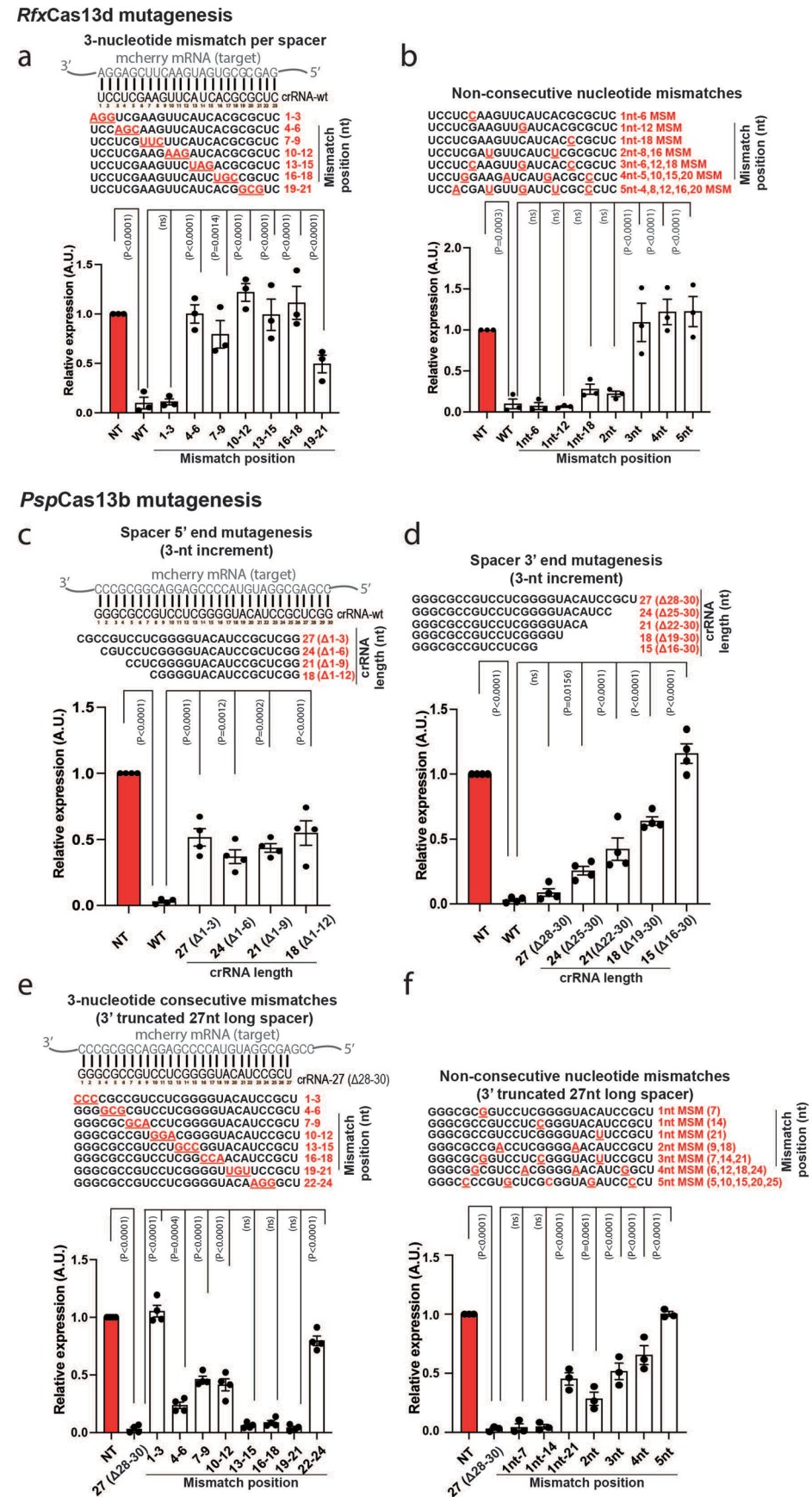

**Extended Data Fig. 8 | See next page for caption.**

**Extended Data Fig. 8 | Comprehensive mutagenesis of *Psp*Cas13b and *Rfx*Cas13d spacer-target interaction revealed that shorter crRNA spacers tolerated fewer mismatches. (a)** Perturbation of spacer-target interaction through spacer mutagenesis to introduce 3-nucleotide consecutive mismatch at various positions of the *Rfx*Cas13d crRNA1 (Fig. 3l) spacer; $N = 3$. **(b)** Perturbation of spacer-target interaction through spacer mutagenesis to introduce various numbers of non-consecutive nucleotide mismatches at various positions of the *Rfx*Cas13d crRNA1 spacer. The nucleotides in red highlight mismatch positions in the spacer sequence; $N = 3$. Silencing of mcherry transcripts by *Psp*Cas13b with a serial shorten crRNAs at the **(c)** 5′ end and **(d)** 3′ end of the spacer (based on *Psp*Cas13b crRNA12 in Fig. 1d); $N = 4$. **(e)** Perturbation of spacer-target interaction through spacer mutagenesis to introduce 3-nucleotide consecutive mismatch at various positions of the 3′ truncated, 27-nt crRNA spacer (crRNA12 Δ28–30); $N = 4$. **(f)** Perturbation of spacer-target interaction through spacer mutagenesis

to introduce various numbers of non-consecutive nucleotide mismatches at various positions of the 3′ truncated, 27-nt crRNA spacer; $N = 3$. Data points in the graph are averages of mean fluorescence from 4 representative fields of view per condition imaged. The data are represented in arbitrary units (A.U.). Errors are SEM and $p$-values of one-way Anova test are indicated (95% confidence interval). $N$ is the number of independent biological replicates. Source data are provided as a Source data file. Note: As opposed to *Psp*Cas13b, *Rfx*Cas13d tolerated consecutive 3-nt mismatches at the 5′end of its spacer but failed to tolerate consecutive 3-nt mismatches at other positions. Single nucleotide mismatches were fully tolerated at positions 6 and 12, but were only partially tolerated at position 16. Two non-consecutive nucleotide mismatches were also partially tolerated (position 8, 16). 3, 4, and 5 non-consecutive nucleotide mismatches led to a complete loss of silencing.

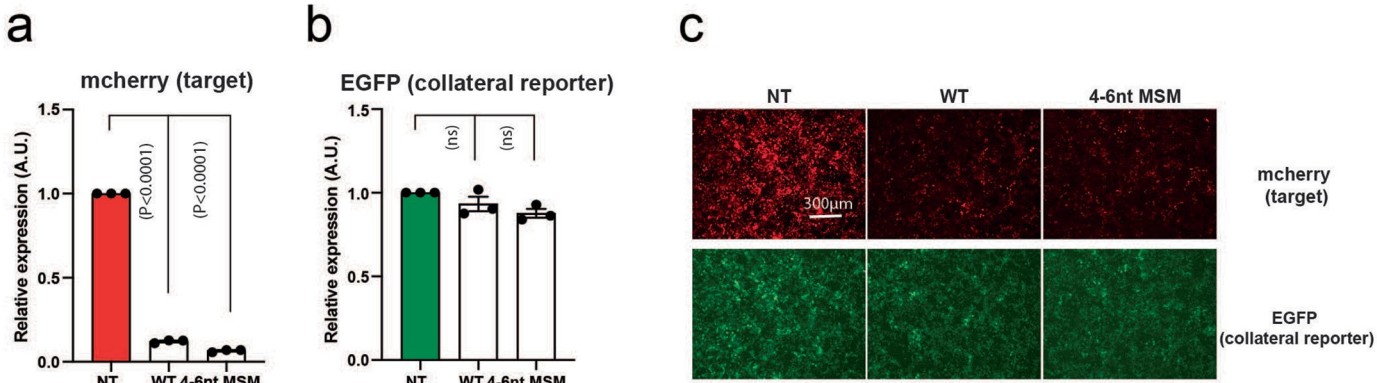

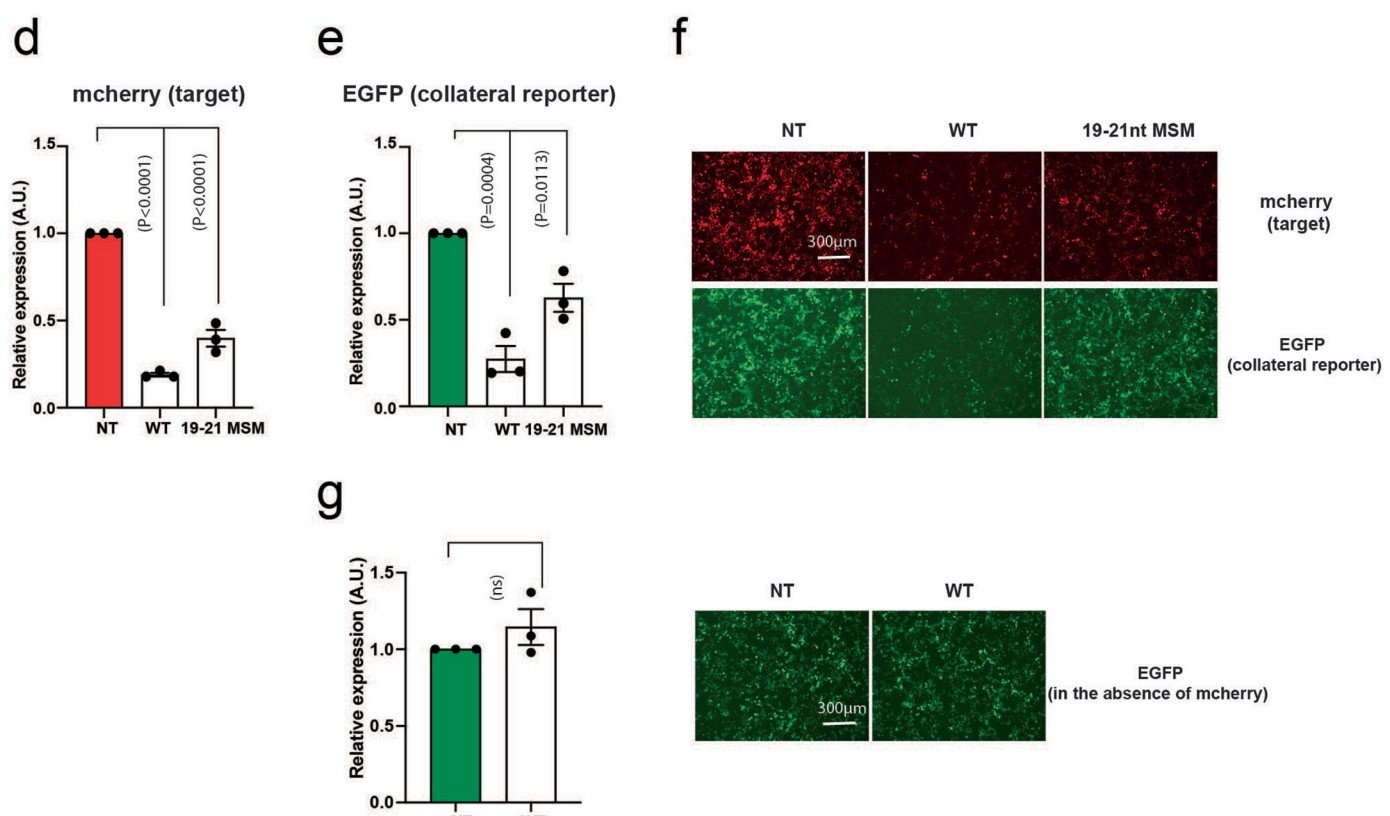

**Extended Data Fig. 9 | *Psp*Cas13b lacks collateral activity in HEK 293T cells compared to *Rfx*Cas13d by fluorescence reporter assays.** Quantifications of **(a)** mcherry and **(b)** eGFP expression and **(c)** their representative fluorescence images in HEK 293T cells transfected with plasmids encoding mcherry, eGFP, *Psp*Cas13b and either non-targeting (NT) crRNA, wildtype (WT) *Psp*Cas13b crRNA12 or mutant (4–6 nt MSM) *Psp*Cas13b crRNA12; $N = 3$. Quantifications of **(d)** mcherry and **(e)** eGFP expression and **(f)** their representative fluorescence images in HEK 293 T cells transfected with plasmids encoding mcherry, eGFP, *Rfx*Cas13d and either non-targeting (NT) crRNA, wildtype (WT) *Rfx*Cas13d crRNA1 or mutant (19–21 nt MSM) *Rfx*Cas13d crRNA1; $N = 3$. **(g)** Quantifications of eGFP expression and its representative fluorescence images in HEK 293T cells transfected with plasmids encoding eGFP, *Rfx*Cas13d and either non-targeting (NT) crRNA or *Rfx*Cas13d crRNA1 (WT) targeting mcherry. Data points in the graph are averages of normalized mean fluorescence from 4 representative fields of view imaged in $N = 3$. The data are represented in arbitrary units (A.U.). Errors are SEM and *p*-values of one-way Anova test are indicated (95% confidence interval). *N* is the number of independent biological replicates. Source data are provided as a Source data file.

## Human transcript input

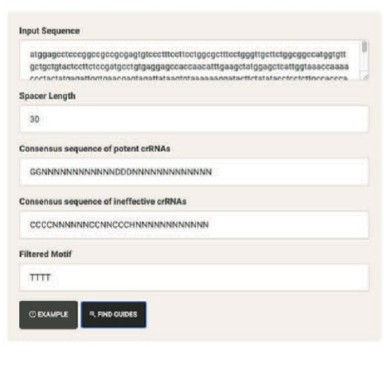

## Non-human transcript input

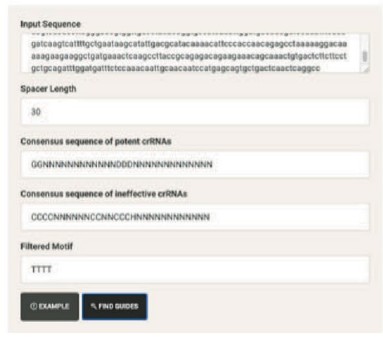

**Extended Data Fig. 10 | PspCas13b crRNA design in-silico tool enables the prediction of potent crRNAs and potential off-targets within the human transcriptome.** *Psp*Cas13b crRNA design tool is open source and available to the wider scientific community at https://cas13target.azurewebsites.net/. In addition to the prediction of the potent crRNAs, the web application can also assess potential off-target transcripts within the human transcriptome for the top 10 potent spacers. When partial matches with human transcripts are found, the users are provided with the percentage of matches, number of mismatches, as well as a link to the NCBI records of these transcripts. For example, the top figure shows the crRNA prediction from a human transcript as the input sequence

(Homo sapiens CD46 molecule variant e, CDS). The output shows the top 10 predicted crRNAs have 100% matching with the input sequence NM_172352 while only one crRNA potentially off targets another CD46 variant m (NM_172358). The bottom figure shows the crRNA prediction from a non-human transcript (SARS-CoV-2 nucleocapsid protein). The 10 predicted crRNAs do not fully match with any human transcripts (no on-target hits). Five crRNAs show a partial match with a few human transcripts (off-target hits), but the mismatches are larger than 6 nucleotides, which are unlikely to silence those partially matched human transcripts according to the mutagenesis study.

# Reporting Summary

## Statistics

For all statistical analyses, confirm that the following items are present in the figure legend, table legend, main text, or Methods section.

| n/a | Confirmed | |
|---|---|---|
| ☐ | ☒ | The exact sample size (*n*) for each experimental group/condition, given as a discrete number and unit of measurement |
| ☐ | ☒ | A statement on whether measurements were taken from distinct samples or whether the same sample was measured repeatedly |
| ☐ | ☒ | The statistical test(s) used AND whether they are one- or two-sided *Only common tests should be described solely by name; describe more complex techniques in the Methods section.* |
| ☒ | ☐ | A description of all covariates tested |
| ☒ | ☐ | A description of any assumptions or corrections, such as tests of normality and adjustment for multiple comparisons |
| ☐ | ☒ | A full description of the statistical parameters including central tendency (e.g. means) or other basic estimates (e.g. regression coefficient) AND variation (e.g. standard deviation) or associated estimates of uncertainty (e.g. confidence intervals) |
| ☐ | ☒ | For null hypothesis testing, the test statistic (e.g. *F*, *t*, *r*) with confidence intervals, effect sizes, degrees of freedom and *P* value noted *Give P values as exact values whenever suitable.* |
| ☒ | ☐ | For Bayesian analysis, information on the choice of priors and Markov chain Monte Carlo settings |
| ☒ | ☐ | For hierarchical and complex designs, identification of the appropriate level for tests and full reporting of outcomes |
| ☒ | ☐ | Estimates of effect sizes (e.g. Cohen's *d*, Pearson's *r*), indicating how they were calculated |

*Our web collection on statistics for biologists contains articles on many of the points above.*

## Software and code

Policy information about availability of computer code

| Data collection | The bioinformatic codes for the design of predicted potent crRNA is available at https://github.com/david-ma/cas13. crRNA design web-server described in this study is available at: https://cas13target.azurewebsites.net/. |
|---|---|
| Data analysis | The design of in silico prediction tool is based on design principles learned from the experimental data presented in this article. The R (version 4.2.2) programming language was used for coding and the R Shiny framework (version 1.7.4) was used to develop the software application. The software program is deployed as a web application using the Microsoft Azure platform. Briefly, the software takes an RNA or DNA sequence and generates all single-nucleotide tiled spacers using the input sequence. The program then removes all spacer sequences that possess more than three consecutive T bases (>3T) that are predicted to act as a transcription termination signal and could yield premature crRNAs. The algorithm scores the remaining spacer sequences based on their nucleotide composition and position. Spacers with a G nucleotide at the first or second positions receive a maximum score of +60 each. In contrast, a C nucleotide at spacer position 1, 2, 3, or 4 receives a penalty score of -60, -60, -50, and -40 each, respectively. Additionally, C bases at positions 11, 12, 15, 16, and 17 receive a -5 score each. All other nucleotides or spacer positions that did not show any enrichment in the potent and ineffective crRNA cohorts receive a score of 0. The algorithm then calculates the cumulative score for each spacer and ranks them accordingly. As a result, the top spacers with high scores are enriched with G bases at 1st and 2nd positions, and depleted from C bases at positions 1, 2, 3, 4, 11, 12, 15, 16, and 17, and are predicted to yield potent silencing. Conversely, the lowest scoring spacers at the bottom of the list are enriched with C bases at positions 1, 2, 3, 4, 11, 12, 15, 16, and 17 and are predicted to yield ineffective silencing. The prediction accuracy of the algorithm is supported by in silico analysis and functional validation data in Figures 3, 4, and 5 and Supplementary Figures 3, 4, 5 & 6. In addition to identifying crRNAs for PspCas13b, the web application also assesses potential off-target effects of the top 10 predicted potent spacer sequences based on their sequence complementarity with various RNA transcripts in the human transcriptome. The web application integrates the NCBI BLAST (Basic Local Alignment Search Tool) command line tool for this purpose and reports the on-target and off-target(s) within the human transcriptome (GRCh38.p14; Annotation Name: GCF_000001405.40-RS_2023_10 (October 2, 2023)). The webpage displays the percentage of match and |

number of nucleotide mismatches with other RNA molecules that possess sequence complementarity with the selected spacer sequence. The software categorizes off-target effects as nonexistent when the number of mismatches is greater than 15. Building on mutagenesis studies, we predict that crRNAs with partial sequence complementarity, involving 6 nucleotide mismatches or longer, are likely to lose their silencing activity. Consequently, any off-target transcript displayed on the prediction webpage with 6 or more mismatches is considered unlikely to be silenced. In cases where potential off-targets are identified, the output provides a link to the NCBI records of these human transcripts. To expedite the processing of this web application, we recommend utilizing DNA or RNA coding sequences (CDS) with a length shorter than 1000 nucleotides as input.

This PspCas13b crRNA design tool developed in this study (version 1) is open source and available to the wider scientific community at https://cas13target.azurewebsites.net (Extended Figure 10).

The predicted RNA secondary structures and minimum free energy were generated using the RNAfold program (ViennaRNA webservices; Lorenz, R. et al. ViennaRNA Package 2.0. Algorithms Mol. Biol. (2011) doi:10.1186/1748-7188-6-26.).

RNA hybridization/interaction energy, crRNA spacer GC content, and A/U/G/C content. The R package 'ggseqlogo' was used to assess nucleotide preference in crRNA spacer and PFS sequence. Delta probability graphs of spacer nucleotides were generated with Matplotlib.

Data analyses and visualizations (graphs) were performed in GraphPad Prism software version 9, unless stated otherwise.

For manuscripts utilizing custom algorithms or software that are central to the research but not yet described in published literature, software must be made available to editors and reviewers. We strongly encourage code deposition in a community repository (e.g. GitHub). See the Nature Portfolio guidelines for submitting code & software for further information.

# Data

Policy information about availability of data

All manuscripts must include a data availability statement. This statement should provide the following information, where applicable:
- Accession codes, unique identifiers, or web links for publicly available datasets
- A description of any restrictions on data availability
- For clinical datasets or third party data, please ensure that the statement adheres to our policy

All the raw data supporting the findings are available in the source Data file submitted with this manuscript.
All data are available in the main text and supplementary materials. Source Data are available on Figshare (10.6084/m9.figshare.25058588). All key plasmids constructed in this study, their sequences, and maps will be deposited to Addgene upon publication.
Code Availability. The bioinformatic codes for the design of predicted potent crRNA is available at https://github.com/faraz107/cas13target. crRNA design webserver described here is available at: https://cas13target.azurewebsites.net/.
Human transcriptome (GRCh38.p14; Annotation Name: GCF_000001405.40-RS_2023_10 (October 2, 2023)) and Human reference proteomes (UP000005640) are used in this study

# Research involving human participants, their data, or biological material

Policy information about studies with human participants or human data. See also policy information about sex, gender (identity/presentation), and sexual orientation and race, ethnicity and racism.

| Reporting on sex and gender | N/A |
|---|---|
| Reporting on race, ethnicity, or other socially relevant groupings | N/A |
| Population characteristics | N/A |
| Recruitment | N/A |
| Ethics oversight | N/A |

Note that full information on the approval of the study protocol must also be provided in the manuscript.

# Field-specific reporting

Please select the one below that is the best fit for your research. If you are not sure, read the appropriate sections before making your selection.

☒ Life sciences ☐ Behavioural & social sciences ☐ Ecological, evolutionary & environmental sciences

For a reference copy of the document with all sections, see nature.com/documents/nr-reporting-summary-flat.pdf

# Life sciences study design

All studies must disclose on these points even when the disclosure is negative.

| Sample size | The sample sizes were determined to match the standards in comparable studies available in the literature (Chunlong Xu et at, Nat Methods, |
|---|---|

| | |
|---|---|
| Sample size | 2021). |
| Data exclusions | Experiments and protocols were optimized in pilot assays before generating high-quality publication data. No data was excluded from the analysis. |
| Replication | All experiments were repeated at least 3 times as biological replicates with the following exceptions:<br>- As mentioned in the figure legend, Data in Fig 2b (screening 61 single-base tiled crRNAs targeting mCherry mRNA) was performed in two biological replicates due to the large size of crRNAs screened.<br>After the initial optimization of the experimental conditions, all experiments were reproducible in independent experiments. RNA targeting with CRISPR-Cas13 is well-established and similar silencing experiments using various Cas13 tools have been reported by independent researchers. |
| Randomization | No randomization was used in this study. Due to the small sample, randomization was not relevant for this study. Covariates were controlled for by running controls in parallel whenever is applicable. Appropriate controls (e.g. non targeting crRNAs, dpspCas13b, crRNA alone, loading controls in WB, and crRNA dose-dependent silencing) were used throughout the study. |
| Blinding | No blinding was used in this study. Blinding is not relevant to this study as RNA targeting with Cas13 is well-established in the field by independent groups using assays that do not require blinding (Chunlong Xu et at, Nat Methods, 2021). Most experiments were performed, analysed, and confirmed by independent co-authors in our labs. |

# Reporting for specific materials, systems and methods

We require information from authors about some types of materials, experimental systems and methods used in many studies. Here, indicate whether each material, system or method listed is relevant to your study. If you are not sure if a list item applies to your research, read the appropriate section before selecting a response.

## Materials & experimental systems

| n/a | Involved in the study |
|---|---|
| ☐ | ☒ Antibodies |
| ☐ | ☒ Eukaryotic cell lines |
| ☒ | ☐ Palaeontology and archaeology |
| ☒ | ☐ Animals and other organisms |
| ☒ | ☐ Clinical data |
| ☒ | ☐ Dual use research of concern |
| ☒ | ☐ Plants |

## Methods

| n/a | Involved in the study |
|---|---|
| ☒ | ☐ ChIP-seq |
| ☐ | ☒ Flow cytometry |
| ☒ | ☐ MRI-based neuroimaging |

## Antibodies

| | |
|---|---|
| Antibodies used | Antibidies used in this study are listed below:<br>- Monoclonal β actin antibody (AC-74) (source: mouse, western blot application: 1:2000) Sigma-Aldrich A2228<br>- Monoclonal ANTI-FLAG antibody (M2) (source: mouse, western blot application: 1:2000 Sigma-Aldrich F1804<br>-Monoclonal ANTI-HA-Tag (6E2) antibody (source: mouse, western blot application: 1:1000) Cell Signalling Technology 2367<br>-Monoclonal ANTI-mCherry (E5D8F) antibody (source: rabbit , western blot application: 1:1000) Cell Signalling Technology 43590<br>- Polyclonal c-ABL antibody (source: rabbit, western blot application: 1:2000) Cell Signaling Technology 2862<br>- Monoclonal PE/Cyanine7 anti-human β2-microglobulin Antibody (2M2) (FACS application: 1:500) BioLegend 316317<br>- Polyclonal (Horseradish peroxidase) HRP conjugated goat anti-mouse IgG secondary Antibody (western blot application: 1:10,000) Abcam ab97023<br>- Polyclonal HRP conjugated goat anti-rabbit IgG secondary Antibody (western blot application: 1:2000) Abcam ab205718 |
| Validation | We used commercial antibodies validated by the suppliers. We confirmed the validation as we used unstained cells, untransfected cells, a secondary-antibody only control, and other appropriate controls to validate the specificity of various antibodies we used in this study.<br>Monoclonal Anti-β-Actin antibody has been used in western blot and two-dimensional gel immunoblot.<br>Anti-flag AB: For highly sensitive and specific detection of FLAG fusion proteins by immunoblotting, immunoprecipitation (IP), immunohistochemisty, immunofluorescence and immunocyotchemistry. Optimized for single banded detection of FLAG fusion proteins in mammalian, plant, and bacterial expression systems.<br>HA-Tag (6E2) Mouse mAb detects recombinant proteins containing the HA epitope tag. The antibody recognizes the HA-tag fused to either the amino or carboxy terminus of targeted proteins in transfected cells.<br>mCherry (E5D8F) Rabbit mAb detects mCherry-tagged proteins (either N-terminal tagged or C-terminal tagged) exogenously expressed in cells. Please note that the mCherry tags add approximately 28kDa to the molecular weight of the fusion protein.<br>Polyclonal antibodies are produced by immunizing animals with a synthetic peptide corresponding to residues surrounding Pro580 of human c-Abl. Antibodies are purified by protein A and peptide affinity chromatography.<br>Monoclonal PE/Cyanine7 anti-human β2-microglobulin Antibody: The antibody was purified by affinity chromatography and conjugated with PE/Cyanine7 under optimal conditions. |

# Eukaryotic cell lines

Policy information about cell lines and Sex and Gender in Research

| | |
|---|---|
| Cell line source(s) | HEK 293T (ATCC CRL-3216) |
| Authentication | Cell lines were authenticated by the supplier ATCC. We did not perform any additional authentication upon reception. We made a bulk stocks for each cell line after recovering from the original frozen vials. We discard the cells after ~20 passages, and thaw new cells from the liquid nitrogen stocks. Cell morphology was monitored at each passage by microscope. |
| Mycoplasma contamination | Cells were monthly tested for mycoplasma contamination (QPCR based test and microscopy) and were mycoplasma negative. |
| Commonly misidentified lines (See ICLAC register) | No commonly misidentified cell lines were used in this manuscript. |

# Flow Cytometry

## Plots

Confirm that:

☒ The axis labels state the marker and fluorochrome used (e.g. CD4-FITC).

☒ The axis scales are clearly visible. Include numbers along axes only for bottom left plot of group (a 'group' is an analysis of identical markers).

☒ All plots are contour plots with outliers or pseudocolor plots.

☒ A numerical value for number of cells or percentage (with statistics) is provided.

## Methodology

| | |
|---|---|
| Sample preparation | For B2M surface marker staining, up to 1x106 cells were incubated in 50 µL of PBS/2% FBS (v/v) containing B2M antibody for 30 minutes on ice in the dark. The cells were then washed twice with 200 µL of PBS/2% (v/v) FBS before being re-suspended in 200 µL of PBS/2% FBS (v/v) for flow cytometry analysis.<br>Supplementary Table 5 lists the antibodies that were used in this study including for FACS analysis.<br>- Monoclonal PE/Cyanine7 anti-human β2-microglobulin Antibody (2M2) (FACS application: 1:500) BioLegend 316317 |
| Instrument | Flow cytometry analysis was performed using either the FACS Symphony Cell Analyzer A5 or A3 (BD Biosciences). |
| Software | All flow cytometry profiles were analyzed using FlowJo V10 software (Tree Star Inc). |
| Cell population abundance | For antibody-stained samples, the cell purify was compared to a unstained control, a secondary-antibody only control or a IgG isotype control. |
| Gating strategy | The gating strategies are detailed in the Supplementary Fig. 1 |

☒ Tick this box to confirm that a figure exemplifying the gating strategy is provided in the Supplementary Information.

