## [Peer Review File · Nature Structural & Molecular Biology]

Peer Review Information

Manuscript Title: Single-base tiled screen unveils design principles of PspCas13b for potent and off-target-free RNA silencing

Corresponding author name(s): Mohamed Fareh

Editorial Notes:

**Redactions –
unpublished data**

Reviewer Comments & Decisions:

Decision Letter, initial version:

Message: 21st Jun 2023

Dear Dr. Fareh,

Thank you again for submitting your manuscript "Single-base tiled screen reveals new design principles of PspCas13b for potent and off-target-free RNA silencing". I apologise for the delay in responding, which resulted from the difficulty in timely obtaining suitable referee reports and from editorial absences in our team precluding us from deliberating earlier. Nevertheless, we now have comments (below) from the 3 reviewers who evaluated your paper. In light of these reports, we remain interested in your study and would like to see your response to the comments of the referees, in the form of a revised manuscript.

You will see that all experts appreciate the potential of PspCas13b as a tool for consistently generating specific KDs with limited off-target effects. However, the experts voice similar concerns about significant parts of the work: lack of important controls, better benchmarking, etc. In addition, they raise important technical issues that need to be addressed in their entirety in a revised manuscript. Finally, Reviewer #1 (R#1), and to some extent Reviewer #2 (R#2), raises mechanistic issues that if addressed would significantly elevate the value of this work.

On the technical front, all experts bring up the lack of pertinent controls, such as crRNA-

only and catalytically inactive dCas9, which must be addressed. Furthermore, both R#1 and R#2 request additional to the 1% agarose gels-experiments to test crRNA length and integrity (R#1 point 4, R#2 point 4). Multiple referees additionally want to see the proteomics experiments either more appropriately benchmarked (R#3 with promiscuous Cas13s and relevant controls) or orthogonally validated with RNA-seq (both R#1 and R#3). Moreover, R#1 and R#2 request improved validation/representation of the potential off-target effects, while R#3 notes that the manuscript would benefit from additional citations and better contextualising of the findings with respect to existing literature. R#1 deems (point 1) that textually highlighting the difference in your approach vis-à-vis what is already known would help the reader. Finally, R#2 in points 1,2, and R#1 in points 3,4,5 amongst others, offer valuable guidance on how to provide additional mechanistic insight to support the functional findings. We editorially agree that such experiments would further boost the manuscript and would encourage you to follow the referee suggestions where feasible.

As always, please be sure to address/respond to all concerns of the referees in full in a point-by-point response and highlight all changes in the revised manuscript text file. If you have comments that are intended for editors only, please include those in a separate cover letter.

We expect to see your revised manuscript within 6 months. If you cannot send it within this time, please contact us to discuss an extension; we would still consider your revision, provided that no similar work has been accepted for publication at NSMB or published elsewhere.

Reporting Summary:

When submitting the revised version of your manuscript, please pay close attention to our [href="https://www.nature.com/nature-portfolio/editorial-policies/image-integrity">Digital Image Integrity Guidelines](https://www.nature.com/nature-portfolio/editorial-policies/image-integrity). and to the following points below:

- that unprocessed scans are clearly labelled and match the gels and western blots presented in figures.
- that control panels for gels and western blots are appropriately described as loading on sample processing controls

-- all images in the paper are checked for duplication of panels and for splicing of gel lanes.

Data availability: this journal strongly supports public availability of data. All data used in accepted papers should be available via a public data repository, or alternatively, as Supplementary Information. If data can only be shared on request, please explain why in your Data Availability Statement, and also in the correspondence with your editor. Please note that for some data types, deposition in a public repository is mandatory - more information on our data deposition policies and available repositories can be found below: <https://www.nature.com/nature-research/editorial-policies/reporting-standards#availability-of-data>

Nature Structural & Molecular Biology is committed to improving transparency in authorship. As part of our efforts in this direction, we are now requesting that all authors identified as 'corresponding author' on published papers create and link their Open Researcher and Contributor Identifier (ORCID) with their account on the Manuscript Tracking System (MTS), prior to acceptance. This applies to primary research papers only. ORCID helps the scientific community achieve unambiguous attribution of all scholarly contributions. You can create and link your ORCID from the home page of the MTS by clicking on 'Modify my Springer Nature account'. For more information please visit please visit www.springernature.com/orcid.

[Redacted]

Sincerely,

Dimitris Typas
Associate Editor
Nature Structural & Molecular Biology
ORCID: 0000-0002-8737-1319

Referee expertise:

Referee #1: CRISPR systems, RNA interference regulation

Referee #2: CRISPR systems/techniques development

Referee #3: CRISPR systems/Cas13 development, Computational CRISPR

Reviewers' Comments:

Reviewer #1:
Remarks to the Author:
Summary

Hu et al explore crRNA design principles in order to select for highly active crRNAs for PspCas13b for use in mammalian RNA KD experiments. They show that by using a single-nt tiled screen across several transcripts, read out via fluorescence and downstream computational analysis that the presence of a 5' rich sequence as well as the presence of an internal G-rich motifs can yield potent crRNAs for RNA KD, and inserting these into poor performing crRNAs can often rescue their performance. The authors generate an online tool to help with the selection of these potent crRNAs. The authors also carry out mismatch tolerance analysis and show that PspCas13b can tolerate several mismatches. Finally, the authors carry out proteomics in attempt to observe whether collateral cleavage of other RNAs occurs and contributes to the off-target profile of PspCas13b, and they observe little off-target activity at the protein.

While these observations will certainly be very useful to researchers wishing to use PspCas13b for RNA KD and potentially other applications, as getting consistent KD with Cas13 proteins has been a topic of discussion and great interest, the authors were not able to demonstrate at the molecular level why these 5'G containing crRNAs are more active, and this is likely of most interest to and scrutiny by the readership of this journal.

Furthermore, as you will see below several claims are not well supported and require additional experiments, or a more specific compare and contrast to the previous literature. For example, the authors make claim of increased catalytic efficiency with their crRNA designs but don't any provide biochemical support for this claim (catalytic efficiency has a very specific meaning biochemically).

Some major gaps and minor concerns must be addressed before the manuscript can be considered for publication, because currently this reviewer feels this manuscript would be better suited to a more specialized genomics/method-development style of journal.

Major Comments

1. In the Abstract and throughout the text, the authors claim no off target or collateral cleavage of PspCas13b but need to qualify this statement with "in HEK 293T cells". Interestingly, this is in conflict with a previously published reports: <https://academic.oup.com/nar/article/50/11/e65/6542487> and <https://www.nature.com/articles/s41586-021-03886-5> that observes off-target collateral cleavage by PspCas13b in HEK293T cells. This needs to be more specifically addressed in the text. What is different about your approach that avoids this? This needs to be communicated to the reader. In addition, the second reference linked above isn't not cited and needs to be as it shows PspCas13b collateral cleavage data!

2. I understand while it's likely the KO at the protein level is due to specific cleavage of the RNA, is there a chance it's either the Cas13:gRNA complex binding and not cleaving and helping to stall ribosomes and/or the crRNA alone potentially acting as an ASO-like molecule? I think it is always key to always include dCas13 (HEPN nuclease-dead mutant) and crRNA-only controls in these experiments to make sure the phenomenon your measuring is what you think it is.

3. The authors claim that PspCas13b offers superior specificity due to its mismatch (in)tolerance. With this claim it's important to make comparisons to other Cas13s that have had their specificities assessed in the literature- are they less or more tolerant? Relatedly in the abstract and throughout the text you claim 30-nt. guides are inherently more specific- while this is true in terms of finding unique sites in the transcriptome, it's not the only consideration. One also needs to factor in an idea known as the "excess energy" phenomenon where shorter crRNAs are more sensitive to mismatches as the per base energetic penalty for a mismatch relative to the free energy of the whole hybridization interaction is proportionally higher. More can be read about this here: <https://doi.org/10.1016/j.cels.2016.12.010>. Have the author's considered generating shorter crRNAs and carrying out mismatch analysis? I think this is an important consideration if you're trying to claim superior specificity.

4. One of the most important findings in this manuscript is how the presence of the a 5' GG sequence on the crRNA leads to much higher KD efficiency. While it was briefly mentioned in the discussion that part of this may be due to the strong preference for RNA pol III has for starting U6 promotor driven transcription at a G, it should be made clearer in the results that this is likely to contribute to the performance of these gRNAs. Testing purified in vitro transcribed crRNA goes part way to addressing the affect of this contribution, however this reviewer cannot interpret these data as they stand. The reason being that T7 polymerase also transcribes RNA most efficiently when starting transcription

with a 5' GG, and from experience and evidence in the literature, starting T7 transcription with a non G can help to generate truncation proteins due to slow initial processivity and polymerase drop off. The authors show a 1% agarose gel to demonstrate they made full length products in all cases. However, this resolution isn't sufficient in my opinion to claim there isn't a population of truncated RNAs present in the non G starting products. Did the authors notice differences in yield with these different RNA IVTs? If so, it might be worth commenting on this in the text. In addition, I think it's pertinent to run these products out on a 12-15% UREA PAGE gel where you can get close to single nt resolution to see if the non-5'G containing RNAs contain more truncation products. If it turns out to be the case, then this likely also explains the preference for 5'G containing IVTed crRNAs as well.

5. Given NSMB places a strong emphasis on functional and mechanistic understanding of how molecular components in a biological process work together, the reviewer would have liked to see additional experiments that try to decipher in addition to transcriptional yields why 5'G crRNAs are better at RNA-targeting (if such mechanisms exist). The authors should try and obtain purified PspCas13b and carry out crRNA-binding assays (EMSA for example) as well as RNA-cleavage assays to demonstrate at what stage of the Cas13 reaction the 5'G is assisting in boosting the activity of the enzyme. If this is outside the expertise of the lab, I think it a collaboration would be required to provide a sufficient molecular understanding for this phenomenon (and for journal fit).

6. It is great that the authors generated an algorithm for the selection of potent crRNAs for PspCas13b. Unfortunately, in my opinion the tool falls short of many other CRISPR-Cas tools as it doesn't include a off-target scoring. Given you have some sense of mismatch tolerance, would the authors consider adding an off-target scoring parameter to the algorithm to make it more useful to users?

7. Relatedly, the addition of the proteomics is a nice touch and a good orthogonal way to look at off-targets. Are the authors concerned at all that the wide distribution of protein half-lives across the proteome may obscure some off-target effects. I mention this because previous reports have used RNA-seq to measure PspCas13b off-target effects (see link #2 in point #1 above) and noticed a drastically different landscape of off-targets. Have the authors considered an RNA-seq experiment to make this comparison more directly?

Minor Comments

1. Abstract: Is it necessary to define an acronym for your screen (SiBTil)? It's only used once in the main text (where its first defined also), suggesting it's not really necessary to define at all.
2. Line 55: I would avoid using the term Prokaryotic RNAi to refer to Cas13 containing systems. To me this evokes prokaryotic Argonauts. I would replace this with RNA-guided nuclease to avoid confusion. Likewise in line 57 and potentially elsewhere.
3. Line 62: It might be best if you define to the reader what a "spacer" is i.e. it's the programmable part of the crRNA.
4. Line 73: For completeness is should be "CRISPR-Cas enzymes"
5. Line 208 should read "CRISPR-Cas variants"
6. Line 238: The authors call the consensus sequence for potent and ineffective crRNAs "a formula" this in my opinion is an odd word choice. I would suggest "consensus sequence" or "sequence motif"
7. Line 251: confusing sentence. Please rephrase.
8. Line 256-260 this section is referring to the wrong figure, please correct.

9. Figure 2 legend: One needs to be clearer that the BCR-ABL1-mCherry is a transgene on a plasmid and it's mCherry fluorescence that is being measured, not BCR ABL expression per se. Likewise in Figure 7 its not made clear that BCR-ABL1-mCherry is a single transcript. I would argue this really isn't an endogenous protein either given it's on a plasmid. To make that claim one should target for example some CD markers and do flow like Wessel et al have previously done.
10. Line 734/Methods: Very detail Methods on some standard molecular biology techniques. For example, I'm not sure specific μ L volumes of DNA used in cloning/transformations etc. is required. If anything, these should be expressed in total ng etc. amounts. It otherwise feels overly superfluous.
11. Sup fig 1A... misleading use of crRNA concentration. Is this the concentration of the plasmid correct and not the final crRNA product? IC50 isn't usually calculated for plasmids the change of units from pM to fM is also confusing. One usually needs more than four points to accurately fit a IC50 curve. Please clarify here and correct where necessary.
12. Typo in Figure 3 ("tagreting"; 3E onwards)
13. qRT-PCR how was the data normalized? Was the ddCt method used? which housekeeping genes were used in each case?

Reviewer #2:

Remarks to the Author:

This manuscript by Hu and coworkers describes an approach for designing highly active and effective pspCas13b crRNAs for RNA knockdown in mammalian cells. The authors do this using a mixture of experimental and computational approaches. The authors find that pspCas13b guide RNAs with a 5' GG are more likely to be highly active, likely due to enhanced transcription of the guide RNA. The mass spec results are really nice. Overall, this is an interesting paper with findings that are of interest to the field. My major comments have more to do with contextualizing the author's results, and performing a few additional control experiments.

Major comments:

1. In Fig. 6, the authors introduce multiple consecutive or non-consecutive mismatches into guide RNAs targeting an mCherry mRNA and observe relatively few effects on cis targeting for sets of three consecutive mismatches. This is quite different from results obtained with other Cas13 orthologs, where one typically observes dramatic shifts in Cas13's trans cleavage activity when 2 consecutive mismatches are introduced. To address this concern, the authors should perform some experiments to look at the trans/collateral cleavage, in addition to cis cleavage, using their mismatched crRNAs (this would go beyond the results shown in Fig. 7, which focus on a single, fully-matched crRNA).
2. For the experiments described in Fig. 4h/i, the results could be influenced by IVT activity, and not just crRNA activity per se. It seems like the crRNA concentrations vary widely, as shown in Supplementary Fig. 6. If the authors want to make claims about potency, they should use chemically synthesized crRNAs, which have much higher purity than IVT products. Our lab, and others, have found that IVT-produced crRNAs are highly variable in their ability to be transcribed. To address this concern, the authors should confirm a few of their findings related to potency using chemically synthesized crRNAs.

3. One challenge with reporter-based experiments is that they cannot distinguish between mRNA cleavage and translational inhibition of the reporter. To address this concern, the authors should perform some controls using catalytically inactivated mutants of *pspCas13b* to ensure that the effects they are observing are not due to translational inhibition, RNA interference, or other Cas13-cleavage-independent mechanisms.

4. The authors mention that they used 1% agarose gel electrophoresis to evaluate crRNA length and integrity. This method does not give high enough resolution to detect issues with crRNA integrity, which can involve missing just a few nucleotides in length. The authors should use denaturing PAGE, or other methods with greater length resolution (e.g. TapeStation, Bioanalyzer).

Minor comments:

1. An intriguing finding in this manuscript is the considerable cell-to-cell variability in reporter fluorescence levels, which is particularly evident in the FACS plots shown in Fig. 5a and 5c. This heterogeneity is particularly pronounced for the inefficient crRNAs, and in the NT controls. This is a somewhat unexpected result. Is B2M expression typically this variable from cell to cell?

2. In Fig. 7e, *PspCas13b* appears to be downregulated by nearly 2-fold. What is going on here?

3. In Fig. 7e, somewhat different fold-changes in BCR-ABL1 and eGFP expression are observed. Is there a reason for this difference?

4. In Fig. 7f, the axes labels are very confusing. Specifically, it does not make sense to have log₁₀-transformed axes that range from only 1.36 to 1.56 (this would suggest a dynamic range of only ~1.6-fold); furthermore the deviation between eGFP/BCR-ABL1 and the line $x=y$ does not line up with the fold-changes shown in Fig. 7e. I am not an expert in mass spectrometry, so perhaps this is a standard metric I am not familiar with, but the math does not seem to line up.

5. Rather than calculating probabilities of *PspCas13b* off-targeting, the authors should look to see if there are any predicted off-target sequences using bioinformatic approaches. This is particularly important because many human genes exist as parts of closely-related families, so the independence assumptions that underlie the calculations the authors have made are not appropriate. Gene sequences are not entirely random, nor are they independent.

Reviewer #3:

Remarks to the Author:

The authors deciphered crRNA design rules for *PspCas13b* that facilitate its use in in vitro RNA targeting applications. RNA interference or targeting is a field with great potential, but with a few associated challenges. Two of these challenges are the lack of specificity (off-target effects) and high collateral activity in vitro, which they seek to remediate using *PspCas13a*. The authors employ a pre-existing dataset to infer important guide design principles and create a simple web interface to apply these design heuristics. I congratulate the authors for their solid work in this manuscript, and I propose several

suggestions (divided into major and minor points) to strengthen this manuscript and that I hope the authors find helpful.

Major points:

- This paper aims to establish crRNA design rules for a given Cas13b ortholog. Other groups have done related work on related Cas13 proteins previously. The authors should discuss these projects both in the introduction and discussion, and state how their approach relates/differs from previous approaches. Important examples include:
 - o Metsky, G. et al. Designing sensitive viral diagnostics with machine learning. *Nature Biotechnology*, (2022).
 - o Chuai, G. et al. DeepCRISPR: optimized CRISPR guide RNA design by deep learning. *Genome Biol.* 19, 80 (2018).
 - o Wessels, H.-H. et al. Massively parallel Cas13 screens reveal principles for guide RNA design. *Nat. Biotechnol.* 38, 722–727 (2020).
 - o Krohannon, A. et al. CASowary: CRISPR–Cas13 guide RNA predictor for transcript depletion. *BMC Genomics.* (2022).
 - o Guo, X. et al. Transcriptome-wide Cas13 guide RNA design for model organisms and viral RNA pathogens. *Cell Genom.* 1, 100001 (2021).
 - o Lin, X. et al. A comprehensive analysis and resource to use CRISPR–Cas13 for broad-spectrum targeting of RNA viruses. *Cell Rep. Med.* 2, 100245 (2021).
- Authors should discuss the results obtained and conclusions reached in the context of the published literature. For example, beyond stating the sequence preferences of the PspCas13b's crRNA, the authors should discuss 1) how these preferences/guidelines differ between PspCas13b and other Cas13b orthologs and Cas13 subtypes and 2) provide a structural explanation/hypothesis for these preferences based on available protein structures of related Cas13b proteins.
- Wherever appropriate, authors should include a crRNA-only control (without Cas13 plasmid). This controls would be appropriate in most panels. Recent reports have suggested that crRNAs by themselves can mediate RNA knockdown and modify protein expression under certain circumstances. See Sharma V. et al. CRISPR guides induce gene silencing in plants in the absence of Cas. *BMC Genome Biology*, 2022 for an example.
- With regards to the collateral activity, there are more sensitive RNA-based methods to probe this activity (as the authors appropriately discuss). I believe the proteomic approach could be relevant here as well, but it is important the authors A) include a crRNA only control using either efficient or inefficient guides (without Cas13) and B) perform the same proteomics approach using RfxCas13d and another Cas13 enzymes with demonstrated collateral activity in vitro. This will be important to assess the sensitivity and specificity of the proteomics approach.
- When performing RNA knockdown in vitro, the sequence of the target RNA cannot be modified. In that case, could you still introduce the GG sequence at the beginning of the crRNA spacer (thereby introducing two synthetic mismatches) and still get efficient knockdown? Or would you only select for guides that naturally start with the GG sequence? If the latter, wouldn't you essentially have a two-nucleotide PFS? Please clarify.
- In the methods section, please include the following:
 - o For the RNA silencing assays with IVT'd crRNAs, the text states that 200ng of crRNA were selected, which differs from that stated in the figures (1-20ng). Please clarify this discrepancy.
 - o Include pertinent information about RfxCas13d use (used in Figure 3).

Minor points:

- Ref. 31 is not relevant for the claim made in lines 97-98. Please update reference.

- Authors should include BFP channel images from Figure 1 as supplementary material to ensure adequate and even expression of Cas13a across cells/conditions.
- In line 227, please clarify what is meant by “filtered” and “unfiltered” samples.
- In Supplementary Table 2 (sequences for oligos used), please include a column detailing what figures & panels used each sequence.

Author Rebuttal to Initial comments

Point-by-point response

Dear reviewers,

We would like to sincerely thank you for giving us the opportunity to revise our work, and for your constructive and valuable feedback. We have revised our manuscript in response to the reviewers’ comments and addressed all the editorial requests. We have performed additional experiments, analyses, restructured the manuscript, and expanded the introduction and discussion. We believe that addressing these comments has greatly improved the quality and the readability of this manuscript. Below, we offer a point-by-point response to each issue raised by the reviewers.

Reviewers’ comments are shown in black while our replies are shown in red with manuscript excerpts highlighted in yellow. All major changes are highlighted in yellow in the manuscript.

Reviewer #1:
 Remarks to the Author:
 Summary

Hu et al explore crRNA design principles in order to select for highly active crRNAs for PspCas13b for use in mammalian RNA KD experiments. They show that by using a single-nt tiled screen across several transcripts, read out via fluorescence and downstream computational analysis that the presence of a 5' rich sequence as well as the presence of an internal G-rich motifs can yield potent crRNAs for RNA KD, and inserting these into poor performing crRNAs can often rescue their performance. The authors generate an online tool to help with the selection of these potent crRNAs. The authors also carry out mismatch tolerance analysis and show that PspCas13b can tolerate several mismatches. Finally, the authors carry out proteomics in attempt to observe whether collateral cleavage of other RNAs occurs and contributes to the off-target profile of PspCas13b, and they observe little off-target activity at the protein.

While these observations will certainly be very useful to researchers wishing to use PspCas13b for RNA KD and potentially other applications, as getting consistent KD with Cas13 proteins has been a topic of discussion and great interest, the authors were not able to demonstrate at the molecular level why these 5'G containing crRNAs are more active, and this is likely of most interest to and scrutiny by the readership of this journal. Furthermore, as you will see below several claims are not well supported and require additional experiments, or a more specific compare and contrast to the previous literature. For example, the authors make claim of increased catalytic efficiency with their crRNA designs but don't any provide biochemical support for this claim (catalytic efficiency has a very specific meaning biochemically).

Some major gaps and minor concerns must be addressed before the manuscript can be considered for publication, because currently this reviewer feels this manuscript would be better suited to a more specialized genomics/method-development style of journal.

→ We thank the reviewer for providing valuable and constructive feedback that has significantly contributed to improving the quality of our data and manuscript. In the following, we provide a detailed point-by-point response addressing the comments and concerns raised by the reviewer.

Major

Comments

1. In the Abstract and throughout the text, the authors claim no off target or collateral cleavage of PspCas13b but need to qualify this statement with "in HEK 293T cells".

→ We added "in HEK293 cells" to the abstract and throughout the manuscript. We also acknowledge this limitation in the 'Limitations of the study' section.

Quote: "In this proof-of-concept study, we investigated the molecular basis of PspCas13b and its specificity in HEK 293T human cell line. Future proteomic analyses in other eukaryotic cell lines, primary cells, and animal models are required for a deeper understanding of the on-target and collateral activity of various Cas13 orthologs".

Interestingly, this is in conflict with a previously published reports: <https://academic.oup.com/nar/article/50/11/e65/6542487> and <https://www.nature.com/articles/s41586-021-03886-5> that observes off-target collateral cleavage by PspCas13b in HEK293T cells. This needs to be more specifically addressed in the text. What is different about your approach that avoids this? This needs to be communicated to the reader.

→ The NAR paper (Yuxi Ai et al, NAR, 2022; link #1) shows a more pronounced collateral activity of RfxCas13d and a limited collateral activity of PspCas13b, consistent with our data. We confirmed this result in the new data we provide in this revised version of the manuscript (Suppl. Fig 10). The RfxCas13d ortholog exhibited pronounced collateral activity against a non-target reporter transcript (eGFP), whereas PspCas13b did not show any collateral activity against this non-target eGFP reporter (also see our response to the 1st comment of Reviewer#2).

However, this paper (Yuxi Ai et al, NAR, 2022) studied overexpressed non target reporter protein as a surrogate marker of collateral activity but did not examine the effect of potential collateral activity on the expression of endogenous proteome.

The paper published by Ozcan et al, 2021 (link #2) presents conflicting data regarding the collateral activity of PspCas13b. The transcriptomic analysis in Extended Data 9i shows that PspCas13b has no off-target activity in HEK293 or U87 cells when comparing the nuclease activity of PspCas13b loaded with a targeting versus a non-targeting crRNA. This is consistent with our proteomic [Redacted] (discussed below in major point 7). However, the transcriptomic data in Fig. 4h (and Extended Data 9j) in the Ozcan manuscript does not have a non-targeting crRNA+Cas control and instead compares differential RNA expression between Cas+crRNA vs crRNA alone (no Cas) for DiCas7-11, PspCas13b, and RfxCas13d. There is significant potential for bias where one condition has no expression of a Cas protein. In fact, the transcriptomic data in Fig. 4h using “no Cas” as a control showed that all the RNA nucleases tested exhibited high levels of off-targeting including the highly specific DiCas7-11 enzyme described in this study (496 significant off-target). In our study, we used Cas+crRNA T vs Cas+crRNA NT, dCas+crRNA T vs dCas+crRNA NT, or crRNA T vs crRNA NT.

Additionally, in our hands, DiCas7-11 exhibited very poor RNA silencing activity compared to PspCas13b or RfxCas13d, and recent studies have shown that DiCas7-11 relies on RNA target binding to activate the protease activity of Csx29, which cleaves the effector Csx30. The RNA nuclease activity of Cas7-11 (Craspase or CASP) appears to play a minor role in bacterial immunity compared to its programmable protease activity. In fact, this nuclease activity is dispensable for protease activation as dCas7-11 (RNA binding) is sufficient to activate Csx29 protease and induce immune response against phage infections (Hu et al, Science, 2022; Strecker et al, Science, 2022).

The field of CRISPR-mediated RNA interference is still in its early stages, marked by ongoing debates in the literature concerning the collateral activity of different Cas13 orthologs. The message that we would like to convey in our study is that accurate examination of the collateral activity of Cas13 requires specific considerations. We highly recommend the use of targeting versus non targeting guides in conjunction with catalytically active Cas13 and dCas13. Additionally, we underscore the importance of incorporating proteomic analysis as a key orthogonal approach. This approach enables a comprehensive understanding of the impact of collateral activity on global protein expression, thereby addressing certain limitations associated

with reporter assays and short-read RNA sequencing. To better convey this message, we have incorporated a paragraph in our revised manuscript that draws attention to potential limitations inherent in the examination of collateral activity among various Cas13 orthologs.

Quote *"It is important to note that conflicting reports exist regarding the collateral activity of different Cas13 enzymes. The controversy in the literature may be attributed to inadequate controls and the absence of rigorous orthogonal approaches to precisely assess the extent of collateral activity on endogenous transcripts and proteins. For example, some earlier studies relied on fluorescence assays as a surrogate to gauge Cas13 collateral activity against non-target, overexpressed reporter transcripts. However, these assays may not fully capture the genuine impact of collateral activity on the endogenous human proteome. Therefore, we suggest that a systematic proteomic analysis of various Cas13 enzymes' activity could provide a more comprehensive understanding of their specificity and the scope of collateral activity in mammalian cells"*.

In addition, the second reference linked above isn't not cited and needs to be as it shows PspCas13b collateral cleavage data!

→ This paper has been cited in this revised manuscript.

2. I understand while it's likely the KO at the protein level is due to specific cleavage of the RNA, is there a chance it's either the Cas13:crRNA complex binding and not cleaving and helping to stall ribosomes and/or the crRNA alone potentially acting as an ASO-like molecule? I think it is always key to always include dCas13 (HEPN nuclease-dead mutant) and crRNA-only controls in these experiments to make sure the phenomenon your measuring is what you think it is.

→ We thank the referee for this suggestion. To address this comment, we've repeated the silencing assays using either catalytically active PspCas13b, catalytically inactive dPspCas13b, or guide RNA (crRNA) alone. Fluorescence reporter assay, western blot, qPCR, and proteomic data all demonstrated no cleavage activity or translational repression when using dPspCas13 or guide RNA alone (**Figure 1e-1g & Figure 7b-7m**). This data indicates that the silencing activity reported here is solely due to the RNA nuclease activity of PspCas13b. We have added these new data to the revised manuscript.

Figure 1

Figure 7

3. The authors claim that *PspCas13b* offers superior specificity due to its mismatch (in)tolerance. With this claim it's important to make comparisons to other Cas13s that have had their specificities assessed in the literature- are they less or more tolerant?

→ The superior specificity of PspCas13 we claim in this study is based on (i) its predicted high fidelity due to the 30-nt long spacer sequence and (ii) the lack of collateral activity. Despite the tolerance of ~4-6 mismatches, the requirement of ~24nt basepaired spacer-target RNA duplex still offers a higher specificity compared to other programmable nucleases with shorter target recognition motifs (e.g., Ago2, Cas9, and RfxCas13d). RfxCas13d has a shorter spacer sequence (23-nt long) which also can tolerate a few mismatches. Moreover, numerous studies have linked RfxCas13d to a promiscuous collateral activity that can degrade nearby non-target RNA molecules in mammalian cells, which may limit its specificity (Li et al, Genome Biology, 2023; Ai et al, NAR, 2022). In this revised manuscript, we have compared the collateral activity of PspCas13b and RfxCas13d using a fluorescence-based assay. We confirmed that target-activated RfxCas13d has a very potent collateral activity against overexpressed eGFP transcript (non-target), which is consistent with previous reports. Conversely, target-activated PspCas13b lacks any significant collateral activity against eGFP (non-target) transcript (see **Suppl. Figure 10** in our response to Reviewer#2).

To further address the referee's comment, we also designed RfxCas13d crRNAs targeting mCherry and compared mismatch tolerance of both RfxCas13d and PspCas13d enzymes. As opposed to PspCas13b, RfxCas13d tolerated consecutive 3nt mismatches at the 5' end of its spacer but failed to tolerate consecutive 3-nt mismatches at other positions.

However, single nucleotide mismatches were fully tolerated at position 12 or position 6, and partially tolerated at position 16. Two non-consecutive mismatches were also partially tolerated (position 8,16). 3, 4, and 5 non-consecutive mismatches weren't tolerated and led to a

RfxCas13d mutagenesis

Suppl. Figure 9

complete loss of silencing. Overall, by comparing the mismatch tolerance/intolerance of PspCas13b and RfxCas13d, we show that these two enzymes exhibit distinct patterns of position-dependent mismatch sensitivity, with spacer regions highly sensitive or resilient to mismatches. This is unsurprising given the low sequence homology between these two Cas13 orthologs. We speculate that minor mismatches at sensitive spacer regions that abolished the silencing activity may have compromised the catalytic activity through the creation of steric hindrance that may have prevented structural rearrangement. These minor mismatches are predicted to have limited effect on the binding or dissociation kinetics due to the extensive spacer-target basepairing and the limited change in the stability of the RNA-RNA duplex (free-energy). This hypothesis remains

to be tested in future work using sensitive biophysical binding and cleavage assays that can decouple binding and cleavage steps.

We updated the revised the manuscript with these new data.

Quote: *“Next, we benchmarked the mismatch tolerance of PspCas13b with the commonly used RfxCas13d ortholog. As anticipated, RfxCas13d crRNA1 (used in Figure 3I; WT) efficiently silenced the mCherry transcript. Next, we generated 14 additional RfxCas13d crRNA mutants that harbour either consecutive or non-consecutive mismatches with the target at various spacer positions. As opposed to PspCas13b, RfxCas13d tolerated consecutive 3-nt mismatches at the 5’end of its spacer but failed to tolerate consecutive 3-nt mismatches at other positions (Supplementary Fig. 9a). Single nucleotide mismatches were fully tolerated at positions 6 and 12, but were only partially tolerated at position 16. Two non-consecutive nucleotide mismatches were also partially tolerated (position 8, 16). 3, 4, and 5 non-consecutive nucleotide mismatches led to a complete loss of silencing (Supplementary Fig. 9b). Overall, by comparing the mismatch tolerance profiles of PspCas13b and RfxCas13d, we show that these two enzymes exhibit distinct patterns of position-dependent mismatch sensitivity, with specific spacer regions highly sensitive or resilient to mismatches. This is unsurprising given the poor sequence homology between these two Cas13 orthologs.”*

Relatedly in the abstract and throughout the text you claim 30-nt. guides are inherently more specific- while this is true in terms of finding unique sites in the transcriptome, it’s not the only consideration. One also needs to factor in an idea known as the “excess energy” phenomenon where shorter crRNAs are more sensitive to mismatches as the per base energetic penalty for a mismatch relative to the free energy of the whole hybridization interaction is proportionally higher. More can be read about this here: <https://doi.org/10.1016/j.cels.2016.12.010>.

→ We fully agree with the referee that the concept of “excess energy” could lead to increased off-targeting, and the usefulness of a sticky kinetic regime to engineer RGN capable of single-base precision targeting. In fact, in a recent preprint (Shembrey et al, BioRxiv, 2023), we used a similar concept to create Cas13 crRNAs design capable of silencing oncogenic SNVs (KRAS G12, NRAS G12, and BRAF 600) with single-base precision.

The assays in this work determine the overall RNA cleavage in eukaryotic cells but are unable to decouple the binding (K_{on}/K_{off}) from cleavage (K_{cat}) steps. Our prediction of “specificity” is purely based on the probability of occurrence of binding sites transcriptome-wide that could basepair with the spacer while accounting for maximum mismatch tolerance. When PspCas13b cleavage is impaired due to unpaired bases, we are unable to determine whether it is due to poor binding, poor cleavage, or both. It would be of great value to perform future bulk and single-molecule kinetic measurements (e.g., smFRET) to determine how spacer-target mismatches can affect the binding and cleavage kinetics with high spatiotemporal resolution.

Of note, when we used a full-length guide that fully basepairs with BCR-ABL1, we observed potent on-target silencing with no evidence of off-targeting against other endogenous transcripts (see proteomic data [Redacted]). Additional data in this revised manuscript (dPspCas13b or crRNA alone) further indicate that the on-target activity is mediated by the nuclease domain of PspCas13b. Although this proteomic data doesn’t offer direct insights into target binding and

cleavage kinetics, the lack of any off-target effect highlights the precision silencing capability of this enzyme in human HEK 293 cells.

Have the author's considered generating shorter crRNAs and carrying out mismatch analysis? I think this is an important consideration if you're trying to claim superior specificity.

→ As the referee suggested, we generated crRNA variants with truncated spacer sequences

(Suppl. Figure 9c-9d). We confirmed that truncating the 5' end of the spacer leads to a drastic reduction in the silencing efficiency of PspCas13b while truncating the 3' end has a moderate impact on the silencing activity. Interestingly, these new experiments with truncated spacers suggested an unexpected effect of truncated spacers versus mismatched full-length (30-nt) spacers. Unpaired nucleotides at 5'/3' regions of the spacer appear to have a pronounced negative impact on the silencing efficiency compared to truncated guides. This suggests that unpaired nucleotides within the spacer-target interface may create steric hindrances that constrain the conformational change required for HEPN nuclease domain activation (Suppl. Figure 9c-9d).

Suppl. Figure 9

We further tested the mismatch tolerance of a truncated PspCas13b spacer (27-nt instead of 30-nt). Overall, the truncated spacer exhibited lower tolerance to 3-consecutive mismatches as well as non-consecutive mismatches. Again, we observed that mismatch tolerance is position-dependent. Overall, these new data support our conclusion that PspCas13b silencing activity and specificity are dependent on the base pairing of large parts of its spacer with the target. We updated the manuscript with this new data.

Quote: *"We further questioned whether introducing mismatches versus truncations in PspCas13b spacers at either their 5' or 3' ends would lead to similar or distinct silencing profiles. The truncation of a 3-nt motif or longer from the 5' end led to a substantial loss of silencing. Conversely, the truncation of a 3-nt motif from the 3' end did not reduce the silencing efficiency, whereas the excision of 6, 9, 12, and 15 nucleotides led to a gradual loss of silencing activity (Supplementary Fig. 9c-9d). Interestingly, the comparison of mismatched and truncated spacers suggests that unpaired nucleotides within the spacer-target duplex may create further steric hindrances that exacerbate the loss of silencing activity.*

Next, we questioned whether mismatch tolerance can be influenced by the length of the spacer. To test this hypothesis, we used a 27-nt spacer sequence with a truncated 3' end ($\Delta 28-30$) that previously exhibited full silencing activity compared to the full-length spacer. Then, we generated 16 additional mutant spacers by introducing consecutive or non-consecutive mismatches at various positions. As expected, the substitution of the 5' GGG nucleotides with a CCC motif led to complete loss of silencing. The introduction of three consecutive mismatches at spacer positions 22-24 also led to near complete loss of silencing, whereas unpaired bases at spacer positions 4-6, 7-9, 10-12 led to substantial loss of silencing. Conversely, three consecutive mismatches at positions 13-15, 16-18, and 19-21 were fully tolerated and did not alter silencing activity (Supplementary Fig. 9e). Additionally, single nucleotide mismatches at spacer positions 7 and 14 were fully tolerated, whereas a single mismatch at position 21 led to a substantial loss of silencing activity. Likewise, two, three, four, and five non-consecutive mismatches led to a substantial or complete loss of activity regardless of their positions (Supplementary Fig. 9f). Together, this data suggests that shortening the PspCas13b spacer decreases its mismatch intolerance.

4. One of the most important findings in this manuscript is how the presence of the a 5' GG sequence on the crRNA leads to much higher KD efficiency. While it was briefly mentioned in the discussion that part of this may be due to the strong preference for RNA pol III has for starting U6 promotor driven transcription at a G, it should be made clearer in the results that this is likely to contribute to the performance of these gRNAs.

→ We added a statement in the result about the preference of G bases for U6 promoter.

Quote *"A previous study indicated that promoters dependent on RNA Polymerase III, such as U6, can achieve an increased transcription rate when the resulting small RNA possesses A or G bases at the 5' end³⁶."*

Testing purified in vitro transcribed crRNA goes part way to addressing the affect of this contribution, however this reviewer cannot interpret these data as they stand. The reason being that T7 polymerase also transcribes RNA most efficiently when starting transcription with a 5' GG, and from experience and evidence in the literature, starting T7 transcription with a non G can help

to generate truncation proteins due to slow initial processivity and polymerase drop off. The authors show a 1% agarose gel to demonstrate they made full length products in all cases. However, this resolution isn't sufficient in my opinion to claim there isn't a population of truncated RNAs present in the non G starting products. Did the authors notice differences in yield with these different RNA IVTs? If so, it might be worth commenting on this in the text.

In addition, I think it's pertinent to run these products out on a 12-15% UREA PAGE gel where you can get close to single nt resolution to see if the non-5'G containing RNAs contain more truncation products. If it turns out to be the case, then this likely also explains the preference for 5'G containing IVTed crRNAs as well.

→ Now we ran a 15% PAGE gel and TapeStation analysis of IVT RNA with or without 5'GG, as well as synthetic crRNA purchased from a commercial provider (*Integrated DNA Technologies - IDT*). Both techniques showed that our IVT generated one dominant RNA product with the expected size (66 nt) (see Suppl. Figure 6 in our response to Reviewer#2).

We observed 3 to 5 times greater yield of IVT crRNA incorporating 5'GG compared to their non-5'GG counterparts using our protocol detailed in the material and methods section. After RNA purification, we measured crRNA concentrations and corrected for any difference in the yield to deliver the same amount of crRNA to the cells. Gels show the yield of crRNA IVT after correction.

Quote” We performed a 6-h incubation to maximize the yield arising from small DNA templates and lower synthesis efficiency for crRNA without 5'GG, followed by DNase treatment. In-vitro transcribed crRNAs were purified with *Monarch RNA Cleanup kits (NEB, T2040L)* to remove enzymes and unincorporated nucleotides according to the manufacturer's instructions. The purified crRNAs were quantified with *NanoDrop 2000/2000c Spectrophotometers (Thermo Fisher, ND-2000)*, and the crRNA length/integrity was evaluated using *15% urea-PAGE and Agilent 2200 TapeStation (Agilent, G2964AA)* with *RNA ScreenTape (Agilent, 5067- 5576)* according to manufacturer's instruction. Of note, we observed 3 to 5 times greater yield of IVT crRNA incorporating 5'GG compared to their non-5'GG counterparts. After RNA purification, we measured crRNA concentrations and corrected for any difference in the yield to deliver the same amount of crRNA to the cells.”

5. Given NSMB places a strong emphasis on functional and mechanistic understanding of how molecular components in a biological process work together, the reviewer would have liked to see additional experiments that try to decipher in addition to transcriptional yields why 5'G crRNAs are better at RNA-targeting (if such mechanisms exist). The authors should try and obtain purified PspCas13b and carry out crRNA-binding assays (EMSA for example) as well as RNA-cleavage assays to demonstrate at what stage of the Cas13 reaction the 5'G is assisting in boosting the activity of the enzyme. If this is outside the expertise of the lab, I think it a collaboration would be required to provide a sufficient molecular understanding for this phenomenon (and for journal fit).

→ To address this comment, we purified recombinant PspCas13b and tested its cleavage efficiency *in vitro*. We first loaded PspCas13b with 5'GG synthetic crRNA and tested its cleavage efficiency against mCherry RNA sequence (100nt) labelled with a 5'end FAM probe. First, we show that RNA cleavage is dependent on the presence of PspCas13b recombinant protein and crRNA that base pairs with the target. In absence of PspCas13 protein, absence of crRNA, or the use of NT crRNA that doesn't basepair with the target there was no target cleavage (**Suppl. Figure 6g**). We then compared the cleavage activity obtained with WT or 5'GG crRNA. Consistent with data obtained from *in vivo* silencing, this *in vitro* cleavage assays revealed higher cleavage activity when PspCas13b is loaded with a crRNA containing unpaired 5'end GG sequence (**Figure 4k-4m**). We updated the manuscript with this new data.

Figure 4

We also performed the suggested EMSA assays to probe the loading process. PspCas13b exhibited similar binding affinity to both WT crRNA and 5'GG crRNA. We've updated the manuscript with EMSA assays (**Suppl. Figure 7**).

Suppl. Figure 7

have updated the manuscript with these new data.

Quote: "We questioned whether the superior RNA silencing activity obtained with crRNAs harbouring a 5'GG motif could be due to enhanced crRNA loading and/or increased cleavage activity. To test this hypothesis, we purified recombinant PspCas13b (Figure 4k) and tested its crRNA loading and RNA cleavage activity in vitro. We first pre-incubated synthetic crRNAs with increasing concentrations of recombinant PspCas13b ranging from 6.25 to 800 nM and performed electrophoretic mobility shift assays (EMSA). The data indicates that PspCas13b can bind both crRNAs with a similar affinity regardless of the 5'GG motif (Supplementary Fig. 7). Next, we developed an in-vitro cleavage assay by reconstituting a tertiary nucleoprotein complex containing recombinant PspCas13b, synthetic crRNA, and a 100-nt long mCherry RNA as a target, the latter is labelled with a 5' 6-FAM fluorescein. When loaded with a targeting crRNA (crRNA39) containing a 5'GG motif, the recombinant PspCas13b exhibited potent cleavage of the target, which resulted in two RNA fragments with sizes ranging from 40 to 50 nucleotides. This cleavage pattern was absent when we used a non-targeting crRNA or a targeting crRNA without PspCas13b protein. (Supplementary Fig. 6g). When we compared the cleavage activity obtained with wildtype or mismatched 5'GG crRNAs, the latter exhibited a higher cleavage potency in-vitro as evidenced in the appearance of two cleaved RNA fragments (Figure 4l-4m)."

6. It is great that the authors generated an algorithm for the selection of potent crRNAs for PspCas13b. Unfortunately, in my opinion the tool falls short of many other CRISPR-Cas tools as it doesn't include a off-target scoring. Given you have some sense of mismatch tolerance, would the authors consider adding an off-target scoring parameter to the algorithm to make it more useful to users?

→ We implemented the software with off-target prediction using the human transcriptome as a reference (Please see <https://cas13target.azurewebsites.net/> , Methods, and the **Suppl. Figure 11** below).

We describe this new off-target prediction function in the updated manuscript.

RESULTS – Quote: “*The web application can also assess potential off-target transcripts within the human transcriptome for the top 10 most potent spacers. When partial matches with human transcripts are found, the users are provided with the percentage of match, number of mismatches, as well as a link to the NCBI records of these transcripts (refer to Methods for more details).*”

DISCUSSION – Quote: “*In addition to the prediction of potent crRNAs, the web application can also predict potential off-target transcripts in the human transcriptome. Based on the mismatch threshold identified in the mutagenesis study, we predict that spacers with partial sequence complementarity, involving 5 nucleotide mismatches or longer, are likely to lose their silencing activity. Consequently, any off-target transcript displayed on the prediction webpage with 5 or more mismatches is unlikely to be silenced (**Supplementary Fig. 11**). On the other hand, it is advisable to avoid spacers that exhibit partial matches with other human transcripts, especially those with fewer than 5 nucleotide mismatches, as they pose a higher risk of off-targeting.*”

METHODS – Quote “*The design of in silico prediction tool is based on design principles learned from the experimental data presented in this article. The R (version 4.2.2) programming language was used for coding and the R Shiny framework (version 1.7.4) was used to develop the software application. The software program is deployed as a web application using the Microsoft Azure platform.*”

METHODS - Quote “*In addition to identifying potent crRNAs for PspCas13b, the web application also assesses potential off-target effects of the top 10 predicted potent spacer sequences based on their sequence complementarity with various RNA transcripts in the human transcriptome. The web application integrates the NCBI BLAST (Basic Local Alignment Search Tool) command line tool for this purpose and reports the on-target and off-target(s) within the human transcriptome (GRCh38.p14; Annotation Name: GCF_000001405.40-RS_2023_10 (October 2, 2023)). The webpage displays the percentage of match and number of nucleotide mismatches with other RNA molecules that possess sequence complementarity with the selected spacer sequence. The software categorizes off-target effects as nonexistent when the number of mismatches is greater than 15. Building on mutagenesis studies, the software predicts that crRNAs with partial sequence complementarity, involving 5 nucleotide mismatches or longer, are likely to lose their silencing activity. Consequently, any off-target transcript displayed on the prediction webpage with 5 or more mismatches is considered unlikely to be silenced. In cases where potential off-targets are identified, the output provides a link to the NCBI records of these human transcripts. To expedite the processing of this web application, we recommend utilizing coding sequences (CDS) with a length shorter than 1000 nucleotides as input.*”

Supplementary Fig. 11

- a. Screenshot of the web application's output when a human transcript is used as the input target sequence.

pspCas13b guide RNA design tool

	30-nt crRNA sequence	Score	On-target in human transcriptome	Off-target in human transcriptome
1	GGGTCTCATAGTTAGGTATGTCCTTCT	170	100% match with NM_172352	None
2	GGTTTAGGATACCTGGATAATTGAGACT	170	100% match with NM_172352	None
3	GGATACCTGGATAATTGAGACTGGAGGC	170	100% match with NM_172352	None
4	GGATAATTGAGACTGGAGCTGTAAGTA	170	100% match with NM_172352	None
5	GGAGGCTGTAAAGTAGGCCCTAGACCTTA	170	100% match with NM_172352	90% match, 3-nt mismatches with NM_172358
6	GGATCCCAAGTACTGTTACTGTCACAGAC	170	100% match with NM_172352	None
7	GGATCCCAAGTACTGTTACTGTCACAGACA	170	100% match with NM_172352	None
8	GGATCACAACTAAGTTACTGCATCAAGA	170	100% match with NM_172352	None
9	GGGACTGCTGGCCATTAAGGATCCGCT	170	100% match with NM_172352	None
10	GGATCCCGTATATAGGACATGTTCTCTA	170	100% match with NM_172352	None

- b. Screenshot of the web application's output when a non-human transcript is used as the input target sequence.

pspCas13b guide RNA design tool

Input Sequence

```
gataagctcatttgcgaaagatgattgacgpatcaaaacattcccaacagagctaaaaggacaa
aaagaagaaggctgaaactcaagcctaccgacagacagaagaacacgaactgacactctctct
gctgcgatttgatgattctccaacaattgcaacatcatgacgagcagtgctgactcaactcagccc
```

Spacer Length

30

Consensus sequence of potent crRNAs

GGNNNNNNNNNDDNNNNNNNNNNNNNN

Consensus sequence of ineffective crRNAs

CCCCNNNNNNCCNCCNNNNNNNNNNNN

Filtered Motif

TTTT

Examined 30-nt crRNA sequence(s) constructed from the 1257-nt input sequence. Showing upto top 10 crRNA sequences for targeting and their on-target/off-target in the human transcriptome.

	30-nt crRNA sequence	Score	On-target in human transcriptome	Off-target in human transcriptome
1	GGATTGTTGCAATTGTTGGAGAAATC	170	None	70% match. 9-nt mismatches with NM_047449442
2	GGGCCAAATGTGCAATTTGCGGCAATGT	170	None	None
3	GGAGTTGAATTTCTTGAACCTGTGCGACTA	170	None	70% match. 9-nt mismatches with NM_001143757
4	GGTAGTAGCAATTTGGTCACTGGACTGC	170	None	60% match. 12-nt mismatches with NM_001410893
5	GGTTACTGCCAGTTGAATCGAGGGTCCAC	170	None	53% match. 14-nt mismatches with NM_031857
6	GGTCCACCAACGTAATCGGGGTGCATT	170	None	None
7	GGGCAAATTTGCAATTTGCGGCAATGTT	165	None	None
8	GGACCAGTCTGCGAAGCTTGTGTACA	165	None	None
9	GGCAATGTTGTTCTTGAGGAAGTTGTACG	165	None	None
10	GGACTGCTATTGGTGTAAATGGAGCGCT	165	None	53% match. 14-nt mismatches with XM_001745325

If you are using this Cas13 guide RNA design tool in your research, please cite:
Design principles of PspCas13b for potent and off-target-free RNA silencing by Hu et al., 2023 (under review).

Copyright © 2024 Peter MacCallum Cancer Centre

7. Relatedly, the addition of the proteomics is a nice touch and a good orthogonal way to look at off-targets. Are the authors concerned at all that the wide distribution of protein half-lives across the proteome may obscure some off-target effects. I mention this because previous reports have used RNA-seq to measure PspCas13b off-target effects (see link #2 in point #1 above) and noticed a drastically different landscape of off-targets. Have the authors considered an RNA-seq experiment to make this comparison more directly?

→ Out of the approximately 4000 proteins identified through our Mass Spectrometry analysis, certain proteins were expected to exhibit a short lifespan, which would have undergone downregulation due to collateral activity 48 hours post PspCas13b delivery. However, our proteomic data revealed no significant downregulation of any protein other than the intended target.

[Redacted]

However, we do provide additional data in this manuscript comparing the collateral activity of PspCas13b and RfxCas13d using fluorescence reporters (**Suppl. Figure 10**). Consistent with the literature, RfxCas13d with target-activated nuclease exhibited significant collateral activity against eGFP (non-target RNA). However, target-activated PspCas13b did not show any collateral activity against eGFP (**Suppl. Figure 10**).

As discussed above, the transcriptomic analysis in Extended Data 9i of the previous report (link #2 in major point #1) shows that PspCas13b has no off-target activity in HEK 293T or U87 cells when comparing a targeting versus a non-targeting crRNA. This is consistent with our proteomic [Redacted].

Minor Comments

1. Abstract: Is it necessary to define an acronym for your screen (SiBTil)? It's only used once in the main text (where its first defined also), suggesting it's not really necessary to define at all.
→ The acronym SiBTil has been removed.
2. Line 55: I would avoid using the term Prokaryotic RNAi to refer to Cas13 containing systems. To me this evokes prokaryotic Argonauts. I would replace this with RNA-guided nuclease to avoid confusion. Likewise in line 57 and potentially elsewhere.
→ Prokaryotic RNAi has been replaced with RNA-guided RNA-nucleases.
3. Line 62: It might be best if you define to the reader what a "spacer" is i.e. it's the programmable part of the crRNA.
→ A definition of the spacer has been added.
4. Line 73: For completeness it should be "CRISPR-Cas enzymes"
→ Corrected.
5. Line 208 should read "CRISPR-Cas variants"
→ Corrected.
6. Line 238: The authors call the consensus sequence for potent and ineffective crRNAs "a formula" this in my opinion is an odd word choice. I would suggest "consensus sequence" or "sequence motif"
→ We corrected this terminology to "consensus sequence".
7. Line 251: confusing sentence. Please rephrase.
→ We rephrased this sentence.
Quote *"By generating predictions based on an existing dataset and verifying their accuracy in previously unexplored transcripts, these findings illustrate that our design, which relies on spacer nucleotides, is both precise and applicable across different transcripts. This showcases its effectiveness in designing crRNAs for the targeted suppression of various transcripts of interest".*
8. Line 256-260 this section is referring to the wrong figure, please correct.
→ We corrected this error.
9. Figure 2 legend: One needs to be clearer that the BCR-ABL1-mCherry is a transgene on a plasmid and it's mCherry fluorescence that is being measured, not BCR ABL expression per se. Likewise in Figure 7 its not made clear that BCR-ABL1-mCherry is a single transcript. I would argue this really isn't an endogenous protein either given it's on a plasmid. To make that claim one should target for example some CD markers and do flow like Wessel et al have previously done.
→ We clarified this point in the figure legends (Figure 2 & 7). In a future study, we will expand our investigation to other target transcripts including CD markers using various Cas13 orthologs.
10. Line 734/Methods: Very detail Methods on some standard molecular biology techniques. For

example, I'm not sure specific μL volumes of DNA used in cloning/transformations etc. is required. If anything, these should be expressed in total ng etc. amounts. It otherwise feels overly superfluous.

→ We thank the reviewer for this valuable comment. We have corrected this issue.

11. Sup fig 1A... misleading use of crRNA concentration. Is this the concentration of the plasmid correct and not the final crRNA product? IC₅₀ isn't usually calculated for plasmids the change of units from pM to fM is also confusing. One usually needs more than four points to accurately fit a IC₅₀ curve. Please clarify here and correct where necessary.

→ We acknowledge that calculating IC₅₀ values for plasmids is not a common practice, given the challenges in quantifying the number of crRNA molecules being produced from a single plasmid in the cell. Thus, we have modified the term "IC₅₀" to "relative IC₅₀". The purpose of this "relative IC₅₀" is to compare the silencing activity of each crRNA in a quantitative manner. We have provided additional clarification in the figure legend.

12. Typo in Figure 3 ("tagreting"; 3E onwards)

→ We corrected this typo.

13. qRT-PCR how was the data normalized? Was the ddCt method used? which housekeeping genes were used in each case?

→ *GAPDH* and *HSP90* were used as housekeeping genes for mCherry and BCR-ABL1 RT-qPCR assays, while 5s rRNA and *HSP90* were chosen as housekeeping genes for crRNA RT-qPCR assays. $2^{-\Delta\Delta CT}$ method was used to normalize the expression of transcripts of interest. Different housekeeping genes gave consistent results. Therefore, results normalized to *GAPDH* and 5s rRNA are shown in the figures. Now we provide these details in the Methods section.

Quote "*GAPDH* and *HSP90* were used as housekeeping genes for mCherry and BCR-ABL1 RT-qPCR assays, while 5s rRNA and *HSP90* were chosen as housekeeping genes for crRNA RT-qPCR assays. $2^{-\Delta\Delta CT}$ method was used to normalize the expression of transcript of interest. Different housekeeping genes gave consistent results. Therefore, results normalized to *GAPDH* and 5s rRNA are shown in figures. Primers for RT-qPCR are detailed in Supplementary Table 4."

Reviewer #2:

Remarks to the Author:

This manuscript by Hu and coworkers describes an approach for designing highly active and effective pspCas13b crRNAs for RNA knockdown in mammalian cells. The authors do this using a mixture of experimental and computational approaches. The authors find that pspCas13b guide RNAs with a 5' GG are more likely to be highly active, likely due to enhanced transcription of the guide RNA. The mass spec results are really nice. Overall, this is an interesting paper with findings that are of interest to the field. My major comments have more to do with contextualizing the author's results, and performing a few additional control experiments.

→ We thank the reviewer for their kind words and for offering valuable and constructive feedback, which has helped us enhance the quality of our data and manuscript. In the subsequent sections, we present a point-by-point response, addressing the comments and concerns raised by the reviewer.

Major comments:

1. In Fig. 6, the authors introduce multiple consecutive or non-consecutive mismatches into guide RNAs targeting an mCherry mRNA and observe relatively few effects on cis targeting for sets of three consecutive mismatches. This is quite different from results obtained with other Cas13 orthologs, where one typically observes dramatic shifts in Cas13's trans cleavage activity when 2 consecutive mismatches are introduced. To address this concern, the authors should perform some experiments to look at the trans/collateral cleavage, in addition to cis cleavage, using their mismatched crRNAs (this would go beyond the results shown in Fig. 7, which focus on a single, fully-matched crRNA).

→ We thank the reviewer for this comment. As opposed to RfxCas13d, PspCas13b has no or very limited Trans/collateral activity even with complete spacer-target base-pairing. To address this comment, we conducted silencing assays using fully matched or 3-nt mismatched spacers targeting mCherry. We examined the on-target activity against mCherry and the Trans/collateral activity against eGFP that was co-expressed with the target. While the fully matched and 3nt mismatched spacers showed potent silencing of the mCherry target, there was no silencing of the eGFP reporter (Suppl. Figure 10a-10c). This data corroborates our conclusion that PspCas13b has no or very limited collateral activity. By contrast, RfxCas13d with a fully-matched guide exhibited both potent target silencing (mCherry) and potent collateral activity against eGFP (non-target). 3nt mismatches with the target reduced both the on-target and collateral activity (**Suppl. Figure 10e-10g**). As a control, we also confirmed that the fully-matched mCherry-targeting crRNA could not silence eGFP in absence of mCherry expression (**Suppl. Figure 10h**). We are currently investigating the molecular basis of collateral activity of various Cas13 orthologs to be published in a separate publication (see also the response to point 7, Reviewer#1).

Supplementary Fig. 10

PspCas13b collateral activity by WT crRNA or mutant crRNA

RfxCas13d collateral activity by WT crRNA or mutant crRNA

2. For the experiments described in Fig. 4h/i, the results could be influenced by IVT activity, and not just crRNA activity per se. It seems like the crRNA concentrations vary widely, as shown in Supplementary Fig. 6. If the authors want to make claims about potency, they should use

chemically synthesized crRNAs, which have much higher purity than IVT products. Our lab, and others, have found that IVT-produced crRNAs are highly variable in their ability to be transcribed. To address this concern, the authors should confirm a few of their findings related to potency using chemically synthesized crRNAs.

→ This is similar to the question raised by reviewer 1 (point 5). In this revised manuscript, we checked the quality of IVT crRNA using 15% PAGE and TapeStation. These techniques showed consistent size (66-nt) that is comparable to the size of synthetic RNAs. We also purchased two pairs of crRNAs from IDT either with or without the 5'GG motif (**Suppl. Figure 6f**). When transfected into HEK 293T cells, synthetic crRNAs with 5'GG showed higher potency compared to wildtype counterparts that lack the 5'GG motif despite the unpaired two GG nucleotides (**Figure 4j**). This is consistent with the data we obtained with plasmid transfection and IVT crRNAs. Additionally, these crRNAs with 5'GG exhibited better cleavage activity in vitro when used with recombinant PspCas13b (**Figure 4k-4m**).

f

In vitro transcription and clean-up of crRNAs (66bp)

Figure 4

We have updated the manuscript with these new data.

Quote: "15% PAGE and TapeStation analysis were conducted to confirm the length of in-vitro transcribed (IVT) crRNAs, both with and without a 5' GG motif. The crRNAs were both 66 nucleotides in length, which is consistent with the size of synthetic crRNAs (purchased from IDT) (Supplementary Fig. 6f).

We co-transfected equal amounts of these IVT crRNAs into HEK 293T cells together with PspCas13b plasmid to compare their silencing efficiency (Figure 4h). 24h post-transfection, all four crRNAs with 5'GG sequence achieved significantly higher silencing efficiency than their unmodified counterparts. At 48h, two out of the four IVT crRNAs containing a 5'GG sequence maintained significantly superior silencing activity compared to their unmodified counterparts (Figure 4i). Similar results were obtained using synthetic commercial crRNAs with or without a 5'GG motif (Figure 4h & 4j). Together, these experiments using IVT and synthetic crRNAs showed that the 5'GG motif further enhances the silencing activity of PspCas13b beyond augmented crRNA transcription."

3. One challenge with reporter-based experiments is that they cannot distinguish between mRNA cleavage and translational inhibition of the reporter. To address this concern, the authors should perform some controls using catalytically inactivated mutants of pspCas13b to ensure that the

effects they are observing are not due to translational inhibition, RNA interference, or other Cas13-cleavage-independent mechanisms.

→ As suggested, we used crRNA alone and catalytically inactive PspCas13b (dPspCas13b) as controls. Fluorescence reporter, qPCR, Western blot, and Mass spec analysis all showed that the silencing of PspCas13b is strictly dependent on its nuclease domain. We have updated the manuscript with this new data (see **Figure 1e-1g & Figure 7b-7m** and our response to 2nd point of Reviewer#1).

4. The authors mention that they used 1% agarose gel electrophoresis to evaluate crRNA length and integrity. This method does not give high enough resolution to detect issues with crRNA integrity, which can involve missing just a few nucleotides in length. The authors should use denaturing PAGE, or other methods with greater length resolution (e.g. TapeStation, Bioanalyzer).

→ As mentioned above (major point 2 of Reviewer#1), we used PAGE gel and TapeStation to examine the quality of crRNA with higher resolution. The IVT crRNAs with or without 5'GG have a similar size (66-nt) comparable to synthetic crRNAs (**Suppl. Figure 6**).

Minor comments:

1. An intriguing finding in this manuscript is the considerable cell-to-cell variability in reporter fluorescence levels, which is particularly evident in the FACS plots shown in Fig. 5a and 5c. This heterogeneity is particularly pronounced for the inefficient crRNAs, and in the NT controls. This is a somewhat unexpected result. Is B2M expression typically this variable from cell to cell?

→ We do see some levels of variability in the fluorescence intensity obtained with the reporter constructs, which is due to transfection efficiency. A minor population of HEK 293T cells that are not transfected won't receive the reporter and Cas13/crRNA constructs and therefore won't express any fluorescence. We do correct for this variability by using a non-targeting guide as a control for the baseline of transfection efficiency. We also perform these assays in at least 3 independent biological replicates and take images from several field of views.

In the context of FACS assays and the expression levels of B2M, it is noteworthy that HEK 293T cells frequently exhibit heterogeneous B2M expression, potentially associated with their cell cycle state. Figure 5's MFI plots provide a quantitative analysis of B2M expression across 3 or 4 biological replicates.

2. In Fig. 7e, PspCas13b appears to be downregulated by nearly 2-fold. What is going on here?

→ Indeed, Western blot and Mass spec analysis showed a moderate downregulation of PspCas13b protein by 18% and 46%, respectively, which we discuss in the manuscript. We do not fully understand why there is this minor downregulation of PspCas13b. We speculate that ectopic expression of PspCas13b mRNA could make it more prone to a minor non-specific degradation by its own nuclease domains. We will investigate this further in our future work.

3. In Fig. 7e, somewhat different fold-changes in BCR-ABL1 and eGFP expression are observed.

Is there a reason for this difference?

→ Both BCR-ABL1 and eGFP show drastic and significant downregulation in the mass spectrometry data. The difference in their fold change is possibly due to the detection accuracy of mass spectrometry and/or various translation rates/stability of BCR-ABL1 and eGFP proteins.

4. In Fig. 7f, the axes labels are very confusing. Specifically, it does not make sense to have log₁₀-transformed axes that range from only 1.36 to 1.56 (this would suggest a dynamic range of only ~1.6-fold); furthermore the deviation between eGFP/BCR-ABL1 and the line x=y does not line up with the fold-changes shown in Fig. 7e. I am not an expert in mass spectrometry, so perhaps this is a standard metric I am not familiar with, but the math does not seem to line up.

→ We thank the reviewer for noticing this error, which was generated during the transformation of Mass spec raw data (log transformation was done twice, the data was log-2 transformed already before log₁₀-transformation). We corrected this error in this revised manuscript.

5. Rather than calculating probabilities of PspCas13b off-targeting, the authors should look to see if there are any predicted off-target sequences using bioinformatic approaches. This is particularly important because many human genes exist as parts of closely-related families, so the independence assumptions that underlie the calculations the authors have made are not appropriate. Gene sequences are not entirely random, nor are they independent.

→ We thank the reviewer for this important suggestion. To address this comment, we updated the guide RNA design software with an off-target prediction function that search for sequence complementarity between the top 10 spacers and the human transcriptome. When there are partial matches with transcripts in the human transcriptome, the users are provided with the percentage of match, number of mismatches, as well as a link to the NCBI records of these transcripts (See our response to the 6th comment of Reviewer#1). We updated the manuscript with a description of this new function of off-target prediction.

Reviewer #3:

Remarks to the Author:
 The authors deciphered crRNA design rules for PspCas13b that facilitate its use in in vitro RNA targeting applications. RNA interference or targeting is a field with great potential, but with a few associated challenges. Two of these challenges are the lack of specificity (off-target effects) and high collateral activity in vitro, which they seek to remediate using PspCas13a. The authors employ a pre-existing dataset to infer important guide design principles and create a simple web interface to apply these design heuristics. I congratulate the authors for their solid work in this manuscript, and I propose several suggestions (divided into major and minor points) to strengthen this manuscript and that I hope the authors find helpful.

→ We thank the reviewer#3 for their kind words and for offering valuable and constructive feedback, which has enhanced the quality of our data and manuscript. Below, we present a point-by-point response addressing the reviewer's comments and concerns.

Major points:
 - This paper aims to establish crRNA design rules for a given Cas13b ortholog. Other groups have done related work on related Cas13 proteins previously. The authors should discuss these projects both in the introduction and discussion, and state how their approach relates/differs from previous approaches. Important examples include:
 o Metsky, G. et al. Designing sensitive viral diagnostics with machine learning. Nature Biotechnology, (2022).
 o Chuai, G. et al. DeepCRISPR: optimized CRISPR guide RNA design by deep learning. Genome Biol. 19, 80 (2018).
 o Wessels, H.-H. et al. Massively parallel Cas13 screens reveal principles for guide RNA design. Nat. Biotechnol. 38, 722–727 (2020).
 o Krohannon, A. et al. CASowary: CRISPR–Cas13 guide RNA predictor for transcript depletion. BMC Genomics. (2022).
 o Guo, X. et al. Transcriptome-wide Cas13 guide RNA design for model organisms and viral RNA pathogens. Cell Genom. 1, 100001 (2021).
 o Lin, X. et al. A comprehensive analysis and resource to use CRISPR–Cas13 for broad-spectrum targeting of RNA viruses. Cell Rep. Med. 2, 100245 (2021).

→ Most of the resources listed above are developed to design efficient guides for Cas9, LwaCas13a, or RfxCas13d systems. To the best of our knowledge, there are no validated studies/tools that can accurately predict and design efficient guides for PspCas13b. As we discuss in this manuscript, various Cas13 orthologs have little sequence homology and follow distinct action mechanisms for RNA cleavage. Our study focused on uncovering the molecular basis of PspCas13b, which has unique features and little sequence homology compared to the well-studied RfxCas13d or LwaCas13a. In this revised manuscript, we discuss the resources available for other Cas13 proteins and the lack of resources for PspCas13b.

Quote *“Several studies have investigated the design principles of widely utilized Cas13 orthologs such as LwaCas13a and RfxCas13d. Employing various methodologies, including library screens and machine learning, these investigations have uncovered key design principles and led to the*

creation of bioinformatic tools that can predict efficient guides for these programmable nucleases. However, the PspCas13b ortholog belongs to a distinct subgroup of Cas13. This subgroup possesses distinctive characteristics, including an inverted orientation of space and direct repeat, as well as a limited protein sequence homology in comparison to RfxCas13b and LwaCas13a. These unique features imply that PspCas13b may exhibit distinct targeting mechanisms and design principles, aspects that have remained largely unknown.”

- Authors should discuss the results obtained and conclusions reached in the context of the published literature. For example, beyond stating the sequence preferences of the PspCas13b's crRNA, the authors should discuss 1) how these preferences/guidelines differ between PspCas13b and other Cas13b orthologs and Cas13 subtypes and 2) provide a structural explanation/hypothesis for these preferences based on available protein structures of related Cas13b proteins.

→ In this revised manuscript, we discuss the variations in design principles among Cas13 orthologs. We also discuss potential structural explanations of the enhanced activity of crRNAs harbouring unpaired 5'GG.

Quote “We speculate that the 5'GG may be buried inside a pocket within PspCas13b protein, potentially not directly engaging in basepairing with the target. Such recognition of RNA termini by specific protein pockets is not uncommon, as analogous interactions have been reported for other enzymes involved in eukaryotic RNA interference including Dicer (5'/3' pockets)⁴⁸ and Argonaute (MID/PAZ pockets)^{3,49}. This potential interaction between the 5'GG motif of a crRNA and protein pocket could effectively lock the enzyme in an active state, thereby enhancing its catalytic activity. Future structural, biophysical, and biochemical studies are needed to experimentally test these hypotheses and shed light on the mechanisms by which the 5'GG motif enhances the cleavage activity of PspCas13b.”

Wherever appropriate, authors should include a crRNA-only control (without Cas13 plasmid). This controls would be appropriate in most panels. Recent reports have suggested that crRNAs by themselves can mediate RNA knockdown and modify protein expression under certain circumstances. See Sharma V. et al. CRISPR guides induce gene silencing in plants in the absence of Cas. BMC Genome Biology, 2022 for an example.

→ As discussed above, we added new data comparing the silencing activity obtained with catalytically active PspCas13b, catalytically inactive PspCas13b (dPspCas13b), or crRNA alone. Fluorescence reporter, qPCR, Western blot, and Mass spec analysis all showed that the silencing of PspCas13b is strictly dependent on its nuclease domain. We have updated the manuscript with this new data (see **Figure 1e-1g & Figure 7b-7m** and our response to 2nd point of Reviewer#1).

- With regards to the collateral activity, there are more sensitive RNA-based methods to probe this activity (as the authors appropriately discuss). I believe the proteomic approach could be relevant here as well, but it is important the authors A) include a crRNA only control using either efficient or inefficient guides (without Cas13) and B) perform the same proteomics approach using RfxCas13d and another Cas13 enzymes with demonstrated collateral activity in vitro. This will be important to assess the sensitivity and specificity of the proteomics approach.

→ We have now performed additional mass spectrometry control experiments to validate its sensitivity. These additional controls included expressing dPspCas13b and a crRNA individually. Only catalytically active PspCas13b showed precise silencing of the target, while inactive dPspCas13b and crRNA used alone failed to silence the target and did not affect the expression of ~4000 co-expressed endogenous proteins. We believe these additional datasets strengthen our conclusions (see updated **Figure 7** in our response to the 2nd comment of Reviewer#1).

As for testing the on-target and collateral activity of other orthologs, the fluorescence reporter assay in **Suppl. Figure 10** suggests that RfxCas13d has pronounced collateral activity against non-target eGFP. This is consistent with previous reports (Ai et al, 2022; Shi et al, 2023; Li et al, 2023). However, target-activated PspCas13b did not show any collateral activity against the non-target eGFP RNA (see **Suppl. Figure 10** above in our response to reviewer#2).

In a future study, we will investigate the on-target, off-target, and collateral activity of various Cas13 orthologs tagged with NLS or NES tags, in various human cell lines and primary cells. As this study requires a significant amount of time and optimizations, it will be published in a separate manuscript.

- When performing RNA knockdown in vitro, the sequence of the target RNA cannot be modified. In that case, could you still introduce the GG sequence at the beginning of the crRNA spacer (thereby introducing two synthetic mismatches) and still get efficient knockdown?

→ The reviewer is correct. A synthetic 5'GG sequence can significantly enhance the silencing efficiency of inefficient crRNAs that lack a 5'G-rich motif. The data in Figure 4 shows that synthetic mutations in the spacer can rescue the silencing efficiency of poorly effective crRNAs targeting mCherry despite introducing mismatches with the target.

Or would you only select for guides that naturally start with the GG sequence? If the latter, wouldn't you essentially have a two-nucleotide PFS? Please clarify.

→ The strategy of design depends on the targeted window. When silencing an entire transcript, there are numerous options for crRNA design, allowing the selection of a crRNA that inherently commences with a 5'GG sequence and can fully basepair with the target. Conversely, when the target window is confined to a specific location on a transcript (e.g., targeting a specific RNA isoform, single-nucleotide variants (SNV), or the breakpoint of gene fusions), the options for guide design with a natural 5'GG motif become limited. In such instances, we opt for the use of a synthetic 5'GG to bolster the silencing efficiency of otherwise ineffective crRNA. We further clarified this point in the manuscript.

Quote "When seeking to silence a selected transcript, there are usually numerous options for crRNA design, allowing the selection of a crRNA that intrinsically commences with a 5'GG and can therefore fully basepair with the target. Conversely, when the target window is restricted to a specific location on a transcript (e.g., targeting a specific RNA isoform, single-nucleotide variants, or the breakpoint of gene fusions), the options for crRNA design with a natural 5'GG motif are more limited. In such instances, introducing a non-cognate 5'GG can augment the silencing efficiency of an otherwise ineffective crRNA."

- In the methods section, please include the following:
 - o For the RNA silencing assays with IVT'd crRNAs, the text states that 200ng of crRNA were selected, which differs from that stated in the figures (1-20ng). Please clarify this discrepancy.

→ In Figure 1, 1-20ng refers to plasmid transfection (plasmid encoding crRNA sequence), while for IVT guide transfection we used 200ng of crRNA. We used a lower quantity of plasmids because the cellular transcription machinery can make excess guides in the cells from the provided plasmid template, while IVT guides delivered as RNA molecules are not amplified by the cell and less stable compared to plasmids. We further clarified this point in the Methods section.

Quote *“Similar transfection conditions in HEK 293T were used except we substituted crRNA plasmid with 200 ng of IVT crRNAs or 400 ng of chemically synthesized crRNAs (IDT). The choice of 200 ng of IVT crRNA and 400 ng of chemically synthesized crRNAs is based on the optimization assays where 200 ng IVT crRNA and 400 ng of chemically synthesized crRNAs achieved good silencing.”*

- o Include pertinent information about RfxCas13d use (used in Figure 3).

→ Now we provide additional information about RfxCas13d in the Result and Methods sections.

Results: Quote *“RfxCas13d is the most commonly used Cas13 ortholog for RNA silencing, thanks to its compact size and robust silencing activity^{31,33–35}. Previous studies have used library screening and machine learning approaches to uncover hidden design principles of RfxCas13d. We sought to compare the efficiency of our design of PspCas13b to the benchmark crRNA design tool that is available for RfxCas13d33 (Figure 3k).”*

Methods: Quote *“Similarly, RfxCas13d crRNA spacers were designed as single stranded forward and reverse DNA oligos containing AAAC and AAAA overhangs, respectively, allowing for ligation into BsmBI (NEB, R0739S) digested plasmid that encodes RfxCas13d direct repeat (Addgene #138150, a gift from Neville Sanjana lab).*

Quote *“All transfection experiments were performed using an optimized Lipofectamine 3000 transfection protocol (Thermo Fisher, L3000015). For RNA silencing in HEK 293T, cells were plated at approximately 30,000 cells/100 μ L/96-well in tissue culture treated flat-bottom 96-well plates (Corning) 18 h prior to transfection. For each well (unless stated otherwise), a total of 100 ng DNA plasmids [22 ng of PspCas13b-NES-3xFLAG-T2A-BFP (addgene #173029) or pC0046-EF1a-PspCas13b-NES-HIV (addgene #103862), or pC0049-EF1a-dPspCas13b-NES-HIV (addgene # 103865) or pLentiRNACRISPR-007-RfxCas13d (addgene #138149) , 22 ng crRNA plasmid, and 56 ng of the target gene plasmid were mixed with 0.2 μ L P3000 reagent in Opti-MEM Serum-free Medium (Thermo Fisher, 31985070) to a total of 5 μ L (mix1). Separately, 4.7 μ L of Opti-MEM was mixed with 0.3 μ L Lipofectamine 3000 (mix2). Mix1 and mix2 were added together and incubated for 20 min at room temperature, then 10 μ L of transfection mixture was added to each well. Supplementary Table 5 summarizes the transfection conditions used in 96, 24, and 12-well plates. After transfection, cells were incubated at 37C°, 10% CO₂, and the transfection efficacy was monitored 24-72 hours post-transfection by fluorescent microscopy. For the RfxCas13d silencing assays, the culture media was supplemented with a final concentration*

of 1µg/ml doxycycline at the moment of transfection.”

Minor

points:

- Ref. 31 is not relevant for the claim made in lines 97-98. Please update reference.

→ This reference has been corrected.

- Authors should include BFP channel images from Figure 1 as supplementary material to ensure adequate and even expression of Cas13a across cells/conditions.

→ We included BFP images (**Suppl. Figure 1**) illustrating transfection efficiency and expression levels of PspCas13b. Additionally, we incorporated Western blot (WB) analyses in both **Figure 1** and **Figure 7**, demonstrating consistent levels of PspCas13b protein expression across different conditions.

Suppl. Figure 1

- In line 227, please clarify what is meant by “filtered” and “unfiltered” samples.

→ In the context of this study, 'filtered' refers to the process of distinguishing between efficient crRNAs (those achieving more than 90% silencing compared to a non-targeting crRNA) and inefficient crRNAs (with less than 50% silencing) from the initial crRNA library. On the other hand, 'unfiltered' encompasses all crRNAs before any selection. We have elaborated on this distinction in the manuscript for clarity.

Quote *“To elucidate the significance of G and C bases at specific positions within the spacer sequence, we performed unbiased analyses of nucleotide composition at all 30 positions of the spacer in both highly potent and ineffective crRNA cohorts. We utilized weight matrix plots and Delta probability analysis to compare spacer nucleotide composition at all positions between the filtered groups (segregated into efficient and inefficient categories) and the unfiltered samples (comprising the entire crRNA cohort before any selection) (Figure 3c-3d; Supplementary Fig. 4f-4g).”*

- In Supplementary Table 2 (sequences for oligos used), please include a column detailing what figures & panels used each sequence.

A column has been added to Supplementary Table 2.

[Redacted]

Decision Letter, first revision:

Message: Our ref: NSMB-A47609A

13th Feb 2024

Dear Dr. Fareh,

Thank you for submitting your revised manuscript "Single-base tiled screen reveals new design principles of PspCas13b for potent and off-target-free RNA silencing" (NSMB-A47609A). It has now been seen by the original referees and their comments are below. The reviewers find that the paper has further improved in revision, and therefore we will be happy to accept it in principle in Nature Structural & Molecular Biology, pending minor revisions to satisfy the referees' final requests and to comply with our editorial and formatting guidelines.

We are now performing detailed checks on your paper and will send you a checklist detailing our editorial and formatting requirements in about two weeks. Please do not upload the final materials and make any revisions until you receive this additional information from us.

Sincerely,

Dimitris Typas
Associate Editor
Nature Structural & Molecular Biology
ORCID: 0000-0002-8737-1319

Reviewer #1 (Remarks to the Author):

The authors have addressed my comments and concerns appropriately, and greatly improved the manuscript overall in response to all reviewers. Even after long revisions, this works remains timely and important.

Reviewer #2 (Remarks to the Author):

The authors have addressed all of my concerns with their revision. I have no remaining concerns.

Reviewer #3 (Remarks to the Author):

Using a combination of experimental and computational approaches, Hu et al. have deciphered crRNA design rules PspCas13b, a less widely used Cas13 protein with interesting attributes for in vitro RNA targeting applications. The authors employ a pre-existing dataset to infer important guide design principles and create a simple web interface to apply these design heuristics.

The authors did a great job at addressing reviewer comments. They have conducted additional experiments (including important controls), clarified key points and strengthened their conclusions. As a result, their manuscript is a lot stronger now.

I have no further comments or requests.

Final Decision Letter:

Message: 15th May 2024

Dear Dr. Fareh,

We are now happy to accept your revised paper "Single-base tiled screen unveils design principles of PspCas13b for potent and off-target-free RNA silencing" for publication as an Article in Nature Structural & Molecular Biology.

Due to the importance of these deadlines, we ask that you please let us know now whether you will be difficult to contact over the next month. If this is the case, we ask you provide us with the contact information (email, phone and fax) of someone who will be able to check the proofs on your behalf, and who will be available to address any last-

minute problems.

Your paper will be published online soon after we receive proof corrections and will appear in print in the next available issue. You can find out your date of online publication by contacting the production team shortly after sending your proof corrections.

Please note that *Nature Structural & Molecular Biology* is a Transformative Journal (TJ). Authors may publish their research with us through the traditional subscription access route or make their paper immediately open access through payment of an article-processing charge (APC). Authors will not be required to make a final decision about access to their article until it has been accepted. Find out more about Transformative Journals

Sincerely,

Dimitris Typas
Senior Editor
Nature Structural & Molecular Biology
ORCID: 0000-0002-8737-1319